# Are flood damage models converging to "reality"? Lessons learnt from a blind test

Daniela Molinari[1], Anna Rita Scorzini[2], Chiara Arrighi[3], Francesca Carisi[4], Fabio Castelli[3], Alessio Domeneghetti[4], Alice Gallazzi[1], Marta Galliani[1], Frédéric Grelot[5], Patric Kellermann[6], Heidi Kreibich[6], Guilherme S. Mohor[7], Markus Mosimann[8], Stephanie Natho[7], Claire Richert[5], Kai Schroeter[6], Annegret H. Thieken[7], Andreas Paul Zischg[8] and Francesco Ballio[1]

[1] Department of Civil and Environmental Engineering, Politecnico di Milano, Piazza Leonardo da Vinci 32, 20133, Milano, Italy

[2] Department of Civil, Environmental and Architectural Engineering, University of L'Aquila, Via Gronchi 18, 67100, L'Aquila, Italy

[3] Department of Civil and Environmental Engineering, University of Florence, Piazza San Marco 4, 50121, Firenze, Italy

[4] Department of Civil, Chemical, Environmental and Material Engineering, University of Bologna, Viale Risorgimento, 2 - 40136, Bologna, Italy

[5] G-EAU, Univ Montpellier, AgroParisTech, CIRAD, IRD, INRAE, Montpellier SupAgro, Montpellier, France

[6] GFZ German Research Centre for Geosciences, Section Hydrology, Telegrafenberg, 14473, Potsdam, Germany

**[7]** Institute of Environmental Science and Geography, University of Potsdam, Karl-Liebknecht-Strasse 24-25, 14476, Potsdam, Germany

[8] Institute of Geography, Mobiliar Lab for Natural Risks, Oeschger Centre for Climate Change Research, University of Bern, Hallerstrasse 12, 3012, Bern, Switzerland

*Correspondence to*: Daniela Molinari (daniela.molinari@polimi.it)

**Abstract.** Effective flood risk management requires a realistic estimation of flood losses. However, available flood damage estimates are still characterised by significant levels of uncertainty, questioning the capacity of flood damage models to depict real damages. With a joint effort of eight international research groups, the objective of this study was to compare, in a blind validation test, the performances of different models for the assessment of the direct flood damage to the residential sector at the building level (i.e. micro scale). The test consisted in a common flood case study characterised by high availability of hazard and building data, but with undisclosed information on observed losses in the implementation stage of the models. The selected nine models were chosen in order to guarantee a good mastery of the models by the research teams, variety of the modelling approaches and heterogeneity of the original calibration context, in relation to both hazard and vulnerability features. By avoiding possible biases in model implementation, this blind comparison provided more objective insights on the transferability of the models and on the reliability of their estimations, especially regarding the potentials of local and multi-variable models. From another perspective, the exercise allowed to increase awareness on strengths and limits of flood damage modelling, which are summarised in the paper in the form of take-home messages from a modeller's perspective.

## 1 Introduction

Efficient and effective flood risk management requires a realistic estimation of flood losses, implying the use of reliable models

for flood hazard, damage and risk assessment (Meyer et al., 2013; Gerl et al., 2016; Zischg et al., 2018; Wagenaar et al., 2018; Molinari et al., 2019). Although several hydraulic models are available (Teng et al., 2017), their variety seems to be overtopped by the variety of flood damage models as, according to Gerl et al. (2016), only in Europe, 28 models (including 652 functions) exist to assess flood losses, whereas almost half of them focus on residential buildings.

Even within the residential sector and with respect to direct damage (i.e. damage due to the direct contact with the flooding
water), the diversity of approaches is manifold. First, the models are classified according to the intended spatial scale of the analysis: while micro-scale models refer to the individual exposed building, meso-scale models work at more aggregated scales, like land use or administrative units, with large-scale spatial units (like regions or countries) being at the base of macro-scale models (Merz et al., 2010).

A second difference lies in the approach adopted for model development, with empirical models using damage data collected
after flood events (e.g. Merz et al., 2004) and synthetic approaches implementing information collected via what-if-questions (e.g. Penning-Rowsell et al., 2005). Still, both categories are characterised by a variety of methods; for example, empirical data can be interpreted by means of different statistical and mathematical tools, ranging from simple regression (e.g. Merz et al., 2004) to more sophisticated machine learning algorithms and data mining approaches (e.g. Merz et al., 2013; Amadio et al., 2019). A distinction can also be made between absolute and relative damage models: the first directly return a value in a
specific currency (Dottori et al., 2016; Rouchon et al., 2018), while relative damage models estimate the physical vulnerability or the degree of loss of an exposed asset (Fuchs et al., 2019a), to be multiplied by its monetary value to assess the damage. Linked to this point is the question of what is defined as exposure in the models: besides the distinction whether a model relies on the value of the whole building or just of the affected floors, it is also important to know if, for instance, the basement is considered as well. Moreover, exposure assessment may differ regarding the monetary value, whether it is based on e.g. market
or replacement values (Röthlisberger et al., 2018), rather than full replacement costs or depreciated values (Merz et al., 2010). A final important difference among the models lies in the number and type of considered input parameters, i.e. on model complexity. Simplest damage models (referred to, in the following, as "low-variable models") take into account a few number of variables, mostly the water depth at building location as well as building area and its monetary value (only in case of relative models). Even in their simplicity, these models can significantly differ from each other, due to the distinct shapes of the
underlying damage functions, e.g. square root function (Dutta et al., 2003; Carisi et al., 2018), beta distribution function (Fuchs et al., 2019b) or graduated function (Jonkman et al., 2008; Arrighi et al., 2018a). On the contrary, multi-variable models consider numerous hazard and exposure/vulnerability input factors and, consequently, are supposed to be more accurate when detailed data is available (Thieken et al., 2008; Schröter et al., 2014; Wagenaar et al., 2017; Amadio et al., 2019). Nevertheless, simple models tend to be the most widely used, due to their ease for implementation and low requirements for input data.
Hence, flood damage modellers have always to envisage the trade-off in the model choice, i.e., using a complex, probably more accurate model with specific data requirements, or a simple, probably less accurate one that can be applied without extensively available data. However, it has been shown that even a small ensemble of models outperforms individual models, with the additional advantage of providing uncertainty information (Figueiredo et al., 2018).

What most models have in common is that they are calibrated in specific contexts, usually representative of a certain spatially

limited region. In many cases, instead, validation of flood damage models is lacking (Merz et al., 2010; Gerl et al., 2016; Molinari et al., 2019). Where it is not lacking, the data used for model validation are often either a subset of the dataset used for calibration or are collected in the same region or country of model development. This implies that, even if a model has been locally validated, it is not necessarily correct to apply it to any other region, unless this latter reflects the context for which the model was derived. For instance, the application of a damage model that has been developed for alpine areas (i.e. house

building tradition of the European Alps and flood processes involving significant sediment transport) to a coastal country like the Netherlands, and vice versa, is prone to lead to large discrepancies from reality (e.g. Cammerer et al., 2013). Hence, flood damage models need to be tested in regions other than those where they were calibrated in, to be confident with their transferability in space.

Nevertheless, what all models and modellers deal with is the lack of data for model calibration and validation (Merz et al.,

2010; Jongman et al., 2012; Meyer et al., 2013; Molinari et al., 2019). The overall economic impact of a flood is hardly reproduced by ex-post data and then biases have also to be taken into account when transferring models to different regions, e.g. due to different insurance conditions, uncompleted claims, etc.; moreover, even years after flood events, monetary losses can be revised due to long-term recovery: as an example, monetary losses of the 2013 flood in Germany were estimated at 6.7 M€ in 2013 (Deutscher Bundestag, 2013) and changed over the following years to 8.2 M€ (Bundesministerium für Verkehr

und digitale Infrastruktur, 2016). For this reason, comparative studies over a broad range of test cases (i.e. different validation datasets) are essential for acquiring a thorough understanding of the performances of the modelling tools that could help in enhancing the confidence in their reliability.

The aim of this study is to contribute to the understanding of models' transferability and reliability by testing and comparing different damage models in a blind validation test. This joint effort of eight international research groups consists in a common

flood case study characterised by high availability of hazard and building data, but with undisclosed information on observed losses in the implementation stage of the models. Tested models have been chosen among those mastered by the authors; indeed, the authors were either developers of the models or experienced users with significant knowledge of them, in order to prevent any possible bias in the results that could arise from an incorrect application of the models (for example, a non-expert user may misunderstand the meaning of some input variables, which would affect the final estimation).

Even though comparative analyses on the performance of damage models have become more frequent in the literature (Jongman et al., 2012; Cammerer et al., 2013; Scorzini and Frank, 2017; Carisi et al., 2018; Figueiredo et al., 2018; Amadio et al., 2019), according to authors' knowledge, this study would represent the first flood damage model comparison performed in a blind-mode. This type of comparison can provide more objective insights for a better understanding of models' capabilities and then for reducing modelling uncertainties, as already demonstrated in similar tests performed for other disciplines like

seismology, hydrology and computational fluid dynamics (Smith et al., 2004; Soares-Frazao et al., 2012; Krogstad and Eriksen, 2013; Zelt et al., 2013; Andreani et al., 2019; Ransley et al., 2019; Skorek et al., 2019). Indeed, possible biases are avoided as participants cannot be influenced by validation data, being them undisclosed in the implementation phase of the models, e.g.

by trying to adjust or tune their models, especially regarding the more qualitative input parameters, in light of observed damages.

This study focuses on micro-scale (i.e. individual item scale) direct damage assessment to residential buildings, in line with the larger availability of damage modelling approaches developed in Europe for this specific sector and scale.

As the research groups use approaches representing many different types and characteristics of models (low-variable – multi-variable; absolute – relative; graduated – regression – machine learning – synthetic), being calibrated on the basis of observed data stemming from different countries (Austria, France, Germany, Italy, Japan, Netherlands), with different landscapes and

level of complexity in exposure/vulnerability, the blind test as performed in this study can provide an in-depth understanding of the links between models features, their transferability and the reliability of the estimated damages.

In particular, the blind test allowed to investigate these specific questions, raised from the evidence supplied by the literature (Thieken et al., 2008; Cammerer et al., 2013; Schröter et al., 2014; Dottori et al., 2016; Wagenaar et al., 2017; Amadio et al., 2019): do local models (i.e. models calibrated with data from a context similar to the investigated one) outperform other

models? Do multi-variable models perform better than simplest ones and if so, why?

The paper is organised as follows. The methodology, models and case study implemented in the blind test are first presented in Sect. 2. Section 3 discusses results of the test, first by considering damage estimates obtained in a blind implementation of the models, and then by comparing damage estimates with documented losses. Answers to the specific research questions are provided in Sect. 4. Finally, in Sect. 5, evidence from the blind test is synthesised in lessons learnt (on flood damage modelling)

from a modeller's perspective, including the identification of research needs for further improvements of flood damage models.

## 2 The blind test: case study, methodology, models

The main idea behind the blind test was to evaluate the performance of different flood damage models by their implementation to a common case study, to obtain enhanced information on their transferability, validity and reliability; the test is defined "blind" as, in order to avoid bias in the estimation process, the value of the observed damage was unknown to modellers in the

implementation stage of the models. In particular, damage data were unblinded only to one group, which was the promoter of the initiative and responsible for data and results management. All required input data to reproduce the damage scenario for the examined event were made available to the participants, who were then asked to submit their results to the exercise manager in an established time frame. Once all contributions from the different groups had been gathered, observed data were disclosed, and models' performances were compared and analysed in a shared discussion between the participants.

**2.1 Case study**

The investigated context is the town of Lodi, North of Italy (Fig. 1), which was hit by a severe flood on 25-26 November 2002, caused by the overflow of the Adda River as a result of two weeks of heavy rainfalls over North-western Italy.

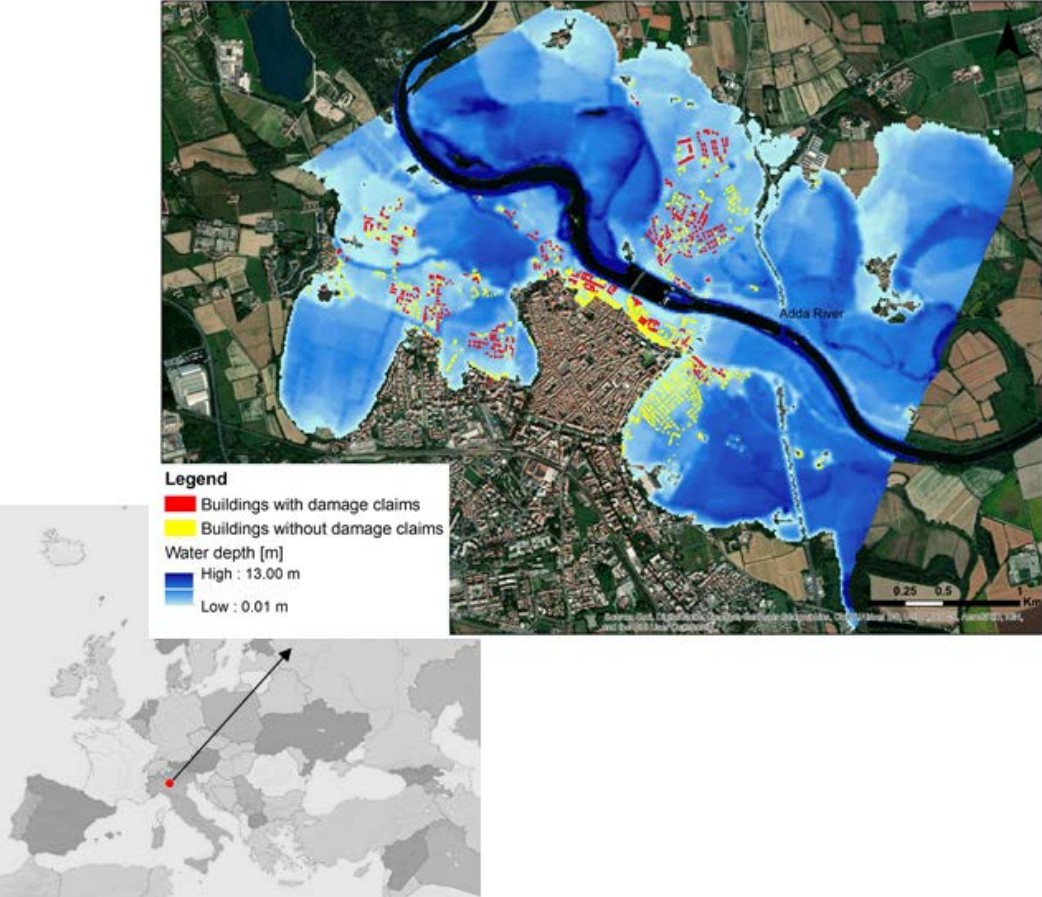

**Figure 1: Map of the flooded area and affected buildings.**

The flood caused severe damage to residential buildings, commercial activities and public services in the area, including the main hospital. Fortunately, no fatalities occurred. The event was chosen as reference for the exercise as it is well documented and characterised by a high availability of hazard, exposure and vulnerability data. In detail, with respect to the hazard, information on observed water depths was available for more than 260 points within the inundated area, deriving from indications provided by municipal technicians and by citizens in damage compensation requests, as well as from interpretation of photographs taken during or immediately after the flood. These data were used for the validation of the 2D hydraulic simulation of the event: the resulting average absolute differences between observed and calculated water depths within the inundated area ranged from 0.2 to 0.4 m, depending on the validation zone in which observed water depth data were aggregated (Scorzini et al., 2018). This is surely a possible source of uncertainty; however, reported differences could be considered to provide relatively small impacts on the damage estimation. Moreover, given that all tested damage model shared the same hazard data, this would be a common source of uncertainty that should not affect the overall results of the blind test.

Available micro-scale data on exposure and vulnerability of residential buildings are shown in Table 1.

**Table 1: available micro-scale data for the blind exercise.**

| Data | Variable | Description | Source | Year |
|------|----------|-------------|--------|------|
| *Area* [m²] | FA | Footprint area of the building | Regional topographical database | 2010 |
| *Perimeter* [m] | EP | External perimeter of the building | Regional topographical database | 2010 |
| *Basement* | BA | Presence of basement yes/no | Lodi cadastral data | 2016 |
| *Building type* | BT | Type of building (apartment, detached or semi-detached house) according to the cadastral data. | Lodi cadastral data | 2016 |
| *Finishing level* | FL | Quality of the building (low, medium or high) according to the cadastral data: | Lodi cadastral data | 2016 |
| *Building structure* | BS | Type of building structure (masonry or reinforced concrete) calculated as the most frequent value for the buildings in the census block it owns. | National Institute of Statistics (ISTAT) | 2001 |
| *Floors* | NF | Number of floors calculated as the most frequent value for the buildings in the census block it owns. | National Institute of Statistics (ISTAT) | 2001 |
| *Level of maintenance* | LM | State of conservation (low, medium or high) of the building calculated as the most frequent value for the buildings in the census block. | National Institute of Statistics (ISTAT) | 2001 |
| *Water_depth* [m] | h | Mean value of water depth in the building area. | 2D hydraulic modelling | 2018 |
| *Flow_velocity* [m s⁻¹] | v | Mean value of flow velocity in the building area. | 2D hydraulic modelling | 2018 |
| *Presence of pollutants* | q | Presence of fuel spillage or other pollutants | Claims forms / photos of the event | 2002 |
| *Replacement value* [€m⁻²] | RV | Reconstruction value of residential building given as a function of the building type and building structure of the building, based on existing literature and official studies | Cresme-Cineas-Ania | 2014* |
| *Market value* [€m⁻²] | MV | Market value of residential buildings, as a function of building type, finishing level and building location | OMI (Osservatorio del Mercato Immobiliare) – Italian real estate and property price database | 2014* |

\* for the objective of the exercise data were discounted to 2002 values

Altogether, observed damage was known for 345 of the 877 buildings in the flooded area (after hydraulic simulation; Fig. 1), as derived from claims compiled by citizens after the flood to ask for public compensation.

Claims were mostly collected by the Municipality of Lodi and, in a small part, by the Regional Authority of the Lombardy region after the event. Available claims data, in their original papery form, were then firstly acquired and successively stored in a georeferenced digital database, by a team of researchers of Politecnico di Milano in summer 2017. As regards data from the Municipality, original claims were organised in forms, including information on the owner, the address of the flooded building, its typology (e.g. apartment, single house), the number of affected floors, a description of the physical damage and its translation into monetary terms (distinguishing, for the different rooms of the building, among damage to walls, windows and doors, floor, systems and content). In few cases, information on water depth inside the building and on clean-up costs,

non-usability of the building and intangible damage (e.g. loss of memorabilia) was also inferred from the qualitative damage description in the forms. The quality/reliability of data included in the claims was not uniform, since only some of the owners justified the costs for fixing damage by means of invoices. As regards data from the secondary source (i.e. the Regional
Authority), they included limited information on the owner, the address of the flooded building and the monetary value of damage, distinguished in damage to structure and contents.

**2.2 Methodology**

The methodological approach followed in the test included the following steps:

*Step 1: identification of damage models to be tested*
The choice was based on several considerations: (i) good mastery of the models by the research team (i.e. damage models regularly used or initially developed by the groups), (ii) heterogeneity of the approaches, by considering simple and multi-variable models, empirical and synthetic approaches, absolute and relative models, and (ii) models being calibrated in a different context than the investigated one. The choice converged to the nine models described in Sect. 2.3.

*Step 2: implementation of the models to the case study in a blind mode*
The models were implemented independently by the research groups (i.e. each group applied one up to three models, according to its specific expertise) to calculate damage to all 877 buildings that were exposed to the 2002 Lodi flood, according to the inundation area simulated by the hydraulic model (Scorzini et al., 2018). All the groups used available and common data on
hazard, exposure and vulnerability, as described in Table 1. While this step was simple for Italian models (which were originally developed to work with the same kind of data available for the case study), some efforts were required for the other models, particularly in the case of multi-variable ones. This is due to a lack of correspondence/consistency among exposure and vulnerability data available in the different countries, on which damage models are usually based. For instance, correspondence had to be defined among building types classified by the Italian cadastre and the ones adopted by the German
and French models and the ones as classified by the Italian cadastre.
The damage assessment was carried out only for building structures, given that not all models are designed to simulate damage to household contents. At this step, observed losses were still blinded to the research groups in order to avoid possible bias in the estimation.

*Step 3: comparison of model outcomes*
Exposure and damage estimates supplied by the different models were compared, at the aggregated and individual level, with the main objectives of (i) understanding the weight of exposure assessment on damage calculation, and (ii) pointing out common or divergent model outcomes.

*Step 4: comparison of model features*

Models were compared in terms of trends and variance of individual damage estimates, for homogeneous classes of input variables, by considering one variable at a time. The objective was to understand whether the inclusion of more explicative variables may be considered as a possible source of variation, as well as to identify the most influencing parameters on the final output of the models.

*Step 5: comparison between estimates and observations*

This phase aimed at investigating the performances of the different models in the analysed context. Calculated damages were compared to observed losses coming from claims. The comparison was possible only for 345 of the buildings included in the flooded area, for which official claims were available.

*Step 6: analysis of claims*

Claim data were analysed with the aim of identifying potential reasons for (in-) consistencies between estimates and observations.

*Step 7: synthesis of results*

Results obtained in the previous steps were critically analysed in order to gain knowledge on model transferability and reliability of damage estimates, with respect to their implementation in a same case study, and from a modeller's perspective. The analysis was conducted jointly by all groups, in the form of brainstorming, during several remote meetings and one face to face meeting.

**2.3 Models**

The main characteristics of the selected models are summarised in Table 2 and briefly described hereinafter.

- The model developed by **Arrighi et al.** (2018a, 2018b) is a relative synthetic model which expresses monetary damage as a function of water depth and recovery cost for buildings with and without basement. A zero-damage threshold is set for a water depth lower than 0.25 m for buildings without basement. The recovery cost is assumed equal to 15 % of the exposure, calculated as the market value of the flooded floor(s) based on the footprint area. The ratio between recovery cost and market value is based on the comparison between residential prices for new buildings and buildings requiring renovation (Italian real estate data). The model was created based on expert judgement for the city of Florence (Italy) and applied both at building and census block scale (Arrighi et al. 2018a, 2018b). It has been validated through comparison with other validated models (Arrighi et al., 2018b) and ex-post damage in another Italian context (Scorzini and Frank, 2017).

- **Carisi et al. - MV** (Carisi et al., 2018) is an empirical multi-variable model, which estimates relative building losses considering six explicative variables: maximum water depth, maximum flow velocity, flood duration, monetary

building value per unit area (based on market value), structural typology and footprint area of each building (Carisi et al., 2018). Calibration data refer to the inundation event occurred in the province of Modena (Italy) in 2014, when a breach in the right embankment of the Secchia river caused about 52 km$^2$ of flooded area and €500 million losses (see, e.g., Orlandini et al., 2015). Observed losses were derived from 1330 claim forms filled by citizens and collected by authorities for the purpose of compensation, while the maximum water depth was reconstructed by means of a fully 2D hydrodynamic model; economic building values per unit area were finally retrieved by the Italian Revenue Agency reports. The model does not consider damage to basements. The model uses the Random Forest approach (Breiman et al., 1984; Breiman, 2001), which is a tree-building algorithm for predicting variables, recursively repeating a subdivision of the given dataset into smaller parts in order to maximize the predictive accuracy. In order to avoid overfitting problems, several bootstrap replica of the learning data are used, for which regression trees are learned, then aggregating the responses from all trees to estimate the final result.

- **Carisi et al. - mono** (Carisi et al., 2018) is an empirical simple model, calibrated on the previously cited 2014 Secchia flood event. The model supplies the relative damage to building (using the market value to relativize the observed monetary damage when developing the model), as a function of the maximum water depth. The model does not consider basements or garages, for coherence with the calibration context, where most of the buildings do not have these elements.

- The model developed by **CEPRI** (European Center for Flood Risk Prevention, (CEPRI, 2014a)) is a synthetic (expert-based) and multi-variable model that expresses absolute damage as the expected sum of the actions that must be performed after a flood to restore to the pre-flood state, including clean-up costs. The flood parameters taken into account are water depth and submersion duration. The considered building characteristics are the building type (single storey house, double storey house, or apartment), the floor area, the presence of a basement and its area. For each type of building, one damage curve indicates the damage to structural components, and one the damage to the furniture. Two separate damage curves are used to estimate the damage to the basements contained in houses or apartment blocks. Initially, the model was developed to estimate damage due to all types of floods. Its estimates have been compared to empirical damage due to fast rise floods (CEPRI, 2014a; Richert and Grelot, 2018) and coastal flooding (CEPRI, 2014b). The model was found acceptable in the first context, but needed calibration in the second case. The French State recommends using this model to conduct cost-benefit analyses of flood management projects (Rouchon et al., 2018).

- The model by **Dutta et al.** (2003) was chosen because it is an early example of a model that describes the relationship between flood intensity and damage. It is a simple model supplying a relative damage (i.e. the degree of loss that describes the ratio of loss to the replacement value of the whole building) based only on flood depth; basement, number of exposed floors or other exposure variables are not separate inputs for the model, but are part of its variance. The stage-damage function was calibrated with data published by the Japanese Ministry of Construction, which are based on site survey data accumulated since 1954. The validation with a flood event of 1996 showed reliable results

for urban areas. The replacement value of the building has to be provided as input data.

- **FLEMO-ps** (Flood Loss Estimation MOdel for the private household sector) is a multi-variable, rule-based model estimating relative monetary flood loss to residential buildings as a function of water depth, building type and building quality, without further differentiating between flooded floors and not explicitly considering the existence of a basement (Thieken et al., 2008). The model is empirically derived from data collected from 1697 households affected by the severe flooding of the rivers Elbe, Danube and some of their tributaries in August 2002 in Germany. It can be applied on both the micro- and the meso-scale. Model evaluations based on historical floods in Germany showed that FLEMO-ps is outperforming traditional stage-damage curves in estimating flood loss in the private household sector, except for damages caused by very high water depths (Thieken et al., 2008).

- The model by **Fuchs et al.** (2019b) is a simple model, which supplies a relative damage (i.e. the degree of loss that describes the ratio of loss to the replacement value of the whole building) considering water depth, building area (of all floors) and building (replacement) value as input variables. Differently from other models, it is a function developed for mountain areas, i.e. referring to house building tradition of the Alps and flood processes with sediment transport. It was chosen to test the transferability of a model specialised for mountain environments to a low-land situation. The model was fitted with empirical damage and hazard data. Model validation took place based on a 5-fold cross validation.

- **INSYDE** (Dottori et al., 2016; Molinari et al., 2017b) is a synthetic model based on the investigation and modelling of damage mechanisms triggered by floods, developed for the Italian context. The model is based on a what-if analysis, consisting of the simulated step-by-step inundation of the building and in the evaluation of the corresponding damage as a function of hazard and building characteristics. In total, INSYDE adopts 23 input variables, six describing the flood event and 17 referring to building features; among them, there are all the variables available for the case study and included in Table 1. For the remaining ones, default values implemented in the model were adopted in the test. The model supplies damage in absolute terms by considering the replacement/reconstruction value of damaged components, and by referring only to flooded floors (including basement, if present); however, if required, the model can supply also an estimation of relative damage. INSYDE was validated for different Italian flood events and its performance has been compared to those of other existing models (Dottori et al., 2016; Molinari et al., 2017b; Amadio et al., 2019).

- The model by **Jonkman et al.** (2008) is a simple relative damage model considering water depth and building (replacement) value of all floors as explicative variables, developed on the basis of empirical flood damage data collected in the Netherlands in combination with existing literature and expert judgment. There is no information concerning validation or the robustness of this model. The model is a combined function of content and structure loss. Therefore, to only consider damage on building structure, the original function was rescaled to possibly reach "total destruction" (degree of loss = 1).

**Table 2: main features of the models implemented in the blind test.**

| Model | Country and year of development | Hazard context of development | Considered explicative variables | Type of model | Type of results | Economic evaluation | Exposure estimation | Other features |
|---|---|---|---|---|---|---|---|---|
| Arrighi et al. | Italy, 2018 | Riverine floods | h, FA, BA, economic value of the building | synthetic | relative damage | Recovery (based on market value) | flooded floors, (considering also FL and LM) | – zero-damage threshold at water depth 0.25 m<br>– the model estimates also absolute damage |
| Carisi et al. - MV | Italy, 2018 | Riverine floods | h, v, FA, BS, economic value of the building | empirical | relative damage | market value | flooded floors (considering also FL and LM) | |
| Carisi et al. - mono | Italy, 2018 | Riverine floods | h, FA, economic value of the building | empirical | relative damage | market value | flooded floors (considering also FL and LM) | |
| CEPRI | France, 2014 | Riverine, coastal floods | h, BT, FA, BA, NF | synthetic | absolute damage | replacement value | flooded floors | – the model estimates also damage to contents (not considered here) |
| Dutta et al. | Japan, 2003 | Riverine floods | h, FA, economic value of the building | empirical | relative damage | replacement value | whole building | |
| FLEMO-ps | Germany, 2008 | Riverine floods | h, q, BT, FL, economic value of the building | empirical | relative damage | replacement value | whole building | – the model is also capable of estimating damage to household contents (not considered here) |
| Fuchs et al. | Austria/ Switzerland, 2019 | Mountain (high velocity) floods, debris flows | h, FA, economic value of the building | empirical | relative damage | replacement value | whole building | |
| INSYDE | Italy, 2016 | Riverine floods | h, v, q, FA, EP, BA, BT, FL, BS, NF, LM | synthetic | absolute damage | replacement value | flooded floors (considering FL and LM) | – the model estimates also relative damage |
| Jonkman | The Netherlands, 2008 | Riverine floods | h, FA, economic value of the building | empirical | relative damage | replacement value | whole building | |

# 3 Results

## 3.1 Implementation of the models to the case study in a blind mode

With the aim of understanding the impact of exposure estimation on damage assessment and identifying possible common features in the results, Table 3 shows the total exposure and loss figures obtained by applying the nine models to all buildings within the simulated inundation area (877 in total; see Figure 1); note that at this stage of the analysis damage observations were not considered yet for comparison purposes (see Section 2).

Total exposure estimates differ among the models by a maximum factor of 2.75. With respect to the mean exposure value,
single estimations diverge instead by a maximum factor of 1.77. These significant differences mainly result from the fact that some models calculate exposure as the monetary value of flooded floors, while others refer to the whole building (see Table 2). Indeed, focusing on the four models that consider only flooded floors (i.e. Arrighi et al., Carisi et al.-MV, Carisi et al.-mono, and INSYDE, see Table 2), total exposure estimates differ by a maximum factor of 1.22. Minor differences are due to the (non-)consideration of the presence of a basement as well as to the adoption of replacement/recovery values rather than
market values as parametric cost for the estimation. These results point out that a first source of variability among model outcomes lies in the approach for exposure assessment.

Total damage estimations differ among the modelling approaches by a maximum factor of 12.6, which is limited to 3.1 with respect to the mean value of total damage estimations, suggesting that the shape of the damage functions exacerbates the variability of models' outcomes due to exposure estimation.

Similar conclusions can be drawn when looking at individual (i.e. building by building) estimations reported in Fig. 2 (exposure values) and Fig. 3 (damage values). Individual estimations of exposure differ by a mean factor of 3.5. The models of Fuchs et al., Jonkman et al., Dutta et al. and FLEMo-ps use the replacement value of the whole building as a reference for calculating the degree of loss and are thus relying on sensibly higher exposure values than others. Individual damage estimates differ on average by a factor of 28, with the highest differences due to the models of Fuchs et al. and Dutta et al. (maximum expected
damage) and to the model of Arrighi et al. (minimum expected damage). Such results can be partly explained by the adoption of the whole building value for exposure estimation (see also Sect. 3.2) as regards high estimations, and by the zero damage threshold for water depths lower than 0.25 m for low estimations. In detail, the weight of the zero damage threshold on the final damage figure has been calculated as a percentage ranging from 7 to 32 %, depending on the considered model.

**Table 3: Estimates of the monetary value of exposed assets and damage, for all the buildings in the flooded area. The first column reports the total value of exposed assets (n.a.= not applicable). The second and the third column report, respectively, the total damage and the unit damage per m². The fourth and the fifth column report the ratio between estimates and mean value of estimates (reported in the last row), for exposed assets and damage, respectively.**

| Model | Monetary value of exposed assets [M€] | Monetary damage [M€] | Unitary monetary damage [€ m⁻²] | Monetary value of exposed assets/mean value [-] | Monetary value of damage/mean value [-] |
|---|---|---|---|---|---|
| Arrighi et al | 392 | 12 | 35 | 0.78 | 0.25 |
| Carisi et al. - MV | 368 | 20 | 80 | 0.73 | 0.40 |
| Carisi et al. - mono | 368 | 30 | 118 | 0.73 | 0.59 |
| CEPRI | n.a. | 25 | 71 | n.a. | 0.50 |
| Dutta et al. | 889 | 155 | 225 | 1.77 | 3.10 |
| FLEMO-ps | 468 | 58 | 230 | 0.93 | 1.15 |
| Fuchs et al. | 889 | 102 | 147 | 1.77 | 2.03 |
| INSYDE | 395 | 21 | 69 | 0.79 | 0.41 |
| Jonkman et al. | 889 | 29 | 42 | 1.77 | 0.58 |
| **Mean** | **502** | **50** | **-** | **-** | **-** |

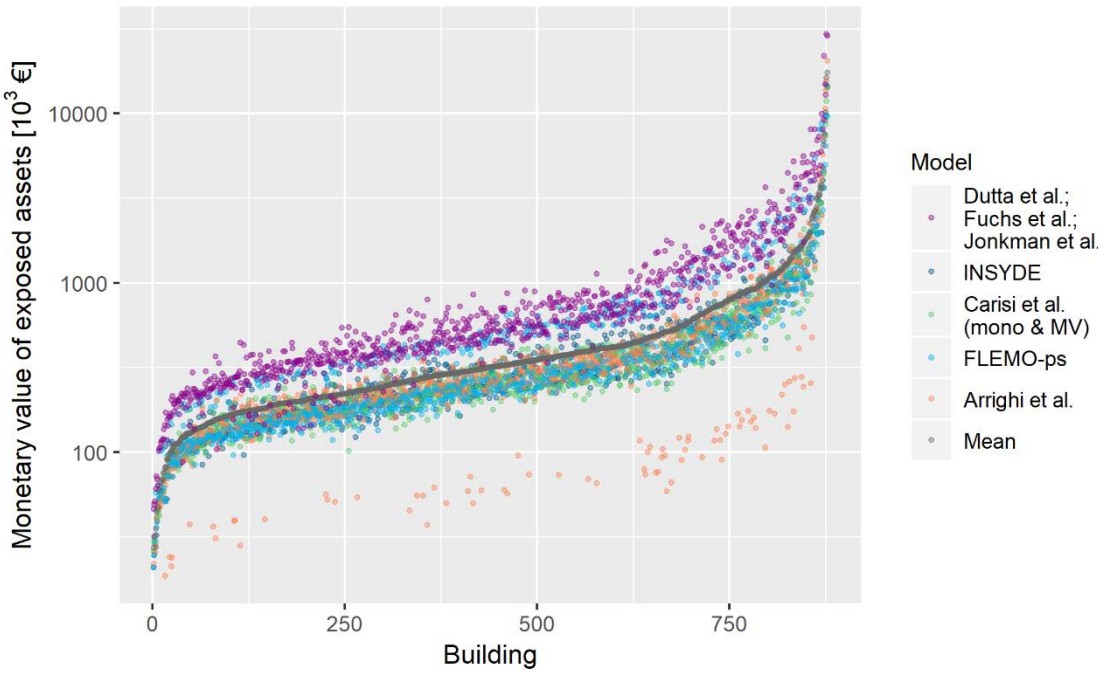

**Figure 2: Individual estimates of the monetary value of the exposed assets for all the buildings in the flooded area. Data are ordered according to increasing value of mean estimate (in grey).**

Figures 2 and 3 further highlight a common trend in exposure and damage values supplied by the different models, also confirmed in Fig. 4 and 5, showing the Pearson's correlation coefficients for individual (i.e. building by building) exposure and damage estimates. The figures show a very high correlation of exposure estimations and a weaker, but still notable, correlation of damage estimations. This finding supports previous results on the importance of damage functions in determining the main differences in model outcomes. In particular, Fig. 5 shows that a higher correlation exists between absolute damage estimates supplied by the two synthetic models INSYDE and CEPRI, among multi-variable models (INSYDE, CEPRI, Carisi et al. - MV and FLEMO-ps), and among simple models (Carisi et al. - mono, Dutta et al., Fuchs et al. and Jonkman et al.), which reflects the consistency between models based on comparable conceptual frameworks.

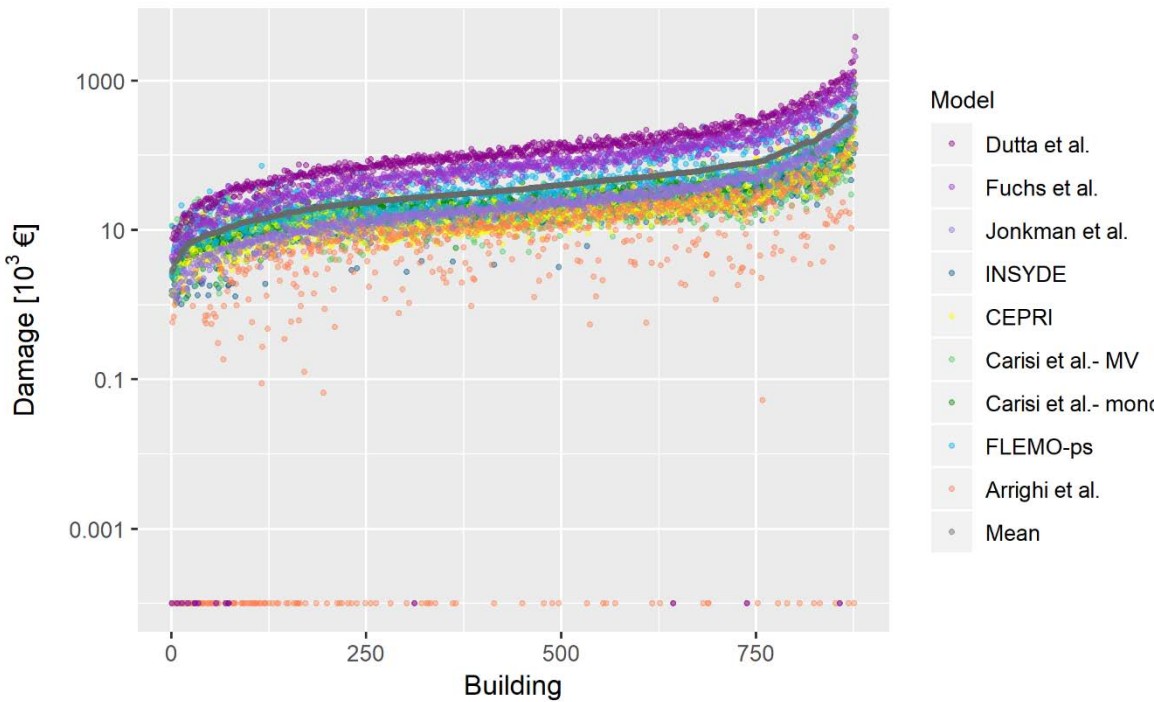

**Figure 3: Individual estimates of the monetary damage for all the buildings in the flooded area. Data are ordered according to increasing value of mean estimate (in grey). Zero damages are due to the modelling assumptions behind the specific damage models (i.e. 0.25 m water depth threshold for damage occurrence in Arrighi et al. and 0.01 m water depth threshold in Dutta et al. and Jonkman et al. to distinguish between flooding and surface water runoff)**

A comparison between correlation coefficients for absolute and relative damage estimations in Fig. 5 conversely highlights the importance of exposure assessment on the final damage figures. For instance, the low correlation among absolute damage estimates supplied by the model of Arrighi et al. with those from similar models (i.e. simple, low-variable models like Carisi et al. - mono, Dutta et al., Fuchs et al. and Jonkman et al.) can be explained by the fact that the approach adopted by Arrighi

et al. for the evaluation of exposure is considerably different from those adopted by the other comparable models; specifically, the model calculates the monetary value of damage as a function of the recovery cost, which is assumed equal to 15 % of the market value of exposed floors (see Sect. 2). Accordingly, when relative damage estimations are considered, the values of Pearson's correlation coefficient increase. The weight of exposure assessment is also evident when correlation among absolute damage estimates supplied by the four simple, empirical models (i.e. Carisi et al. – mono, Dutta et al., Fuchs et al. and Jonkman

et al.) are considered, with models of Dutta et al., Fuchs et al. and Jonkman et al. using the same exposure assessment approach (see Sect. 2) and thus being more correlated among them than with the model Carisi et al. – mono; on the opposite, when relative damage estimations are considered, the correlation coefficients for the four models are comparable. At last, the weight of exposure arises when correlation between absolute damage estimates supplied by Carisi et al. – mono versus INSYDE are considered. The couple consists of two conceptually different models (in particular, a simple, empirical model versus a multi-

variable model), but it shows high correlation. This can be explained by the adoption of very similar approaches for exposure estimation by the considered models (see Sect. 2 and Table 3); in fact, when relative damage estimates are considered correlation decreases.

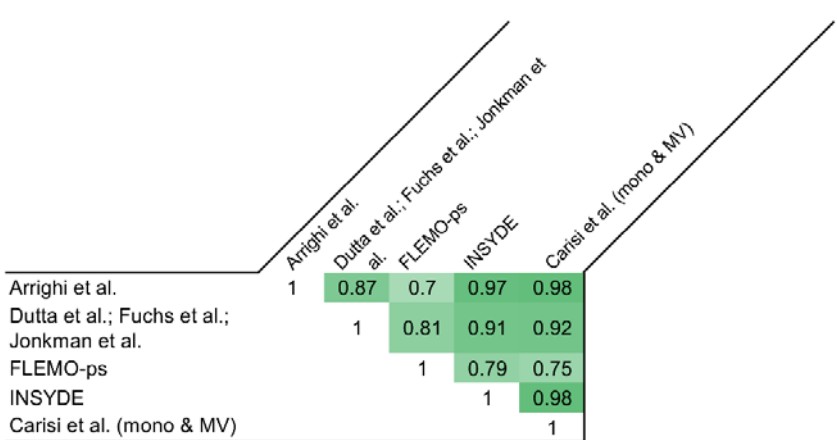

**Figure 4: Pearson's correlation coefficient for individual exposure estimates supplied by the models with reference to all the buildings in the flooded area (the darker the colour, the stronger the correlation).**

Figure 5: Pearson's correlation coefficients for absolute damage estimations (top-right of the matrix, in blue) and relative damage estimations (bottom-left of the matrix - in red) supplied by the models with reference to all the buildings in the flooded area (the darker the colour, the stronger the correlation).

## 3.2 Role of input variables in the determination of divergent models' outcomes

In order to explain the differences observed in the blind implementation, models were compared in terms of trends and variance of individual damage estimates, for classes of values of input variables, and by considering one variable at a time. The objectives of the analyses were to investigate whether the consideration of a specific input variable influences the outcome of a model with respect to the other ones, whether the inclusion of more explicative variables may be considered as a possible source of variation, and to identify the most influencing parameters on the final output of the models.

The input variables considered were: the mean value of the water depth in the building area (h), the footprint area of the building (FA), its external perimeter (EP), the presence of basement (BA), the building type (BT), the building structure (BS), the finishing level of the building (FL), the number of floors (NF), and the level of maintenance (LM). The results are shown in the boxplots reported in Fig. 6 and 7.

An expected increasing trend in damage as a function of the variables related to the extensive properties of the buildings (FA and EP) can be seen, with limited data variance in the case of those models considering other explicative variables than FA (e.g. EP), as INSYDE. As highlighted in the previous section, the models of Dutta et al. and Fuchs et al. show markedly different results, i.e. higher estimates than other models in all classes. This cannot be totally attributed to the fact that such models consider the whole building for calculating exposure, as this is true also for the model of Jonkman et al., which supplies results that are comparable with the ones of other models. Instead, one possible reason may be found in the different origins of the models. In fact, contrarily to all other models, the model of Fuchs et al. was developed for mountainous regions where floods are usually characterised by high sediment transport and deposition, which increases the damage, other variables being equal. In the case of Dutta et al. the detection of the reason for the remarkably higher damages is more elusive, given the lack of detailed information on model derivation, which makes the original model environment not known either for hazard or

exposure variables. In addition, this model is based on survey data collected since 1954 in Japan, meaning that the data used might not be consistently representative for the current flood vulnerability (and in a European environment). The general increasing variance of the estimates with FA and EP classes can be explained by the intrinsic variability of the features characterising larger buildings: they can be apartment buildings rather than semi-detached houses or big villas, with one or more floors; moreover, in the case of apartment buildings, the level of maintenance can change from flat to flat.

Figure 6 indicates the importance of BA as an influencing variable in modelling flood damage for the given event. This is particularly evident in the results provided by CEPRI and INSYDE, which estimate median damages ranging respectively from 13 600 €and 15 400 €for buildings without basement to 26 300 €and 24 500 €for buildings with basement, as opposed to the performances of other models, which did not differ significantly for the two building categories.

Regarding damage estimates for different water depth classes, Fig. 6 indicates an acceptable convergence among model results, especially for the shallower water depth classes, if excluding the results of the models of Dutta et al. and Fuchs et al. (as discussed earlier). However, larger differences are apparent for the highest water depth class (h > 1.5 m). Overall, this result seems reasonable as most of the tested models were calibrated and/or validated for flood events characterised by shallow or medium inundation depths.

Finally, as also emerged in previous studies (Wagenaar et al., 2017; Amadio et al., 2019), Fig. 7 denotes that other variables related to building features do not significantly influence model behaviour. Larger scatter is observed only for the "Apartment" category, which is intrinsically characterised by larger variability, especially in terms of extensive parameters.

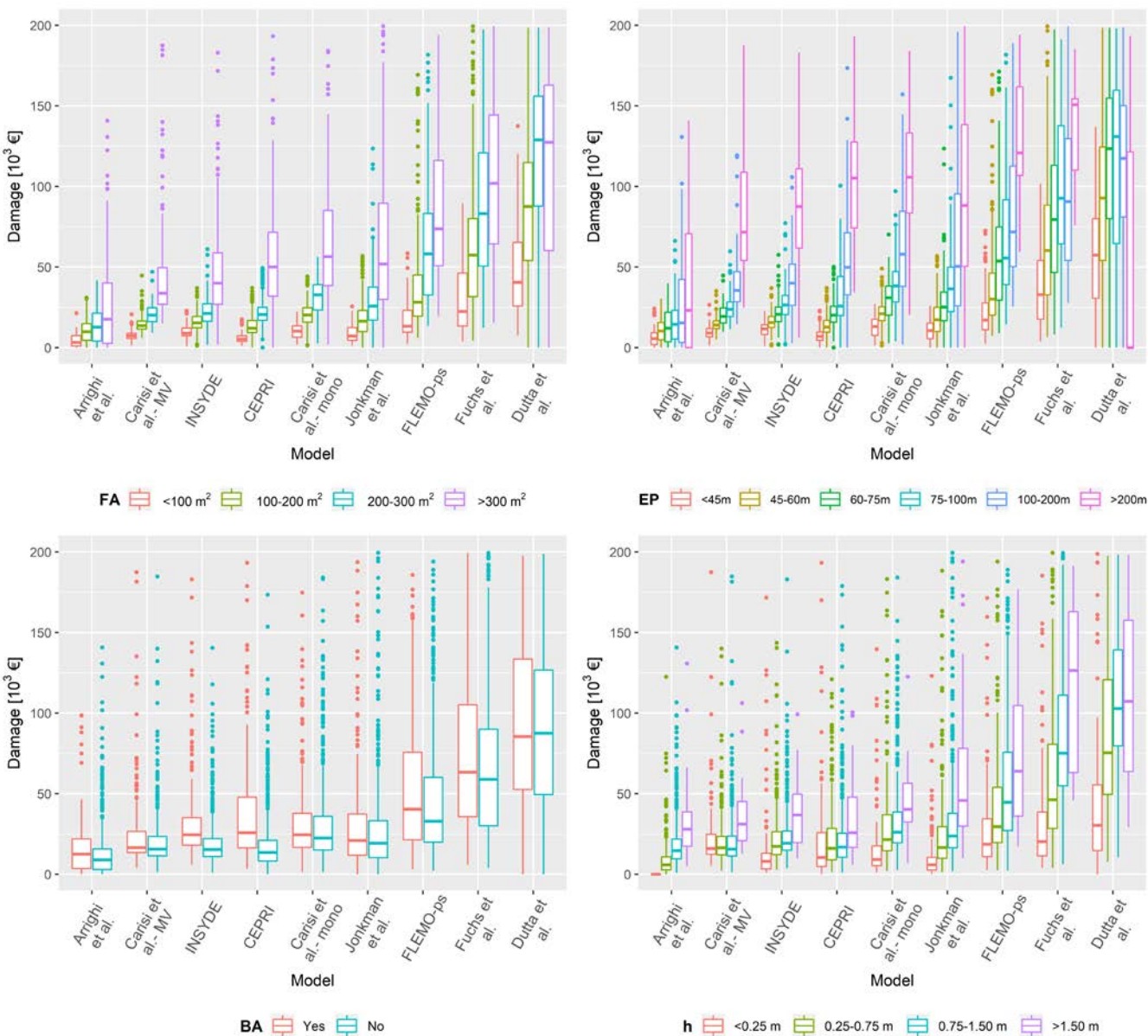

**Figure 6: Boxplots of damage estimates obtained with the tested models, for different classes of: footprint area – FA (Top-left), external perimeter – EP (Top-right), presence of basement – BA (Bottom-left) and water depth – h (Bottom-right). Models are organised according to increasing value of total damage estimates.**

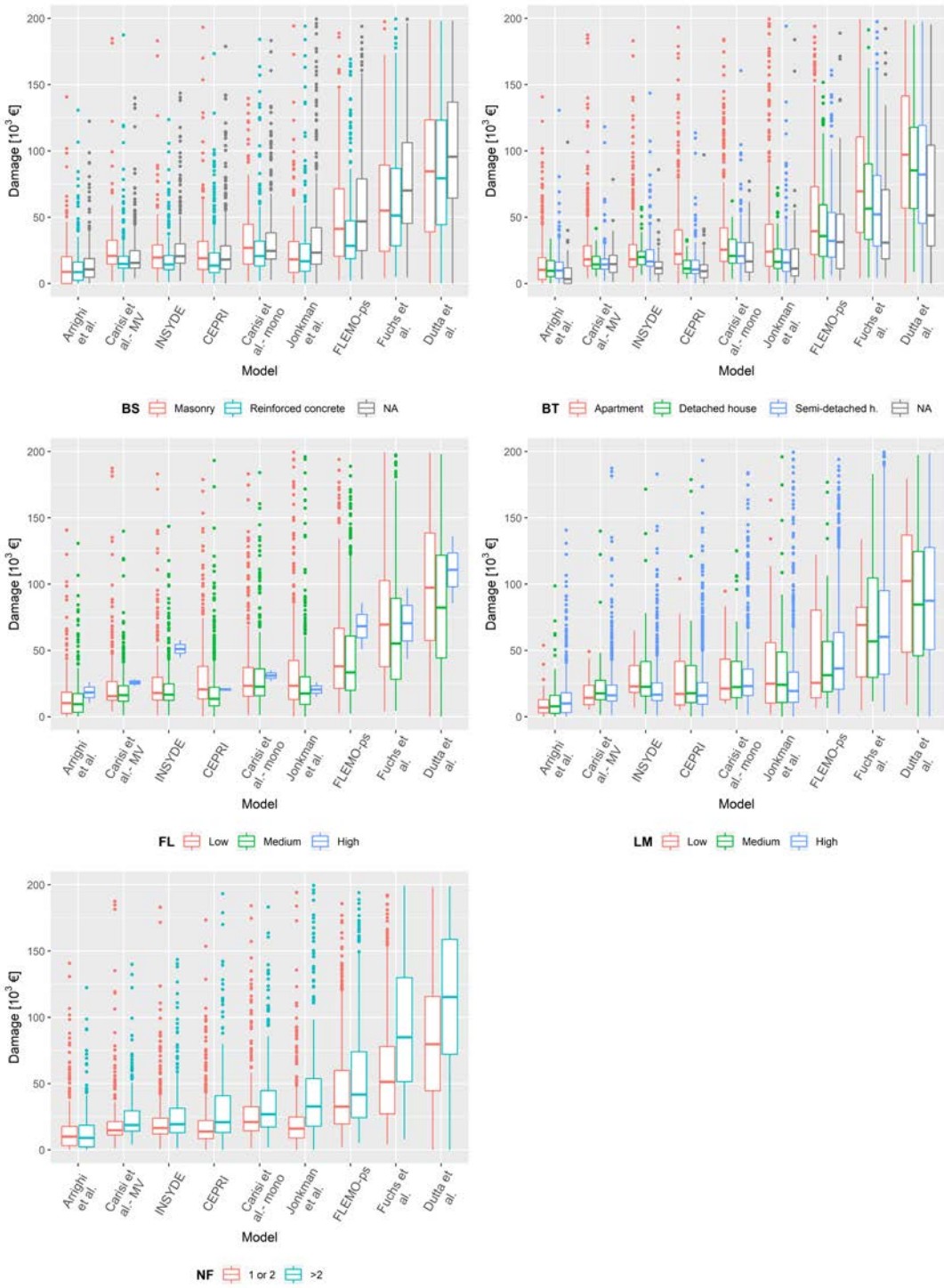

**Figure 7: Boxplots of damage estimates obtained with the tested models, for different classes of building structure – BS (Top-left), building type – BT (Top-right), finishing level – FL (Middle-left), level of maintenance – LM (Middle-right) and number of floors – NF (Bottom-left). Models are organised according to increasing value of total damage estimates.**

## 3.3 Comparison between estimates and observations

In order to gain knowledge on models' reliability in the investigated context, estimated losses were compared to observed damages derived from claims. For this purpose, a subset of the buildings within the simulated inundation area was considered, given that claims presented by private owners were available for only 345 buildings. Table 4 summarises the results of the sensitivity analysis by comparing the total observed damage to the total damage estimates obtained with the implementation of the nine models to the subset of buildings. The table confirms the results presented in Sect. 3.1 (i.e. models estimations differ by a factor of around 13) and highlights the systematic overestimation provided by the models with respect to observed damage, up to a maximum difference ratio of 13.97. Figure S1 in the Supplementary Material, displaying the ratios between estimated and observed damage at the building scale for different flood depth classes, suggests that detected differences do not depend on the hydraulic features in the inundated area but mainly on the damage models, for which the individual differences are similar across all flood depth classes. In this regards, Table 4 indicates the better performances of the Italian/local models (marked with the "IT" suffix in the table), with Arrighi et al. showing the lowest difference. However, by looking at its features, it is possible to state that even this last model tends to overestimate damage. First, because it does not consider clean-up costs (like INSYDE and CEPRI), which are instead included in the observations. Second, because the lower value of the total damage with respect to other models is partly due to the effect of the zero damage threshold for water depths lower than 0.25 m (see Sect. 3.1); indeed, as highlighted in Fig. 8 (showing the comparison between individual observed and estimated damages), a zero damage was expected by this model also for those buildings which experienced a significant loss.

**Table 4: Observed damage data versus estimates of the total monetary damage for the subset of buildings with claims (n.a.= not applicable). The second and the third columns report, respectively, the total damage and the unit value of damage per m². Mean value of estimates is reported in the last row. The fourth column reports the ratio between estimates and observed damage. Suffixes are used to track the original country of the models (IT=Italy, FR=France, JP=Japan, DE=Germany, AT= Austria, NL= The Netherlands).**

| Model | Monetary damage (M€) | (Unitary monetary damage [€m⁻²]) | Calculated damage/observed damage [-] |
|---|---|---|---|
| *observed* | *6* | *60* | - |
| Arrighi et al. (IT) | 6 | 43 | 1.00 |
| Carisi et al. - MV (IT) | 8 | 85 | 1.4 |
| Carisi et al. – mono (IT) | 12 | 132 | 2.19 |
| CEPRI (FR) | 10 | 74 | 1.72 |
| Dutta et al. (JP) | 77 | 265 | 13.97 |
| FLEMO-ps (DE) | 30 | 320 | 5.30 |
| Fuchs et al. (AT) | 50 | 171 | 9.03 |
| INSYDE (IT) | 9 | 85 | 1.69 |
| Jonkman et al. (NL) | 14 | 49 | 2.61 |
| Mean | 24 | n.a | 4.06 |

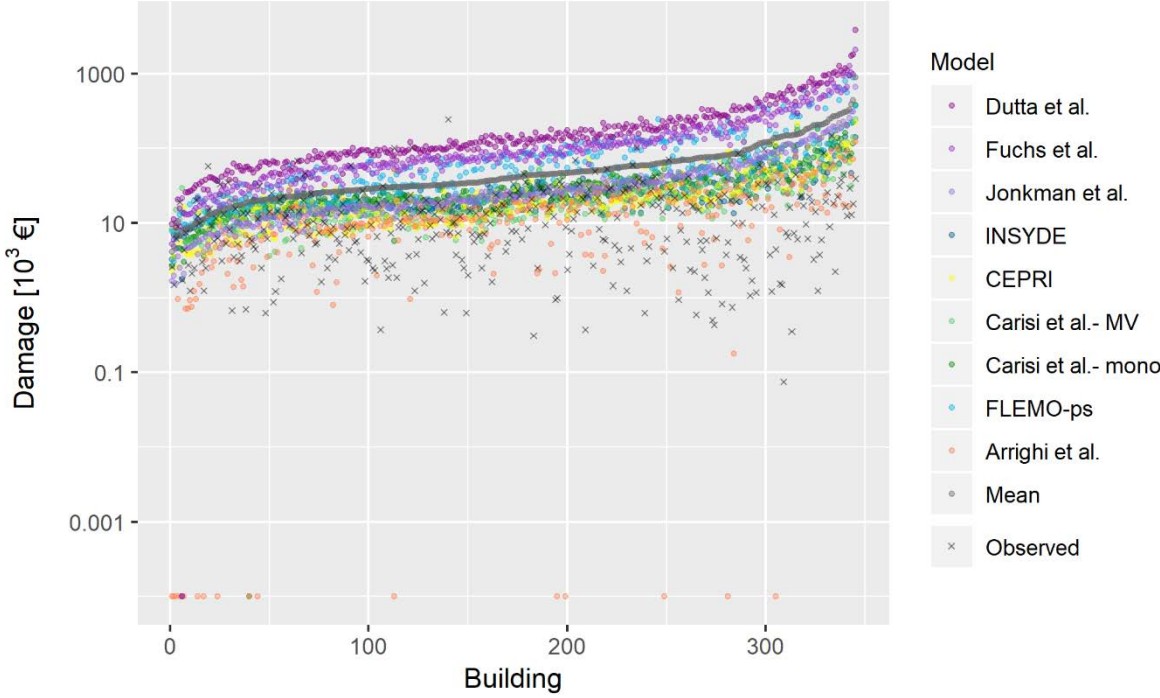

**Figure 8: Observed damage versus individual estimates of the monetary damage for the subset of buildings with claims. Data are ordered according to increasing value of mean estimate (in black).**

Interestingly, Table 5 finally shows that some of the imported models perform similarly or better than Italian models, with specifically high performance of CEPRI.

Figure 8 generally corroborates findings of Sect. 3.1, depicting a common trend in the models with largely different individual damage estimates. Moreover, it also emphasises the overestimation made by the models with respect to observations, with the latter not showing the common trend followed by the models. This evidence is supported by the results of the correlation analysis (Table 5), which reveals only marginal correlation between calculated losses and reported claims. On the contrary, the high correlation among models (see Fig. 5) raises the question of whether reported claims and damage estimation are comparable.

**Table 5: Pearson's correlation coefficient of observed damage and estimates supplied by the models with reference to the subset of buildings with claims. The acronyms in parentheses indicate the original countries of the models (IT=Italy, FR=France, JP=Japan,**
**DE=Germany, AT= Austria, NL= The Netherlands).**

|  | Observed |
|---|---|
| **Arrighi et al. (IT)** | 0.26 |
| **Carisi et al. - MV (IT)** | 0.10 |
| **Carisi et al. – mono (IT)** | 0.12 |
| **CEPRI (FR)** | 0.15 |
| **Dutta et al. (JP)** | 0.13 |
| **FLEMO-ps (DE)** | 0.13 |
| **Fuchs et al. (AT)** | 0.15 |
| **INSYDE (IT)** | 0.18 |
| **Jonkman et al. (NL)** | 0.13 |

### 3.4 Analysis of damage claims

In order to explain the differences between model results and observations, a thorough analysis of claims data was carried out. Given the general overestimation provided by the models, first we focused our attention on 44 buildings that are characterised
by very low values of observed damage (less than 1500 € in 2002 currency), referred to as "outliers" hereinafter. Table 6 reports the mean value of water depth, footprint area and external perimeter (i.e. the variables which most influence damage according to the analysis performed in Sect. 3.2) calculated for this subset of buildings and for all the buildings with claims. Table 6 indicates that low damages cannot be explained by significant differences in these influencing variables, given that both datasets show comparable values. Moreover, based on informal conversation with representatives of the Committee of
Flooded Citizens in Lodi, it is possible to postulate that existing outliers cannot even be explained by the adoption of individual mitigation actions (like temporary flood barriers or pumps), because no official flood warning was issued and, consequently, no lead time was available to undertake precautionary measures. Finally, from the analysis of building pictures available in Google Street View, we can state that outliers are not due to the presence of steps or other elements which increase the height of the building with respect to the ground level, reducing its vulnerability.

**Table 6: Mean value of water depth (h), footprint area (FA) and external perimeters (EP) for all buildings with claims and for the outliers' subset.**

| Dataset | Mean value of influence variables | | |
|---|---|---|---|
| | H [m] | FA [m$^2$] | EP [m] |
| outliers | 0.79 | 264.80 | 78.07 |
| all claims | 0.86 | 265.56 | 77.32 |

On the contrary, examining in detail the outlier claims, the following evidence arose:

- 27 % of outliers refer to claims with no detailed information about the type of damage, hindering the thorough understanding of low loss values in these cases;

- 32 % of outliers can be explained by the fact that declared damage regards only garages or boilers, while damage models typically assume a residential use of the building, with the presence/damage of all technical systems (i.e. heating, electrical, and water);

- 41 % of outliers refer to paltry claims, even in case of significant water depths (around 1 m), which are mostly related to painting of walls and replacement of doors and windows.

In view of the large proportion of paltry claims, it was attempted to understand the causes of declared damages. For this, we calculated the frequency of damage occurrence to different building components (i.e. damage to walls, damage to floor, damage to doors and windows and damage to systems) in the different claims and for three water depth classes (Fig. 9). Findings reveal an unexpected behaviour with respect to existing knowledge on damage mechanisms; in particular:

- damage to floors is found to be declared mostly for water depths higher than 1.5 m, although in principle this type of damage should be poorly related to water depth;

- frequency of damage to doors and windows decreases moving from the middle to the highest water depth class, as opposed to expectations (because of the occurrence of damage to windows with higher water depths);

- no damage to water, sanitary and heating systems is found to be declared for water depths higher than 1.5 m, contrarily to what can be expected by considering the typical height of the technical installations in Italian houses (Dottori et al., 2016).

According to our interpretation, inconsistency between expected and declared damage can be attributed to the fact that what is declared by citizens does not correspond to the actual budget required to replace or reconstruct the whole physical damage suffered by the building, but rather to the amount of money needed to bring the building back to a desired level of functionality, according to the financial resources of the owner: for this reason, for example, not all flooded doors are replaced and flooded floors always rebuilt. This would explain why synthetic models overestimate observed damage, as they are usually based on

full replacement/reconstruction costs. Likewise, it would explain why the model by Arrighi et al. performs better than others: indeed, the recovery value adopted by this model is defined as the average difference between the market value of new buildings and that of equivalent, older buildings requiring renovation. It is then sensible that this value reflects a balance between the two opposite extreme behaviours of buyers (which, in turn, depend on their financial resources): i.e. to completely renovate the building or to bring it back to a minimum level of functioning. In our view, such behaviours can be compared with those of flooded owners.

Moreover, declared monetary damage is strongly correlated to the expectations that citizens have to be reimbursed. This expectation is low in Italy, when in most cases limited funding is available for the compensation of private damage, which implies strict criteria and thresholds for compensation (often much lower than the effective damage). In addition, all costs must be proved by the citizens by means of official invoices. For all these reasons, citizens often prefer taking advantage of the "black market" rather than declaring damage (Cellerino, 2004). This would also explain why empirical models (derived from claims) developed in regions with high expectations and then high values of declared damage (like Germany), overestimate the observed damage in this case study.

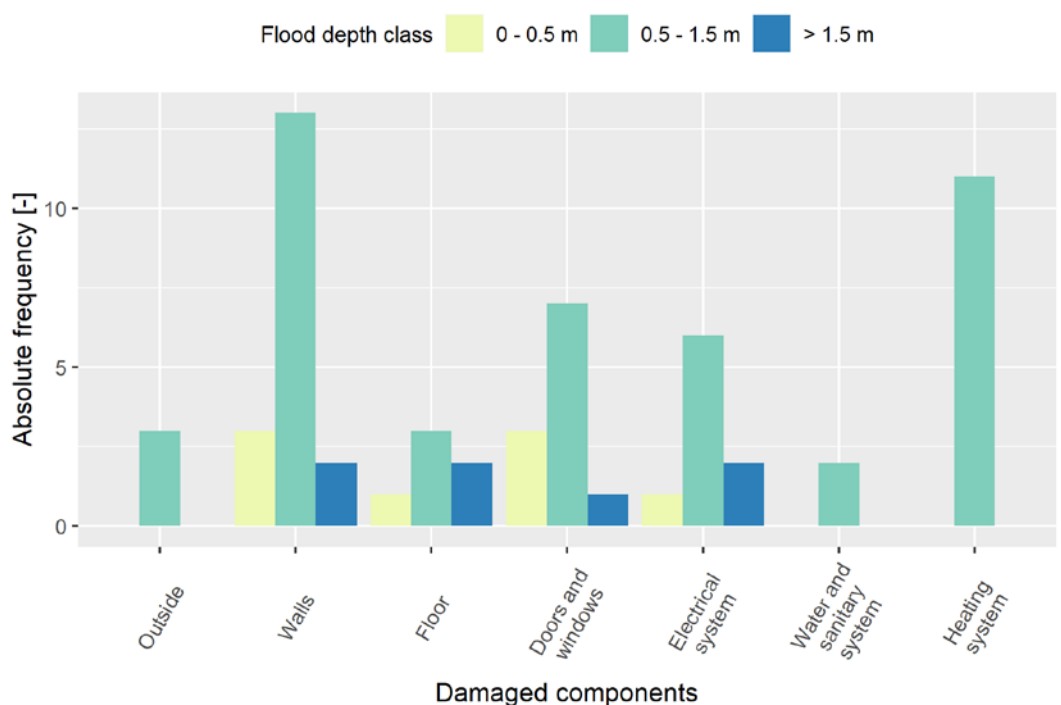

**Figure 9: Absolute frequency of declared damage to the different building components in the outlier dataset for different water depth (h) classes.**

From another perspective, in order to explain the scatter that is generally observed in real damage data with respect to water

depth (note that the value of Pearson's correlation coefficient between observed damage and water depth is 0.11), we focused the attention on 13 paired buildings, whereby the term "paired" refers to buildings with the same vulnerability characteristics (i.e. building type, building structure, level of maintenance and finishing level) as well as similar values of hazard parameters (i.e. water depth and flow velocity), but significant difference in the declared unit damage ($€m^{-2}$).

The analysis revealed that:

- considerable differences are attributable to declared or undeclared replacement costs of systems, rather than of doors and windows; this can be explained again by what is considered as monetary damage by citizens.

- in other cases, costs related to similar damage (e.g. cost of painting, cost of replacement of doors) differ a lot, even by a factor of 10. This discrepancy might be explained by wrong assumptions concerning the finishing level and/or the building type. More specifically, the actual conditions of buildings with high damage values could have been

better than what was assumed for the blind test, using cadastral data as reference (see Table 1).

- sometimes the above two factors add up, further increasing the differences among paired buildings in terms of declared damage.

Scatter in claims data can then be partially explained by the influence of local parameters (like the finishing level or the building type) which are difficult to assess at the micro-scale without a detailed field survey; nonetheless, it seems that the influence of

such parameters on damage estimation for the analysed models is very low (see Sect. 3.2) so that the latter are reliable only when applied at the meso-scale.

Overall, the analysis of claims highlighted that observed damage data need to be carefully analysed before being used for model validation, since their comparability with damage estimates is not always guaranteed.

**4 Discussion**

Results from the previous analyses were critically analysed in order to gain general knowledge on the transferability of damage models and reliability of damage estimates, and, in particular, to answer to the two specific research questions set in the Introduction.

Concerning the performance of local versus imported models, the blind test corroborated literature results (Cammerer et al., 2013), suggesting that model transferability depends on the consistency between the context of implementation and the original

calibration context, as far as both hazard and exposure/vulnerability features of exposed buildings are concerned. In fact, in the blind test, models developed for the Italian territory and for riverine floods performed generally better than models derived in other countries or for different flooding features, e.g. mountain areas. Such a result was not surprising as models providing good results have proven to perform well also in other Italian validation case studies, e.g., Arrighi et al. worked well also for the 2010 flood in Veneto Region (Scorzini and Frank 2017); the same applies to INSYDE and the two models by Carisi et al.,

which were tested in other Italian flood events (Amadio et al. 2019). On the contrary, the imported model of Dutta et al. was already found to not properly work in Italian cases (Scorzini and Frank 2017). Still, the analysis of damage claims revealed

that, as far as empirical models are considered, transferability could depend also on comparability of the compensation contexts, given that observed losses on which empirical models are calibrated may depend on citizens' expectations of reimbursement.

Regarding instead the second question, literature suggests that the inclusion of several influencing variables should increase the accuracy of a model (Merz et al., 2013; Schröter et al., 2014; Van Ootegem et al., 2018). Still, the blind test highlighted that such an evidence can be invalidated by the lack of availability/consistency of input data between the calibration and the implementation context. Indeed, the models considered in the blind test were designed to be used with the type of data usually available in the original context, which generally differ from the data available in the Lodi case study, i.e., models use different

proxy variables for the same explicative parameters. For this reason, assumptions had to be undertaken to allow the application of a model in the case study area (see Sect. 2). For example, the building categories (BT) assumed by CEPRI ("apartment"/ "single storey building"/ "multi-storeys buildings") are different than the Italian ones ("apartment"/ "detached"/ "semi-detached") so that a correspondence has to be defined, also on the basis of the number of floors (NF); specifically apartment is defined by BT ="apartment", "single-storey" is defined by BT = "detached" or "semi-detached" and NF = 1, "multiple-

storeys" is defined by BT ="detached" or "semi-detached" and NF > 1. Correspondence among building categories was defined also for the implementation of FLEMO-ps, although in this case the task was quite straightforward, since the German building categories are almost coincident with the Italian ones (FLEMO-ps distinguishes between "Multi-family house" / "Semi-detached house" / "One-family home"). Assumptions on input variables may reduce the reliability of the original model because of an improper/inaccurate "adaptation" of the available data, thus reducing the advantage of using many variables.

This also explains why the simple models by Jonkman et al. and Carisi et al. - mono provided comparable or better results than those obtained from multi-variable models like FLEMO-ps or CEPRI. Also, the use of additional variables may have different impact depending if, in the application area and differently for the original model development strategy, this information is retrieved at the building scale or known as aggregated variable. Consultations of experts with local knowledge were needed to help in the correct interpretation and use of the available input data for the Lodi case study. Importantly, the blind test

highlighted that none of the tested models (being them local or imported, simple or multi-variable) seemed appropriate to estimate flood damage at the building scale in the given context; still, models' performance improved when aggregated damage data were taken into account. In fact, considering the 345 buildings for which a claim was known, all models' estimates differed significantly individually (Fig. 8), but some of them indicated a total damage figure close to the observations (Table 4). Besides the already discussed potential biases of claim data, this duality suggests that model uncertainty may be balanced in aggregated

results, i.e. the lump-sum might be more reliable than the individual results. This raises the question of which is the right spatial scale (that is the level of complexity) of analysis to get reliable results, and for which objective. For example, by implementing the simpler, lump-sum model DELENAH_M (Natho and Thieken, 2018), an adaptation of the UNISDR method for national damage estimates (UNISDR 2015) in developed countries taking Germany as a study case, the estimate of the aggregated damage for the 345 buildings with claim data is 4.3 M€ This estimation is affected by an error which is comparable or lower

than errors supplied by the micro-scale models (see Table 7), although being obtained with a simple calculation and in a blind

mode, i.e. using the average damage ratio for severe floods and the average housing size derived from German survey data (Thieken et al., 2017) on flood losses in the housing sector (note that in this case underestimation of total damage is due to the adoption of a conservative housing size, so that the estimation must be intended as a minimum estimate or a lower bound). Is this assessment useful for flood risk mitigation? Which is then the advantage of using micro-scale models? Is there a level of spatial aggregation which supply reliable, more informative estimation than a simple lump-sum at the municipality level? Answers to these questions will be objective of further investigations by the research groups involved in the test.

## 5 Conclusions: lessons learnt from a modeller's perspective

The blind test conducted in this study represented an opportunity not only to deeply investigate the transferability of tested models and the reliability of their estimations, especially regarding the potentialities of local and multi-variable models, but also to increase authors' awareness on strengths and limits of flood damage modelling tools. As concluding remarks, we report in the following section take-home messages synthesising lessons learnt from the blind test, from a modeller's perspective.

First, a former source of variability among models' outcomes lies in the approach for exposure assessment, which then represents a critical, often overlooked, step in flood damage modelling. In particular, assessing exposure coherently with the approach originally adopted in model development is key to preserve the original reliability; in this regard, the blind test showed that the different approaches applied within the models demand for a clear definition and differentiation of the terms "exposure value" and "building value". Nonetheless, the blind test indicated a common overestimation, confirmed also in other case studies (Zischg et al., 2018; Cammerer et al., 2013; Thieken et al., 2008; Fuchs et al., 2019b; Arrighi et al., 2018a, 2018b), in terms of number of buildings damaged by a flood event (i.e. the number of buildings with claims is significantly lower than those exposed to the flood). This might be attributed to the fact that not all affected building owners asked for compensation, or that some buildings are not affected by the flood due to local micro-topographical conditions or due to the installation of protection measures. However, it might also highlight problems in the current strategy adopted to identify exposure (e.g. by not considering building elevation).

A second critical issue in flood damage modelling is the transfer of models in space and time, with difficulties on predicting the expected performance of a given model when applied in a different context (e.g. Jongman et al. 2012, Cammerer et al., 2013; Wagenaar et al., 2018). Accordingly, flood damage modellers should always be cautious when applying a flood damage model to a new context. Their general trust towards the model performance in the new study area must be in the first instance limited; however, model validation (ideally, over multiple datasets) can significantly increase the trust level.

But validation of damage models invariably relies on observed damage data, either from insurance claims, governmental reimbursement claims or direct surveys, all of which are generally intended as "reality". Indeed, it is often the case that empirical data are used in validation analyses without any possible preliminary evaluation on their quality and significance, simply because no ancillary information is available, as for instance for insurance data (André et al. 2013; Spekkers et al. 2013; Zhou et al. 2013; Wing et al. 2020). In this context, the blind test highlighted that "reality" depicted by observations is not

univocal, so that observed data must be carefully investigated before their comparison with model outcomes, as they may be addressing different types of damage, damage to different components, or being incomplete. Based on this consideration, flood

damage modellers must be always cautious when drawing conclusions from validation analyses: if a model does not fit well to some empirical data, this does not necessarily reflect the inability of the model in general terms but attention has to be drawn to input data quality and vice versa. This also points out the importance of collecting not only flood damage data, but also ancillary information on flood hazard and vulnerability of affected assets in the ex-post flood phase (Merz et al., 2004; Thieken et al., 2005; Ballio et al., 2015; Thieken et al., 2016; Molinari et al., 2017a; Molinari et al., 2019). Moreover, consultations of

experts with local knowledge can help in the correct interpretation and use of observed damage data.

In absence of data (or appropriate data) for validation, the application of several models might be useful to quantify mean and variance and provide a range of uncertainty of the estimations (Figueiredo et al., 2018); a good agreement of model results, in particular with the models developed for context similar to the one under investigation, can significantly increase the trust level in model performance. In this regard, the blind test stressed that damage models have to be compared in their original

form, meaning that, for instance, relative damage models relying on the total building value cannot be directly compared to the ones relying on only the first floor.

As a general recommendation, to select a damage model for an application in a different country, it is important to verify the comparability between the original and the investigated physical (in terms of hazard and building features) and compensation context, as well as the availability and coherence of the input data. Moreover, when transferring a model (in space or time),

proxies of input variables are frequently needed, and the modeller must be prudent in this step. A good understanding of both the data used during the model development and the data gathered for the new application is crucial, as the attribution of uncertainty becomes elusive afterwards, if this step is neglected. The blind test highlighted that the real effort of transferring the models to the given implementation context was related to finding the "right" required data, while the costs of implementing assumptions about exposure and calculating the damage value were negligible. To support transferability, there is then a need

to precisely describe how the models were developed, which variables were included and for which specific context. In this regard, a protocol or standardised information for all models would help in finding the most appropriate tool in a given context; in fact, at present, details about origin, calibration, assumptions, field of application, etc. of existing models in the literature are few and sparse. A new promising attempt in this direction is represented by the Flood Damage Model Repository, recently launched by Politecnico di Milano (www.fdm.polimi.it) as a research community effort.

Given these considerations, and in contrast with the general approach in which each research group develops its own model for a limited context, authors support a call for a community effort in setting up a common model, with different sub-modules useable for many purposes and regions, and with a flexibility in the required input data.

**Author contributions.** *Conceptualisation of the blind test:* Francesco Ballio; *Management of the blind test:* Daniela Molinari; *Data and results management:* Daniela Molinari, Alice Gallazzi, Marta Galliani; *Models implementation*: Chiara Arrighi,

Francesca Carisi, Marta Galliani, Patric Kellermann, Markus Mosimann, Stephanie Natho, Claire Richert; *Elaboration of results:* Daniela Molinari, Anna Rita Scorzini, Alice Gallazzi; *Interpretation of Results – Original Draft:* Daniela Molinari, Anna Rita Scorzini, Francesco Ballio; *Interpretation of Results – Review:* all; *Writing-Original Draft:* Daniela Molinari, Markus Mosimann, Francesca Carisi, Alessio Domeneghetti, Guilherme S. Mohor; *Writing-Review:* Daniela Molinari, all; *Figures and Tables:* Anna Rita Scorzini, Daniela Molinari.

**Acknowledgements.** Authors acknowledge with gratitude Andrea Nardini (from the Italian Centre for River Restoration – CIRF) and Marianne Skov (from Rambøll, Denmark) for their fruitful suggestions and hints during the developing of the test. Authors are also grateful to three anonymous Reviewers for their meaningful comments and suggestions.

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

# Are flood damage models converging to "reality"? Lessons learnt from a blind test

Daniela Molinari[1], Anna Rita Scorzini[2], Chiara Arrighi[3], Francesca Carisi[4], Fabio Castelli[3], Alessio Domeneghetti[4], Alice Gallazzi[1], Marta Galliani[1], Frédéric Grelot[5], Patric Kellermann[6], Heidi Kreibich[6], Guilherme S. Mohor[7], Markus Mosimann[8], Stephanie Natho[7], Claire Richert[5], Kai Schroeter[6], Annegret H. Thieken[7], Andreas Paul Zischg[8] and Francesco Ballio[1]

[1] Department of Civil and Environmental Engineering, Politecnico di Milano, Piazza Leonardo da Vinci 32, 20133, Milano, Italy

[2] Department of Civil, Environmental and Architectural Engineering, University of L'Aquila, Via Gronchi 18, 67100, L'Aquila, Italy

[3] Department of Civil and Environmental Engineering, University of Florence, Piazza San Marco 4, 50121, Firenze, Italy

[4] Department of Civil, Chemical, Environmental and Material Engineering, University of Bologna, Viale Risorgimento, 2 - 40136, Bologna, Italy

[5] G-EAU, Univ Montpellier, AgroParisTech, CIRAD, IRD, INRAE, Montpellier SupAgro, Montpellier, France

[6] GFZ German Research Centre for Geosciences, Section Hydrology, Telegrafenberg, 14473, Potsdam, Germany

**[7]** Institute of Environmental Science and Geography, University of Potsdam, Karl-Liebknecht-Strasse 24-25, 14476, Potsdam, Germany

[8] Institute of Geography, Mobiliar Lab for Natural Risks, Oeschger Centre for Climate Change Research, University of Bern, Hallerstrasse 12, 3012, Bern, Switzerland

*Correspondence to*: Daniela Molinari (daniela.molinari@polimi.it)

**Abstract.** Effective flood risk management requires a realistic estimation of flood losses. However, available flood damage estimates are still characterised by significant levels of uncertainty, questioning the capacity of flood damage models to depict real damages. With a joint effort of eight international research groups, the objective of this study was to compare, in a blind validation test, the performances of different models for the assessment of the direct flood damage to the residential sector at the building level (i.e. micro scale). The test consisted in a common flood case study characterised by high availability of hazard and building data, but with undisclosed information on observed losses in the implementation stage of the models. The selected nine models were chosen in order to guarantee a good mastery of the models by the research teams, variety of the modelling approaches and heterogeneity of the original calibration context, in relation to both hazard and vulnerability features. By avoiding possible biases in model implementation, this blind comparison provided more objective insights on the transferability of the models and on the reliability of their estimations, especially regarding the potentials of local and multi-variable models. From another perspective, the exercise allowed to increase awareness on strengths and limits of flood damage modelling, which are summarised in the paper in the form of take-home messages from a modeller's perspective.

## 1 Introduction

Efficient and effective flood risk management requires a realistic estimation of flood losses, implying the use of reliable models

35 for flood hazard, damage and risk assessment (Meyer et al., 2013; Gerl et al., 2016; Zischg et al., 2018; Wagenaar et al., 2018; Molinari et al., 2019). Although several hydraulic models are available (Teng et al., 2017), their variety seems to be overtopped by the variety of flood damage models as, according to Gerl et al. (2016), only in Europe, 28 models (including 652 functions) exist to assess flood losses, whereas almost half of them focus on residential buildings.

Even within the residential sector and with respect to direct damage (i.e. damage due to the direct contact with the flooding

40 water), the diversity of approaches is manifold. First, the models are classified according to the intended spatial scale of the analysis: while micro-scale models refer to the individual exposed building, meso-scale models work at more aggregated scales, like land use or administrative units, with large-scale spatial units (like regions or countries) being at the base of macro-scale models (Merz et al., 2010).

A second difference lies in the approach adopted for model development, with empirical models using damage data collected

45 after flood events (e.g. Merz et al., 2004) and synthetic approaches implementing information collected via what-if-questions (e.g. Penning-Rowsell et al., 2005). Still, both categories are characterised by a variety of methods; for example, empirical data can be interpreted by means of different statistical and mathematical tools, ranging from simple regression (e.g. Merz et al., 2004) to more sophisticated machine learning algorithms and data mining approaches (e.g. Merz et al., 2013; Amadio et al., 2019). A distinction can also be made between absolute and relative damage models: the first directly return a value in a

50 specific currency (Dottori et al., 2016; Rouchon et al., 2018), while relative damage models estimate the physical vulnerability or the degree of loss of an exposed asset (Fuchs et al., 2019a), to be multiplied by its monetary value to assess the damage. Linked to this point is the question of what is defined as exposure in the models: besides the distinction whether a model relies on the value of the whole building or just of the affected floors, it is also important to know if, for instance, the basement is considered as well. Moreover, exposure assessment may differ regarding the monetary value, whether it is based on e.g. market

55 or replacement values (Röthlisberger et al., 2018), rather than full replacement costs or depreciated values (Merz et al., 2010). A final important difference among the models lies in the number and type of considered input parameters, i.e. on model complexity. Simplest damage models (referred to, in the following, as "low-variable models") take into account a few number of variables, mostly the water depth at building location as well as building area and its monetary value (only in case of relative models). Even in their simplicity, these models can significantly differ from each other, due to the distinct shapes of the

60 underlying damage functions, e.g. square root function (Dutta et al., 2003; Carisi et al., 2018), beta distribution function (Fuchs et al., 2019b) or graduated function (Jonkman et al., 2008; Arrighi et al., 2018a). On the contrary, multi-variable models consider numerous hazard and exposure/vulnerability input factors and, consequently, are supposed to be more accurate when detailed data is available (Thieken et al., 2008; Schröter et al., 2014; Wagenaar et al., 2017; Amadio et al., 2019). Nevertheless, simple models tend to be the most widely used, due to their ease for implementation and low requirements for input data.

65 Hence, flood damage modellers have always to envisage the trade-off in the model choice, i.e., using a complex, probably more accurate model with specific data requirements, or a simple, probably less accurate one that can be applied without extensively available data. However, it has been shown that even a small ensemble of models outperforms individual models, with the additional advantage of providing uncertainty information (Figueiredo et al., 2018).

What most models have in common is that they are calibrated in specific contexts, usually representative of a certain spatially limited region. In many cases, instead, validation of flood damage models is lacking (Merz et al., 2010; Gerl et al., 2016; Molinari et al., 2019). Where it is not lacking, the data used for model validation are often either a subset of the dataset used for calibration or are collected in the same region or country of model development. This implies that, even if a model has been locally validated, it is not necessarily correct to apply it to any other region, unless this latter reflects the context for which the model was derived. For instance, the application of a damage model that has been developed for alpine areas (i.e. house building tradition of the European Alps and flood processes involving significant sediment transport) to a coastal country like the Netherlands, and vice versa, is prone to lead to large discrepancies from reality (e.g. Cammerer et al., 2013). Hence, flood damage models need to be tested in regions other than those where they were calibrated in, to be confident with their transferability in space.

Nevertheless, what all models and modellers deal with is the lack of data for model calibration and validation (Merz et al., 2010; Jongman et al., 2012; Meyer et al., 2013; Molinari et al., 2019). The overall economic impact of a flood is hardly reproduced by ex-post data and then biases have also to be taken into account when transferring models to different regions, e.g. due to different insurance conditions, uncompleted claims, etc.; moreover, even years after flood events, monetary losses can be revised due to long-term recovery: as an example, monetary losses of the 2013 flood in Germany were estimated at 6.7 M€ in 2013 (Deutscher Bundestag, 2013) and changed over the following years to 8.2 M€ (Bundesministerium für Verkehr und digitale Infrastruktur, 2016). For this reason, comparative studies over a broad range of test cases (i.e. different validation datasets) are essential for acquiring a thorough understanding of the performances of the modelling tools that could help in enhancing the confidence in their reliability.

The aim of this study is to contribute to the understanding of models' transferability and reliability by testing and comparing different damage models in a blind validation test. This joint effort of eight international research groups consists in a common flood case study characterised by high availability of hazard and building data, but with undisclosed information on observed losses in the implementation stage of the models. Tested models have been chosen among those mastered by the authors; indeed, the authors were either developers of the models or experienced users with significant knowledge of them, in order to prevent any possible bias in the results that could arise from an incorrect application of the models (for example, a non-expert user may misunderstand the meaning of some input variables, which would affect the final estimation).

Even though comparative analyses on the performance of damage models have become more frequent in the literature (Jongman et al., 2012; Cammerer et al., 2013; Scorzini and Frank, 2017; Carisi et al., 2018; Figueiredo et al., 2018; Amadio et al., 2019), according to authors' knowledge, this study would represent the first flood damage model comparison performed in a blind-mode. This type of comparison can provide more objective insights for a better understanding of models' capabilities and then for reducing modelling uncertainties, as already demonstrated in similar tests performed for other disciplines like seismology, hydrology and computational fluid dynamics (Smith et al., 2004; Soares-Frazao et al., 2012; Krogstad and Eriksen, 2013; Zelt et al., 2013; Andreani et al., 2019; Ransley et al., 2019; Skorek et al., 2019). Indeed, possible biases are avoided as participants cannot be influenced by validation data, being them undisclosed in the implementation phase of the models, e.g.

by trying to adjust or tune their models, especially regarding the more qualitative input parameters, in light of observed damages.

This study focuses on micro-scale (i.e. individual item scale) direct damage assessment to residential buildings, in line with the larger availability of damage modelling approaches developed in Europe for this specific sector and scale.

As the research groups use approaches representing many different types and characteristics of models (low-variable – multi-variable; absolute – relative; graduated – regression – machine learning – synthetic), being calibrated on the basis of observed data stemming from different countries (Austria, France, Germany, Italy, Japan, Netherlands), with different landscapes and

level of complexity in exposure/vulnerability, the blind test as performed in this study can provide an in-depth understanding of the links between models features, their transferability and the reliability of the estimated damages.

In particular, the blind test allowed to investigate these specific questions, raised from the evidence supplied by the literature (Thieken et al., 2008; Cammerer et al., 2013; Schröter et al., 2014; Dottori et al., 2016; Wagenaar et al., 2017; Amadio et al., 2019): do local models (i.e. models calibrated with data from a context similar to the investigated one) outperform other

models? Do multi-variable models perform better than simplest ones and if so, why?

The paper is organised as follows. The methodology, models and case study implemented in the blind test are first presented in Sect. 2. Section 3 discusses results of the test, first by considering damage estimates obtained in a blind implementation of the models, and then by comparing damage estimates with documented losses. Answers to the specific research questions are provided in Sect. 4. Finally, in Sect. 5, evidence from the blind test is synthesised in lessons learnt (on flood damage modelling)

from a modeller's perspective, including the identification of research needs for further improvements of flood damage models.

## 2 The blind test: case study, methodology, models

The main idea behind the blind test was to evaluate the performance of different flood damage models by their implementation to a common case study, to obtain enhanced information on their transferability, validity and reliability; the test is defined "blind" as, in order to avoid bias in the estimation process, the value of the observed damage was unknown to modellers in the

implementation stage of the models. In particular, damage data were unblinded only to one group, which was the promoter of the initiative and responsible for data and results management. All required input data to reproduce the damage scenario for the examined event were made available to the participants, who were then asked to submit their results to the exercise manager in an established time frame. Once all contributions from the different groups had been gathered, observed data were disclosed, and models' performances were compared and analysed in a shared discussion between the participants.

**2.1 Case study**

The investigated context is the town of Lodi, North of Italy (Fig. 1), which was hit by a severe flood on 25-26 November 2002, caused by the overflow of the Adda River as a result of two weeks of heavy rainfalls over North-western Italy.

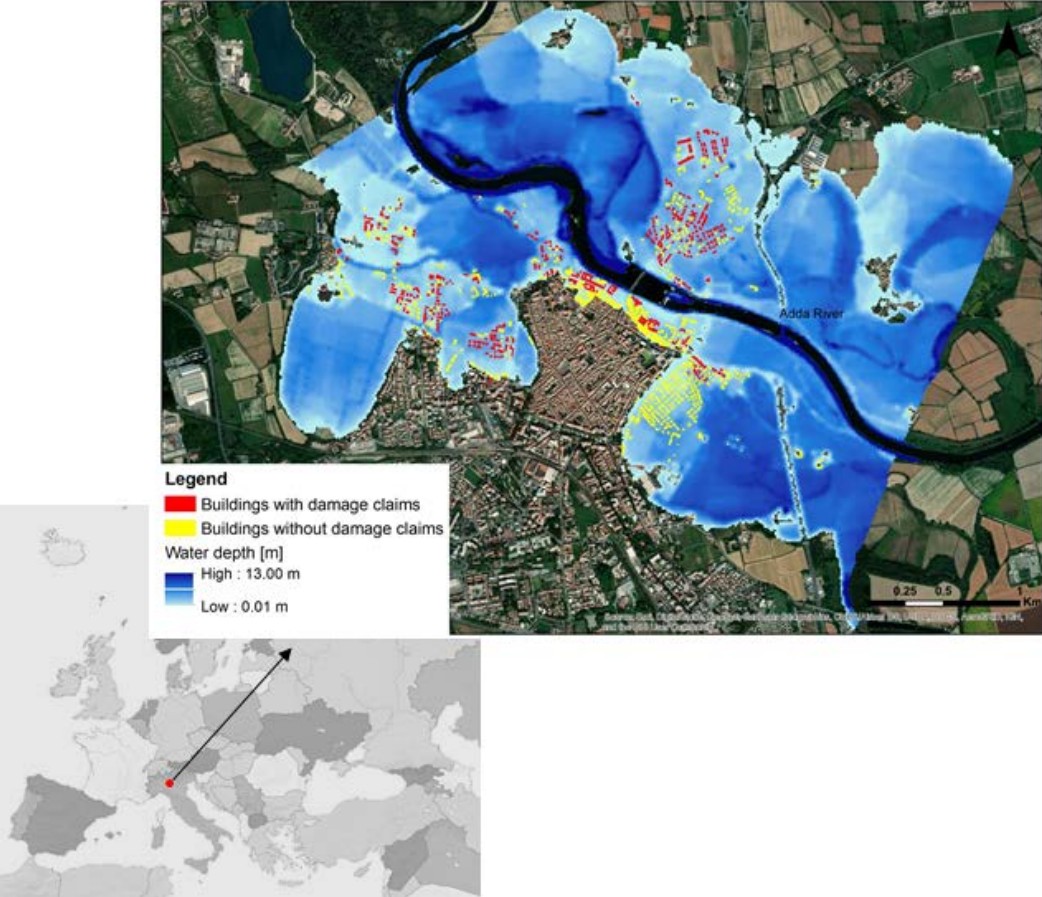

**Figure 1: Map of the flooded area and affected buildings.**


The flood caused severe damage to residential buildings, commercial activities and public services in the area, including the main hospital. Fortunately, no fatalities occurred. The event was chosen as reference for the exercise as it is well documented and characterised by a high availability of hazard, exposure and vulnerability data. In detail, with respect to the hazard, information on observed water depths was available for more than 260 points within the inundated area, deriving from

indications provided by municipal technicians and by citizens in damage compensation requests, as well as from interpretation of photographs taken during or immediately after the flood. These data were used for the validation of the 2D hydraulic simulation of the event: the resulting average absolute differences between observed and calculated water depths within the inundated area ranged from 0.2 to 0.4 m, depending on the validation zone in which observed water depth data were aggregated (Scorzini et al., 2018). This is surely a possible source of uncertainty; however, reported differences could be considered to

provide relatively small impacts on the damage estimation. Moreover, given that all tested damage model shared the same hazard data, this would be a common source of uncertainty that should not affect the overall results of the blind test.

Available micro-scale data on exposure and vulnerability of residential buildings are shown in Table 1. Altogether, observed

damage was known for 345 of the 877 buildings in the flooded area (after hydraulic simulation; Fig. 1), as derived from claims compiled by citizens after the flood to ask for public compensation.

Claims were mostly collected by the Municipality of Lodi and, in a small part, by the Regional Authority of the Lombardy region after the event. Available claims data, in their original papery form, were then firstly acquired and successively stored in a georeferenced digital database, by a team of researchers of Politecnico di Milano in summer 2017. As regards data from the Municipality, original claims were organised in forms, including information on the owner, the address of the flooded building, its typology (e.g. apartment, single house), the number of affected floors, a description of the physical damage and

its translation into monetary terms (distinguishing, for the different rooms of the building, among damage to walls, windows and doors, floor, systems and content). In few cases, information on water depth inside the building and on clean-up costs, non-usability of the building and intangible damage (e.g. loss of memorabilia) was also inferred from the qualitative damage description in the forms. The quality/reliability of data included in the claims was not uniform, since only some of the owners justified the costs for fixing damage by means of invoices. As regards data from the secondary source (i.e. the Regional

Authority), they included limited information on the owner, the address of the flooded building and the monetary value of damage, distinguished in damage to structure and contents.

**Table 1: available micro-scale data for the blind exercise.**

| Data | Variable | Description | Source | Year |
|---|---|---|---|---|
| *Area* [m²] | FA | Footprint area of the building | Regional topographical database | 2010 |
| *Perimeter* [m] | EP | External perimeter of the building | Regional topographical database | 2010 |
| *Basement* | BA | Presence of basement yes/no | Lodi cadastral data | 2016 |
| *Building type* | BT | Type of building (apartment, detached or semi-detached house) according to the cadastral data. | Lodi cadastral data | 2016 |
| *Finishing level* | FL | Quality of the building (low, medium or high) according to the cadastral data: | Lodi cadastral data | 2016 |
| *Building structure* | BS | Type of building structure (masonry or reinforced concrete) calculated as the most frequent value for the buildings in the census block it owns. | National Institute of Statistics (ISTAT) | 2001 |
| *Floors* | NF | Number of floors calculated as the most frequent value for the buildings in the census block it owns. | National Institute of Statistics (ISTAT) | 2001 |
| *Level of maintenance* | LM | State of conservation (low, medium or high) of the building calculated as the most frequent value for the buildings in the census block. | National Institute of Statistics (ISTAT) | 2001 |
| *Water_depth* [m] | h | Mean value of water depth in the building area. | 2D hydraulic modelling | 2018 |
| *Flow_velocity* [m s⁻¹] | v | Mean value of flow velocity in the building area. | 2D hydraulic modelling | 2018 |
| *Presence of pollutants* | q | Presence of fuel spillage or other pollutants | Claims forms / photos of the event | 2002 |
| *Replacement value* [€m⁻²] | RV | Reconstruction value of residential building given as a function of the building type and building structure of the building, based on existing literature and official studies | Cresme-Cineas-Ania | 2014* |
| *Market value* | MV | Market value of residential buildings, as a function of building | OMI (Osservatorio del | 2014* |

| | | type, finishing level and building location | Mercato Immobiliare) – Italian real estate and property price database | |
| [€m⁻²] | | | | |

* for the objective of the exercise data were discounted to 2002 values

## 2.2 Methodology

The methodological approach followed in the test included the following steps:

### Step 1: identification of damage models to be tested

The choice was based on several considerations: (i) good mastery of the models by the research team (i.e. damage models regularly used or initially developed by the groups), (ii) heterogeneity of the approaches, by considering simple and multi-variable models, empirical and synthetic approaches, absolute and relative models, and (ii) models being calibrated in a different context than the investigated one. The choice converged to the nine models described in Sect. 2.3.

### Step 2: implementation of the models to the case study in a blind mode

The models were implemented independently by ~~each~~ the research groups (i.e. each group applied one up to three models, according to its specific expertise) to calculate damage to all 877 buildings that were exposed to the 2002 Lodi flood, according to the inundation area simulated by the hydraulic model (Scorzini et al., 2018). All the groups used available and common data on hazard, exposure and vulnerability, as described in Table 1. While this step was simple for Italian models (which were originally developed to work with the same kind of data available for the case study), some efforts were required for the other models, particularly in the case of multi-variable ones. This is due to a lack of correspondence/consistency among exposure and vulnerability data available in the different countries, on which damage models are usually based. For instance, correspondence had to be defined among building types classified by the Italian cadastre and the ones adopted by the German and French models and the ones as classified by the Italian cadastre.

The damage assessment was carried out only for building structures, given that not all models are designed to simulate damage to household contents. At this step, observed losses were still blinded to the research groups in order to avoid possible bias in the estimation.

### Step 3: comparison of model outcomes

Exposure and damage estimates supplied by the different models were compared, at the aggregated and individual level, with the main objectives of (i) understanding the weight of exposure assessment on damage calculation, and (ii) pointing out common or divergent model outcomes.

### Step 4: comparison of model features

Models were compared in terms of trends and variance of individual damage estimates, for homogeneous classes of input

variables, by considering one variable at a time. The objective was to understand whether the inclusion of more explicative variables may be considered as a possible source of variation, as well as to identify the most influencing parameters on the final output of the models.

*Step 5: comparison between estimates and observations*

This phase aimed at investigating the performances of the different models in the analysed context. Calculated damages were compared to observed losses coming from claims. The comparison was possible only for 345 of the buildings included in the flooded area, for which official claims were available.

*Step 6: analysis of claims*

Claim data were analysed with the aim of identifying potential reasons for (in-) consistencies between estimates and observations.

*Step 7: synthesis of results*

Results obtained in the previous steps were critically analysed in order to gain knowledge on model transferability and reliability of damage estimates, with respect to their implementation in a same case study, and from a modeller's perspective. The analysis was conducted jointly by all groups, in the form of brainstorming, during several remote meetings and one face to face meeting.

**2.3 Models**

The main characteristics of the selected models are summarised in Table 2 and briefly described hereinafter.

- The model developed by **Arrighi et al.** (2018a, 2018b) is a relative synthetic model which expresses monetary damage as a function of water depth and recovery cost for buildings with and without basement. A zero-damage threshold is set for a water depth lower than 0.25 m for buildings without basement. The recovery cost is assumed equal to 15 % of the exposure, calculated as the market value of the flooded floor(s) based on the footprint area. The ratio between recovery cost and market value is based on the comparison between residential prices for new buildings and buildings requiring renovation (Italian real estate data). The model was created based on expert judgement for the city of Florence (Italy) and applied both at building and census block scale (Arrighi et al. 2018a, 2018b). It has been validated through comparison with other validated models (Arrighi et al., 2018b) and ex-post damage in another Italian context (Scorzini and Frank, 2017).

- **Carisi et al. - MV** (Carisi et al., 2018) is an empirical multi-variable model, which estimates relative building losses considering six explicative variables: maximum water depth, maximum flow velocity, flood duration, monetary building value per unit area (based on market value), structural typology and footprint area of each building (Carisi et al., 2018). Calibration data refer to the inundation event occurred in the province of Modena (Italy) in 2014, when

a breach in the right embankment of the Secchia river caused about 52 km$^2$ of flooded area and €500 million losses (see, e.g., Orlandini et al., 2015). Observed losses were derived from 1330 claim forms filled by citizens and collected by authorities for the purpose of compensation, while the maximum water depth was reconstructed by means of a fully 2D hydrodynamic model; economic building values per unit area were finally retrieved by the Italian Revenue Agency reports. The model does not consider damage to basements. The model uses the Random Forest approach (Breiman et al., 1984; Breiman, 2001), which is a tree-building algorithm for predicting variables, recursively repeating a subdivision of the given dataset into smaller parts in order to maximize the predictive accuracy. In order to avoid overfitting problems, several bootstrap replica of the learning data are used, for which regression trees are learned, then aggregating the responses from all trees to estimate the final result.

- **Carisi et al. - mono** (Carisi et al., 2018) is an empirical simple model, calibrated on the previously cited 2014 Secchia flood event. The model supplies the relative damage to building (using the market value to relativize the observed monetary damage when developing the model), as a function of the maximum water depth. The model does not consider basements or garages, for coherence with the calibration context, where most of the buildings do not have these elements.

- The model developed by **CEPRI** (European Center for Flood Risk Prevention, (CEPRI, 2014a)) is a synthetic (expert-based) and multi-variable model that expresses absolute damage as the expected sum of the actions that must be performed after a flood to restore to the pre-flood state, including clean-up costs. The flood parameters taken into account are water depth and submersion duration. The considered building characteristics are the building type (single storey house, double storey house, or apartment), the floor area, the presence of a basement and its area. For each type of building, one damage curve indicates the damage to structural components, and one the damage to the furniture. Two separate damage curves are used to estimate the damage to the basements contained in houses or apartment blocks. Initially, the model was developed to estimate damage due to all types of floods. Its estimates have been compared to empirical damage due to fast rise floods (CEPRI, 2014a; Richert and Grelot, 2018) and coastal flooding (CEPRI, 2014b). The model was found acceptable in the first context, but needed calibration in the second case. The French State recommends using this model to conduct cost-benefit analyses of flood management projects (Rouchon et al., 2018).

- The model by **Dutta et al.** (2003) was chosen because it is an early example of a model that describes the relationship between flood intensity and damage. It is a simple model supplying a relative damage (i.e. the degree of loss that describes the ratio of loss to the replacement value of the whole building) based only on flood depth; basement, number of exposed floors or other exposure variables are not separate inputs for the model, but are part of its variance. The stage-damage function was calibrated with data published by the Japanese Ministry of Construction, which are based on site survey data accumulated since 1954. The validation with a flood event of 1996 showed reliable results for urban areas. The replacement value of the building has to be provided as input data.

- **FLEMO-ps** (Flood Loss Estimation MOdel for the private household sector) is a multi-variable, rule-based model

estimating relative monetary flood loss to residential buildings as a function of water depth, building type and building quality, without further differentiating between flooded floors and not explicitly considering the existence of a basement (Thieken et al., 2008). The model is empirically derived from data collected from 1697 households affected by the severe flooding of the rivers Elbe, Danube and some of their tributaries in August 2002 in Germany. It can be applied on both the micro- and the meso-scale. Model evaluations based on historical floods in Germany showed that FLEMO-ps is outperforming traditional stage-damage curves in estimating flood loss in the private household sector, except for damages caused by very high water depths (Thieken et al., 2008).

- The model by **Fuchs et al.** (2019b) is a simple model, which supplies a relative damage (i.e. the degree of loss that describes the ratio of loss to the replacement value of the whole building) considering water depth, building area (of all floors) and building (replacement) value as input variables. Differently from other models, it is a function developed for mountain areas, i.e. referring to house building tradition of the Alps and flood processes with sediment transport. It was chosen to test the transferability of a model specialised for mountain environments to a low-land situation. The model was fitted with empirical damage and hazard data. Model validation took place based on a 5-fold cross validation.

- **INSYDE** (Dottori et al., 2016; Molinari et al., 2017b) is a synthetic model based on the investigation and modelling of damage mechanisms triggered by floods, developed for the Italian context. The model is based on a what-if analysis, consisting of the simulated step-by-step inundation of the building and in the evaluation of the corresponding damage as a function of hazard and building characteristics. In total, INSYDE adopts 23 input variables, six describing the flood event and 17 referring to building features; among them, there are all the variables available for the case study and included in Table 1. For the remaining ones, default values implemented in the model were adopted in the test. The model supplies damage in absolute terms by considering the replacement/reconstruction value of damaged components, and by referring only to flooded floors (including basement, if present); however, if required, the model can supply also an estimation of relative damage. INSYDE was validated for different Italian flood events and its performance has been compared to those of other existing models (Dottori et al., 2016; Molinari et al., 2017b; Amadio et al., 2019).

- The model by **Jonkman et al.** (2008) is a simple relative damage model considering water depth and building (replacement) value of all floors as explicative variables, developed on the basis of empirical flood damage data collected in the Netherlands in combination with existing literature and expert judgment. There is no information concerning validation or the robustness of this model. The model is a combined function of content and structure loss. Therefore, to only consider damage on building structure, the original function was rescaled to possibly reach "total destruction" (degree of loss = 1).

295 **Table 2: main features of the models implemented in the blind test.**

| Model | Country and year of development | Hazard context of development | Considered explicative variables | Type of model | Type of results | Economic evaluation | Exposure estimation | Other features |
|---|---|---|---|---|---|---|---|---|
| Arrighi et al. | Italy, 2018 | Riverine floods | h, FA, BA, economic value of the building | synthetic | relative damage | Recovery (based on market value) | flooded floors, (considering also FL and LM) | − zero-damage threshold at water depth 0.25 m<br>− the model estimates also absolute damage |
| Carisi et al. - MV | Italy, 2018 | Riverine floods | h, v, FA, BS, economic value of the building | empirical | relative damage | market value | flooded floors (considering also FL and LM) | |
| Carisi et al. - mono | Italy, 2018 | Riverine floods | h, FA, economic value of the building | empirical | relative damage | market value | flooded floors (considering also FL and LM) | |
| CEPRI | France, 2014 | Riverine, coastal floods | h, BT, FA, BA, NF | synthetic | absolute damage | replacement value | flooded floors | − the model estimates also damage to contents (not considered here) |
| Dutta et al. | Japan, 2003 | Riverine floods | h, FA, economic value of the building | empirical | relative damage | replacement value | whole building | |
| FLEMO-ps | Germany, 2008 | Riverine floods | h, q, BT, FL, economic value of the building | empirical | relative damage | replacement value | whole building | − the model is also capable of estimating damage to household contents (not considered here) |
| Fuchs et al. | Austria/ Switzerland, 2019 | Mountain (high velocity) floods, debris flows | h, FA, economic value of the building | empirical | relative damage | replacement value | whole building | |
| INSYDE | Italy, 2016 | Riverine floods | h, v, q, FA, EP, BA, BT, FL, BS, NF, LM | synthetic | absolute damage | replacement value | flooded floors (considering FL and LM) | − the model estimates also relative damage |
| Jonkman | The Netherlands, 2008 | Riverine floods | h, FA, economic value of the building | empirical | relative damage | replacement value | whole building | |

## 3 Results

### 3.1 Implementation of the models to the case study in a blind mode

With the aim of understanding the impact of exposure estimation on damage assessment and identifying possible common features in the results, Table 3 shows the total exposure and loss figures obtained by applying the nine models to all buildings within the simulated inundation area (877 in total; see Figure 1); note that at this stage of the analysis damage observations were not considered yet for comparison purposes (see Section 2).

Total exposure estimates differ among the models by a maximum factor of 2.75. With respect to the mean exposure value, single estimations diverge instead by a maximum factor of 1.77. These significant differences mainly result from the fact that some models calculate exposure as the monetary value of flooded floors, while others refer to the whole building (see Table 2). Indeed, focusing on the four models that consider only flooded floors (i.e. Arrighi et al., Carisi et al.-MV, Carisi et al.-mono, and INSYDE, see Table 2), total exposure estimates differ by a maximum factor of 1.22. Minor differences are due to the (non-)consideration of the presence of a basement as well as to the adoption of replacement/recovery values rather than market values as parametric cost for the estimation. These results point out that a first source of variability among model outcomes lies in the approach for exposure assessment.

Total damage estimations differ among the modelling approaches by a maximum factor of 12.6, which is limited to 3.1 with respect to the mean value of total damage estimations, suggesting that the shape of the damage functions exacerbates the variability of models' outcomes due to exposure estimation.

Similar conclusions can be drawn when looking at individual (i.e. building by building) estimations reported in Fig. 2 (exposure values) and Fig. 3 (damage values). Individual estimations of exposure differ by a mean factor of 3.5. The models of Fuchs et al., Jonkman et al., Dutta et al. and FLEMo-ps use the replacement value of the whole building as a reference for calculating the degree of loss and are thus relying on sensibly higher exposure values than others. Individual damage estimates differ on average by a factor of 28, with the highest differences due to the models of Fuchs et al. and Dutta et al. (maximum expected damage) and to the model of Arrighi et al. (minimum expected damage). Such results can be partly explained by the adoption of the whole building value for exposure estimation (see also Sect. 3.2) as regards high estimations, and by the zero damage threshold for water depths lower than 0.25 m for low estimations. In detail, the weight of the zero damage threshold on the final damage figure has been calculated as a percentage ranging from 7 to 32 %, depending on the considered model.

330 **Table 3: Estimates of the monetary value of exposed assets and damage, for all the buildings in the flooded area. The first column reports the total value of exposed assets (n.a.= not applicable). The second and the third column report, respectively, the total damage and the unit damage per m². The fourth and the fifth column report the ratio between estimates and mean value of estimates (reported in the last row), for exposed assets and damage, respectively.**

| Model | Monetary value of exposed assets [M€] | Monetary damage [M€] | Unitary monetary damage [€m⁻²] | Monetary value of exposed assets/mean value [-] | Monetary value of damage/mean value [-] |
|---|---|---|---|---|---|
| Arrighi et al | 392 | 12 | 35 | 0.78 | 0.25 |
| Carisi et al. - MV | 368 | 20 | 80 | 0.73 | 0.40 |
| Carisi et al. - mono | 368 | 30 | 118 | 0.73 | 0.59 |
| CEPRI | n.a. | 25 | 71 | n.a. | 0.50 |
| Dutta et al. | 889 | 155 | 225 | 1.77 | 3.10 |
| FLEMO-ps | 468 | 58 | 230 | 0.93 | 1.15 |
| Fuchs et al. | 889 | 102 | 147 | 1.77 | 2.03 |
| INSYDE | 395 | 21 | 69 | 0.79 | 0.41 |
| Jonkman et al. | 889 | 29 | 42 | 1.77 | 0.58 |
| **Mean** | **502** | **50** | **-** | **-** | **-** |

335

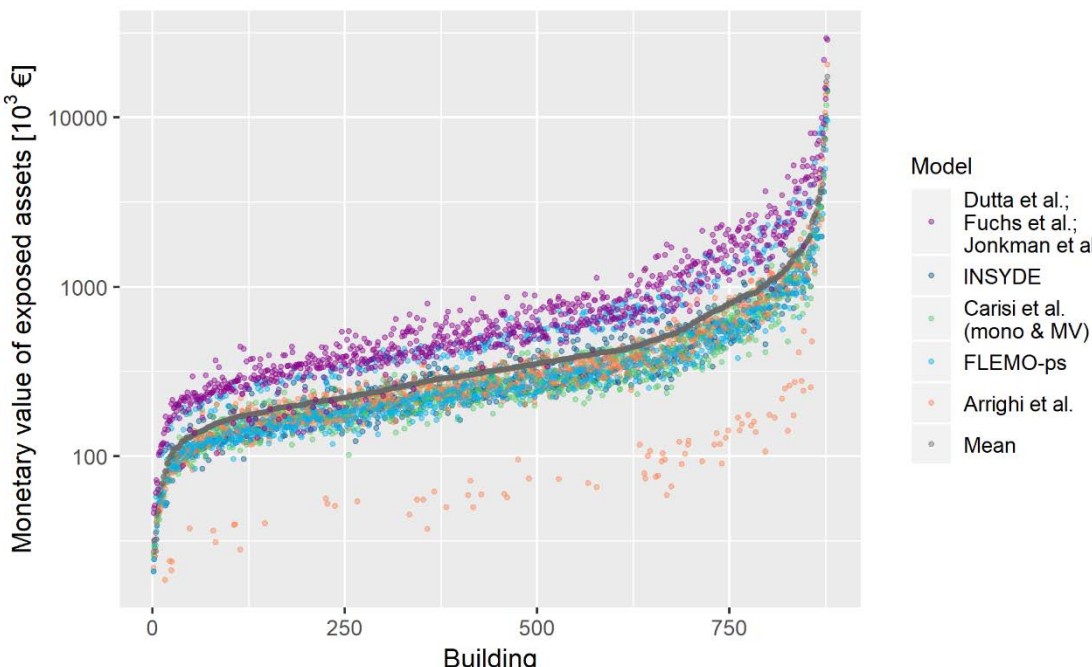

**Figure 2: Individual estimates of the monetary value of the exposed assets for all the buildings in the flooded area. Data are ordered according to increasing value of mean estimate (in grey).**

Figures 2 and 3 further highlight a common trend in exposure and damage values supplied by the different models, also confirmed in Fig. 4 and 5, showing the Pearson's correlation coefficients for individual (i.e. building by building) exposure and damage estimates. The figures show a very high correlation of exposure estimations and a weaker, but still notable, correlation of damage estimations. This finding supports previous results on the importance of damage functions in determining the main differences in model outcomes. In particular, Fig. 5 shows that a higher correlation exists between absolute damage estimates supplied by the two synthetic models INSYDE and CEPRI, among multi-variable models (INSYDE, CEPRI, Carisi et al. - MV and FLEMO-ps), and among simple models (Carisi et al. - mono, Dutta et al., Fuchs et al. and Jonkman et al.), which reflects the consistency between models based on comparable conceptual frameworks.

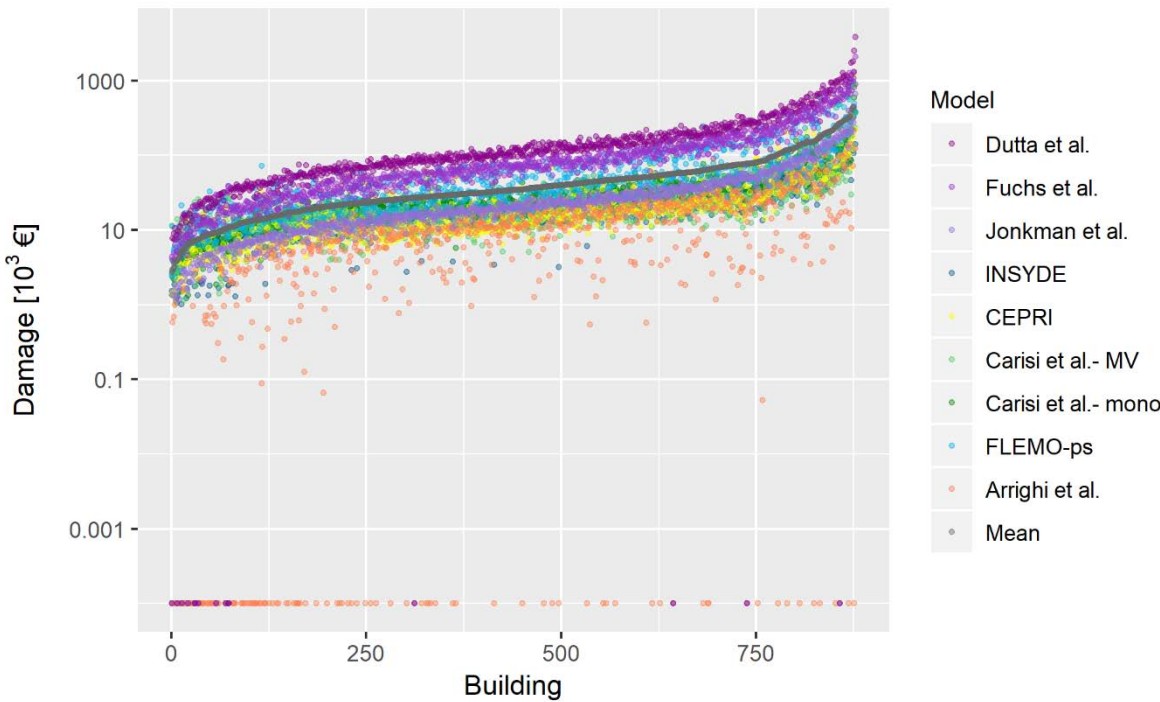

Figure 3: Individual estimates of the monetary damage for all the buildings in the flooded area. Data are ordered according to increasing value of mean estimate (in grey). Zero damages are due to the modelling assumptions behind the specific damage models (i.e. 0.25 m water depth threshold for damage occurrence in Arrighi et al. and 0.01 m water depth threshold in Dutta et al. and Jonkman et al. to distinguish between flooding and surface water runoff)

A comparison between correlation coefficients for absolute and relative damage estimations in Fig. 5 conversely highlights the importance of exposure assessment on the final damage figures. For instance, the low correlation among absolute damage estimates supplied by the model of Arrighi et al. with those from similar models (i.e. simple, low-variable models like Carisi et al. - mono, Dutta et al., Fuchs et al. and Jonkman et al.) can be explained by the fact that the approach adopted by Arrighi

et al. for the evaluation of exposure is considerably different from those adopted by the other comparable models; specifically, the model calculates the monetary value of damage as a function of the recovery cost, which is assumed equal to 15 % of the

360 market value of exposed floors (see Sect. 2). Accordingly, when relative damage estimations are considered, the values of Pearson's correlation coefficient increase. The weight of exposure assessment is also evident when correlation among absolute damage estimates supplied by the four simple, empirical models (i.e. Carisi et al. – mono, Dutta et al., Fuchs et al. and Jonkman et al.) are considered, with models of Dutta et al., Fuchs et al. and Jonkman et al. using the same exposure assessment approach (see Sect. 2) and thus being more correlated among them than with the model Carisi et al. – mono; on the opposite, when

relative damage estimations are considered, the correlation coefficients for the four models are comparable. At last, the weight of exposure arises when correlation between absolute damage estimates supplied by Carisi et al. – mono versus INSYDE are considered. The couple consists of two conceptually different models (in particular, a simple, empirical model versus a multi-variable model), but it shows high correlation. This can be explained by the adoption of very similar approaches for exposure estimation by the considered models (see Sect. 2 and Table 3); in fact, when relative damage estimates are considered

correlation decreases.

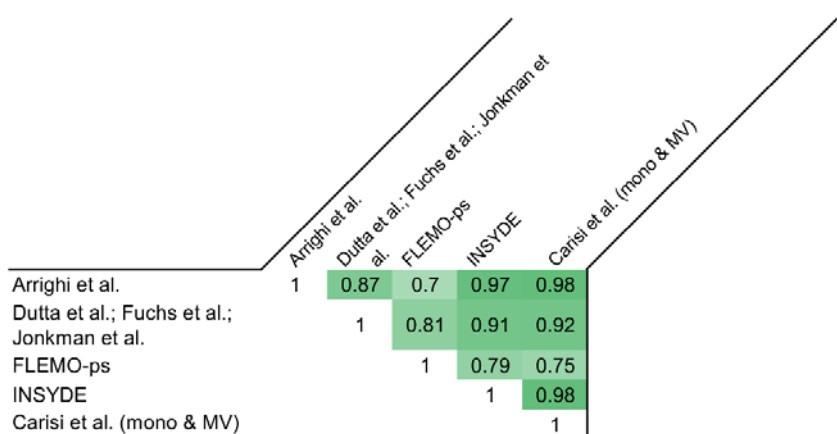

**Figure 4: Pearson's correlation coefficient for individual exposure estimates supplied by the models with reference to all the buildings in the flooded area (the darker the colour, the stronger the correlation).**

| | | | FLEMO-ps | Carisi et al.- MV | CEPRI | INSYDE | Arrighi et al. | Dutta et al. | Fuchs et al. | Jonkman et al. | Carisi et al.- mono |
|---|---|---|---|---|---|---|---|---|---|---|---|
| MULTI | EMP. | FLEMO-ps | 1 | 0.52 | 0.50 | 0.67 | 0.48 | 0.67 | 0.72 | 0.69 | 0.60 |
| | | Carisi et al.- MV | 0.31 | 1 | 0.89 | 0.86 | 0.41 | 0.60 | 0.62 | 0.59 | 0.77 |
| | SYNTH. | CEPRI | -- | -- | 1 | 0.89 | 0.29 | 0.57 | 0.56 | 0.56 | 0.76 |
| | | INSYDE | 0.60 | 0.46 | -- | 1 | 0.55 | 0.73 | 0.74 | 0.73 | 0.87 |
| | | Arrighi et al. | 0.87 | 0.34 | -- | 0.64 | 1 | 0.61 | 0.66 | 0.63 | 0.70 |
| SIMPLE | EMP. | Dutta et al. | 0.87 | 0.33 | -- | 0.70 | 0.97 | 1 | 0.95 | 0.99 | 0.85 |
| | | Fuchs et al. | 0.87 | 0.46 | -- | 0.69 | 0.94 | 0.93 | 1 | 0.98 | 0.82 |
| | | Jonkman et al. | 0.88 | 0.42 | -- | 0.72 | 0.96 | 0.97 | 0.98 | 1 | 0.85 |
| | | Carisi et al.- mono | 0.83 | 0.30 | -- | 0.71 | 0.94 | 0.99 | 0.88 | 0.95 | 1 |

**Figure 5: Pearson's correlation coefficients for absolute damage estimations (top-right of the matrix, in blue) and relative damage estimations (bottom-left of the matrix - in red) supplied by the models with reference to all the buildings in the flooded area (the darker the colour, the stronger the correlation).**

## 3.2 Role of input variables in the determination of divergent models' outcomes

In order to explain the differences observed in the blind implementation, models were compared in terms of trends and variance of individual damage estimates, for classes of values of input variables, and by considering one variable at a time. The objectives of the analyses were to investigate whether the consideration of a specific input variable influences the outcome of a model with respect to the other ones, whether the inclusion of more explicative variables may be considered as a possible source of variation, and to identify the most influencing parameters on the final output of the models.

The input variables considered were: the mean value of the water depth in the building area (h), the footprint area of the building (FA), its external perimeter (EP), the presence of basement (BA), the building type (BT), the building structure (BS), the finishing level of the building (FL), the number of floors (NF), and the level of maintenance (LM). The results are shown in the boxplots reported in Fig. 6 and 7.

An expected increasing trend in damage as a function of the variables related to the extensive properties of the buildings (FA and EP) can be seen, with limited data variance in the case of those models considering other explicative variables than FA (e.g. EP), as INSYDE. As highlighted in the previous section, the models of Dutta et al. and Fuchs et al. show markedly different results, i.e. higher estimates than other models in all classes. This cannot be totally attributed to the fact that such models consider the whole building for calculating exposure, as this is true also for the model of Jonkman et al., which supplies results that are comparable with the ones of other models. Instead, one possible reason may be found in the different origins of the models. In fact, contrarily to all other models, the model of Fuchs et al. was developed for mountainous regions where floods are usually characterised by high sediment transport and deposition, which increases the damage, other variables being equal. In the case of Dutta et al. the detection of the reason for the remarkably higher damages is more elusive, given the lack of detailed information on model derivation, which makes the original model environment not known either for hazard or

exposure variables. In addition, this model is based on survey data collected since 1954 in Japan, meaning that the data used might not be consistently representative for the current flood vulnerability (and in a European environment). The general increasing variance of the estimates with FA and EP classes can be explained by the intrinsic variability of the features characterising larger buildings: they can be apartment buildings rather than semi-detached houses or big villas, with one or more floors; moreover, in the case of apartment buildings, the level of maintenance can change from flat to flat.

Figure 6 indicates the importance of BA as an influencing variable in modelling flood damage for the given event. This is particularly evident in the results provided by CEPRI and INSYDE, which estimate median damages ranging respectively from 13 600 €and 15 400 €for buildings without basement to 26 300 €and 24 500 €for buildings with basement, as opposed to the performances of other models, which did not differ significantly for the two building categories.

Regarding damage estimates for different water depth classes, Fig. 6 indicates an acceptable convergence among model results, especially for the shallower water depth classes, if excluding the results of the models of Dutta et al. and Fuchs et al. (as discussed earlier). However, larger differences are apparent for the highest water depth class (h > 1.5 m). Overall, this result seems reasonable as most of the tested models were calibrated and/or validated for flood events characterised by shallow or medium inundation depths.

Finally, as also emerged in previous studies (Wagenaar et al., 2017; Amadio et al., 2019), Fig. 7 denotes that other variables related to building features do not significantly influence model behaviour. Larger scatter is observed only for the "Apartment" category, which is intrinsically characterised by larger variability, especially in terms of extensive parameters.

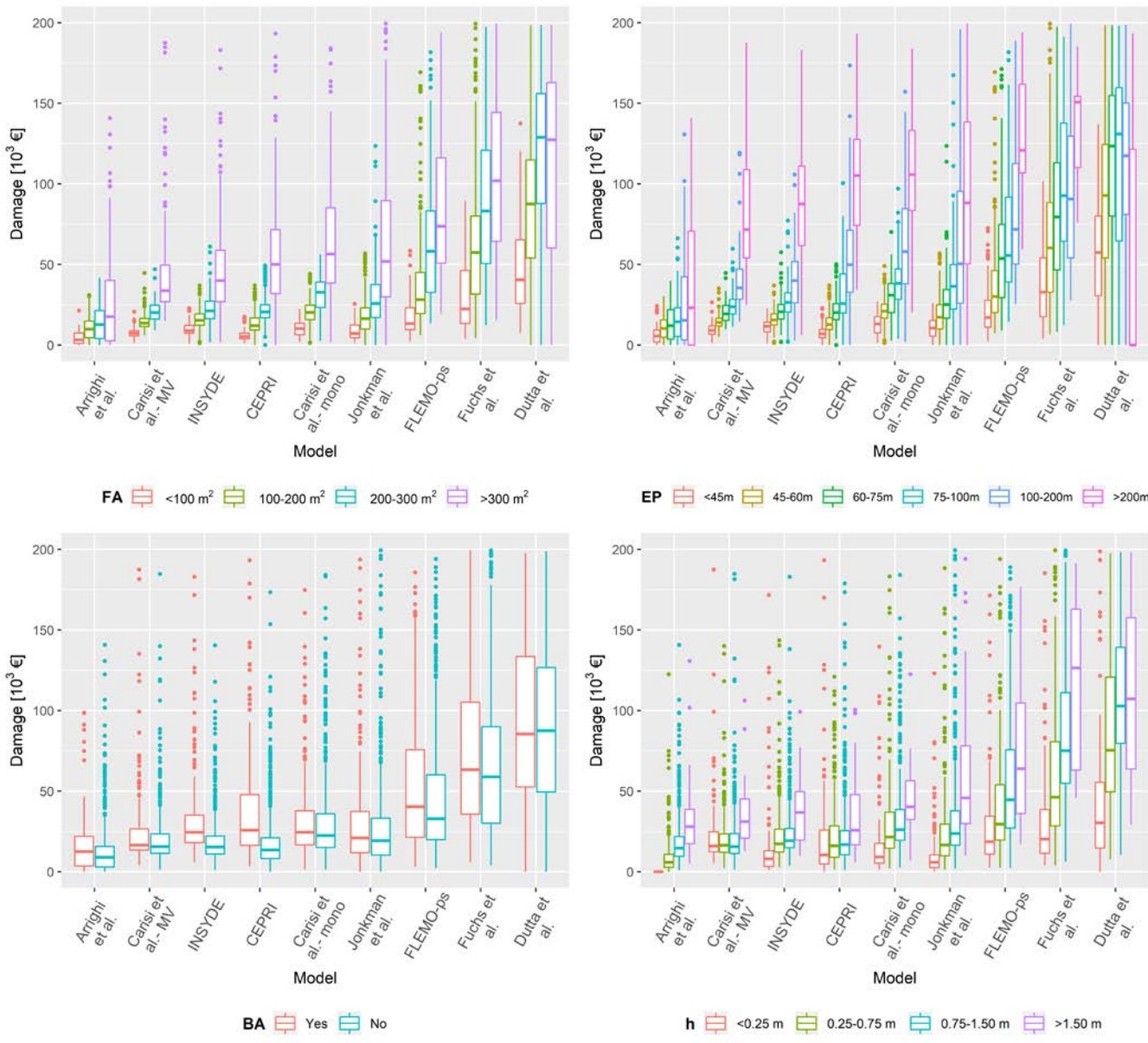


**Figure 6: Boxplots of damage estimates obtained with the tested models, for different classes of: footprint area – FA (Top-left), external perimeter – EP (Top-right), presence of basement – BA (Bottom-left) and water depth – h (Bottom-right). Models are organised according to increasing value of total damage estimates.**

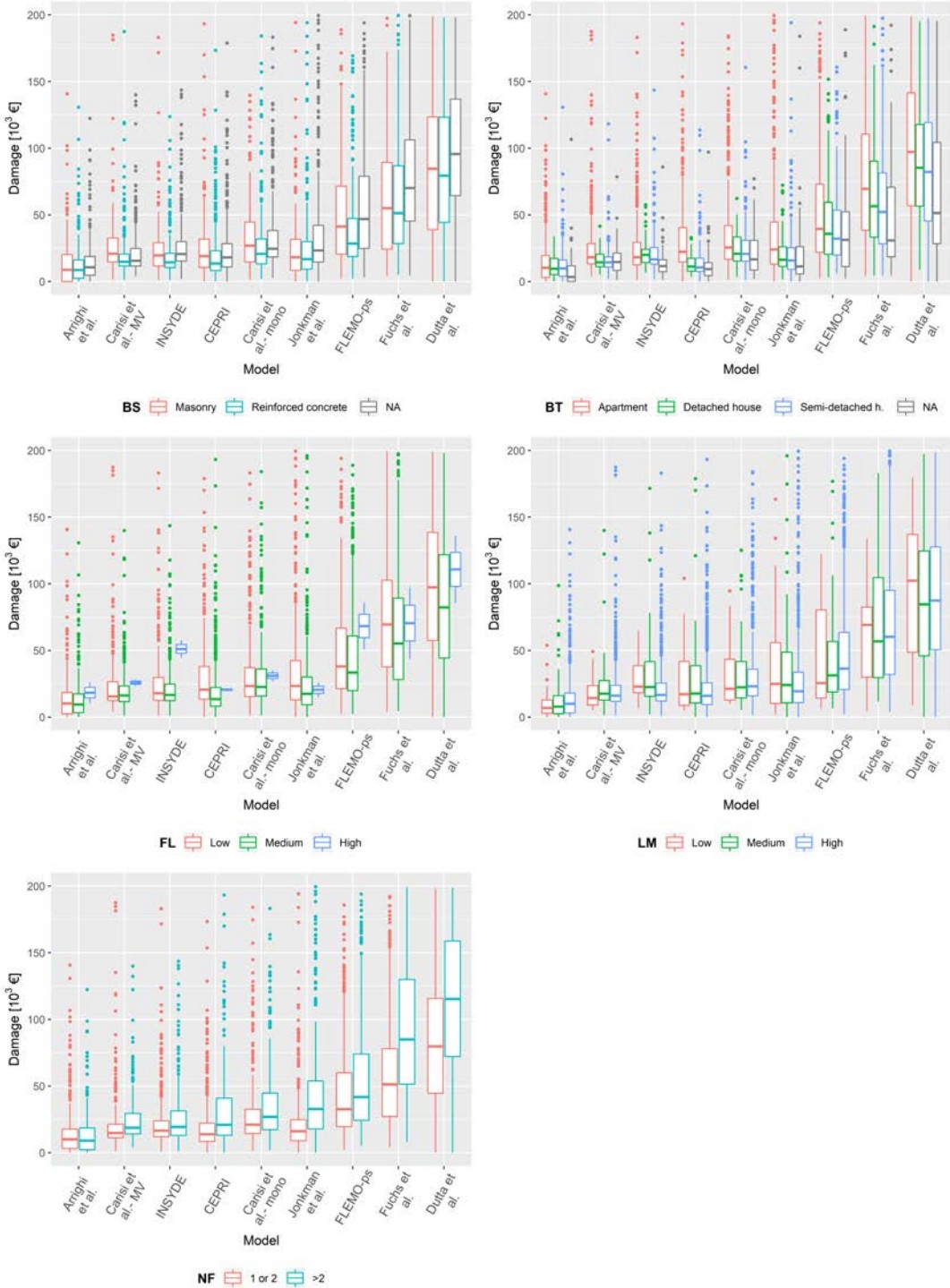

**Figure 7: Boxplots of damage estimates obtained with the tested models, for different classes of building structure – BS (Top-left), building type – BT (Top-right), finishing level – FL (Middle-left), level of maintenance – LM (Middle-right) and number of floors – NF (Bottom-left). Models are organised according to increasing value of total damage estimates.**

## 3.3 Comparison between estimates and observations

In order to gain knowledge on models' reliability in the investigated context, estimated losses were compared to observed damages derived from claims. For this purpose, a subset of the buildings within the simulated inundation area was considered, given that claims presented by private owners were available for only 345 buildings. Table 4 summarises the results of the sensitivity analysis by comparing the total observed damage to the total damage estimates obtained with the implementation of the nine models to the subset of buildings. The table confirms the results presented in Sect. 3.1 (i.e. models estimations differ by a factor of around 13) and highlights the systematic overestimation provided by the models with respect to observed

damage, up to a maximum difference ratio of 13.97. Figure S1 in the Supplementary Material, displaying the ratios between estimated and observed damage at the building scale for different flood depth classes, suggests that detected differences do not depend on the hydraulic features in the inundated area but mainly on the damage models, for which the individual differences are similar across all flood depth classes. In this regards, Table 4 The table also indicates the better performances of the Italian/local models (marked with the "IT" suffix in the table), with Arrighi et al. showing the lowest difference. However, by

looking at its features, it is possible to state that even this last model tends to overestimate damage. First, because it does not consider clean-up costs (like INSYDE and CEPRI), which are instead included in the observations. Second, because the lower value of the total damage with respect to other models is partly due to the effect of the zero damage threshold for water depths lower than 0.25 m (see Sect. 3.1); indeed, as highlighted in Fig. 8 (showing the comparison between individual observed and estimated damages), a zero damage was expected by this model also for those buildings which experienced a significant loss.

Interestingly, Table 5 finally shows that some of the imported models perform similarly or better than Italian models, with specifically high performance of CEPRI.

**Table 4: Observed damage data versus estimates of the total monetary damage for the subset of buildings with claims (n.a.= not applicable). The second and the third columns report, respectively, the total damage and the unit value of damage per m². Mean**
**value of estimates is reported in the last row. The fourth column reports the ratio between estimates and observed damage. Suffixes are used to track the original country of the models (IT=Italy, FR=France, JP=Japan, DE=Germany, AT= Austria, NL= The Netherlands).**

| Model | Monetary damage (M€) | (Unitary monetary damage [€m⁻²]) | Calculated damage/observed damage [-] |
|---|---|---|---|
| *observed* | *6* | *60* | - |
| Arrighi et al. (IT) | 6 | 43 | 1.00 |
| Carisi et al. - MV (IT) | 8 | 85 | 1.4 |
| Carisi et al. – mono (IT) | 12 | 132 | 2.19 |
| CEPRI (FR) | 10 | 74 | 1.72 |
| Dutta et al. (JP) | 77 | 265 | 13.97 |
| FLEMO-ps (DE) | 30 | 320 | 5.30 |
| Fuchs et al. (AT) | 50 | 171 | 9.03 |
| INSYDE (IT) | 9 | 85 | 1.69 |

| | | | |
|---|---|---|---|
| Jonkman et al. (NL) | 14 | 49 | 2.61 |
| Mean | 24 | n.a | 4.06 |

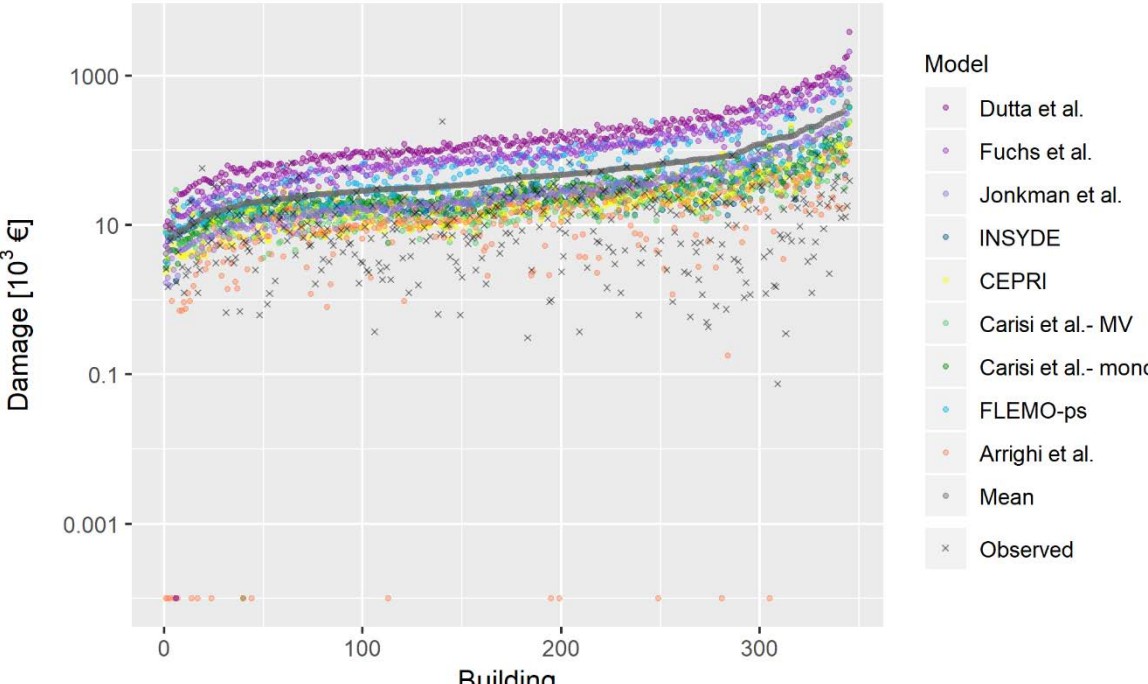

**Figure 8: Observed damage versus individual estimates of the monetary damage for the subset of buildings with claims. Data are ordered according to increasing value of mean estimate (in black).**


Figure 8 generally corroborates findings of Sect. 3.1, depicting a common trend in the models with largely different individual damage estimates. Moreover, it also emphasises the overestimation made by the models with respect to observations, with the latter not showing the common trend followed by the models. This evidence is supported by the results of the correlation analysis (Table 5), which reveals only marginal correlation between calculated losses and reported claims. On the contrary, 465 the high correlation among models (see Fig. 5) raises the question of whether reported claims and damage estimation are comparable.



**Table 5: Pearson's correlation coefficient of observed damage and estimates supplied by the models with reference to the subset of buildings with claims. The acronyms in parentheses indicate the original countries of the models (IT=Italy, FR=France, JP=Japan, DE=Germany, AT= Austria, NL= The Netherlands).**

|  | **Observed** |
|---|---|
| **Arrighi et al. (IT)** | 0.26 |
| **Carisi et al. - MV (IT)** | 0.10 |
| **Carisi et al. – mono (IT)** | 0.12 |
| **CEPRI (FR)** | 0.15 |
| **Dutta et al. (JP)** | 0.13 |
| **FLEMO-ps (DE)** | 0.13 |
| **Fuchs et al. (AT)** | 0.15 |
| **INSYDE (IT)** | 0.18 |
| **Jonkman et al. (NL)** | 0.13 |

**3.4 Analysis of damage claims**

In order to explain the differences between model results and observations, a thorough analysis of claims data was carried out. Given the general overestimation provided by the models, first we focused our attention on 44 buildings that are characterised by very low values of observed damage (less than 1500 € in 2002 currency), referred to as "outliers" hereinafter. Table 6 reports the mean value of water depth, footprint area and external perimeter (i.e. the variables which most influence damage according to the analysis performed in Sect. 3.2) calculated for this subset of buildings and for all the buildings with claims.

Table 6 indicates that low damages cannot be explained by significant differences in these influencing variables, given that both datasets show comparable values. Moreover, based on informal conversation with representatives of the Committee of Flooded Citizens in Lodi, it is possible to postulate that existing outliers cannot even be explained by the adoption of individual mitigation actions (like temporary flood barriers or pumps), because no official flood warning was issued and, consequently,

no lead time was available to undertake precautionary measures. Finally, from the analysis of building pictures available in Google Street View, we can state that outliers are not due to the presence of steps or other elements which increase the height of the building with respect to the ground level, reducing its vulnerability.

**Table 6: Mean value of water depth (h), footprint area (FA) and external perimeters (EP) for all buildings with claims and for the**

 **outliers' subset.**

| Dataset | Mean value of influence variables | | |
|---------|---------|---------|---------|
| | H [m] | FA [m$^2$] | EP [m] |
| **outliers** | 0.79 | 264.80 | 78.07 |
| **all claims** | 0.86 | 265.56 | 77.32 |

On the contrary, examining in detail the outlier claims, the following evidence arose:

- 27 % of outliers refer to claims with no detailed information about the type of damage, hindering the thorough understanding of low loss values in these cases;
- 32 % of outliers can be explained by the fact that declared damage regards only garages or boilers, while damage models typically assume a residential use of the building, with the presence/damage of all technical systems (i.e. heating, electrical, and water);
- 41 % of outliers refer to paltry claims, even in case of significant water depths (around 1 m), which are mostly related to painting of walls and replacement of doors and windows.

In view of the large proportion of paltry claims, it was attempted to understand the causes of declared damages. For this, we calculated the frequency of damage occurrence to different building components (i.e. damage to walls, damage to floor, damage to doors and windows and damage to systems) in the different claims and for three water depth classes (Fig. 9). Findings reveal an unexpected behaviour with respect to existing knowledge on damage mechanisms; in particular:

- damage to floors is found to be declared mostly for water depths higher than 1.5 m, although in principle this type of damage should be poorly related to water depth;
- frequency of damage to doors and windows decreases moving from the middle to the highest water depth class, as opposed to expectations (because of the occurrence of damage to windows with higher water depths);
- no damage to water, sanitary and heating systems is found to be declared for water depths higher than 1.5 m, contrarily to what can be expected by considering the typical height of the technical installations in Italian houses (Dottori et al., 2016).

According to our interpretation, inconsistency between expected and declared damage can be attributed to the fact that what is declared by citizens does not correspond to the actual budget required to replace or reconstruct the whole physical damage suffered by the building, but rather to the amount of money needed to bring the building back to a desired level of functionality, according to the financial resources of the owner: for this reason, for example, not all flooded doors are replaced and flooded floors always rebuilt. This would explain why synthetic models overestimate observed damage, as they are usually based on full replacement/reconstruction costs. Likewise, it would explain why the model by Arrighi et al. performs better than others: indeed, the recovery value adopted by this model is defined as the average difference between the market value of new buildings and that of equivalent, older buildings requiring renovation. It is then sensible that this value reflects a balance

between the two opposite extreme behaviours of buyers (which, in turn, depend on their financial resources): i.e. to completely renovate the building or to bring it back to a minimum level of functioning. In our view, such behaviours can be compared with those of flooded owners.

Moreover, declared monetary damage is strongly correlated to the expectations that citizens have to be reimbursed. This expectation is low in Italy, when in most cases limited funding is available for the compensation of private damage, which

implies strict criteria and thresholds for compensation (often much lower than the effective damage). In addition, all costs must be proved by the citizens by means of official invoices. For all these reasons, citizens often prefer taking advantage of the "black market" rather than declaring damage (Cellerino, 2004). This would also explain why empirical models (derived from claims) developed in regions with high expectations and then high values of declared damage (like Germany), overestimate the observed damage in this case study.


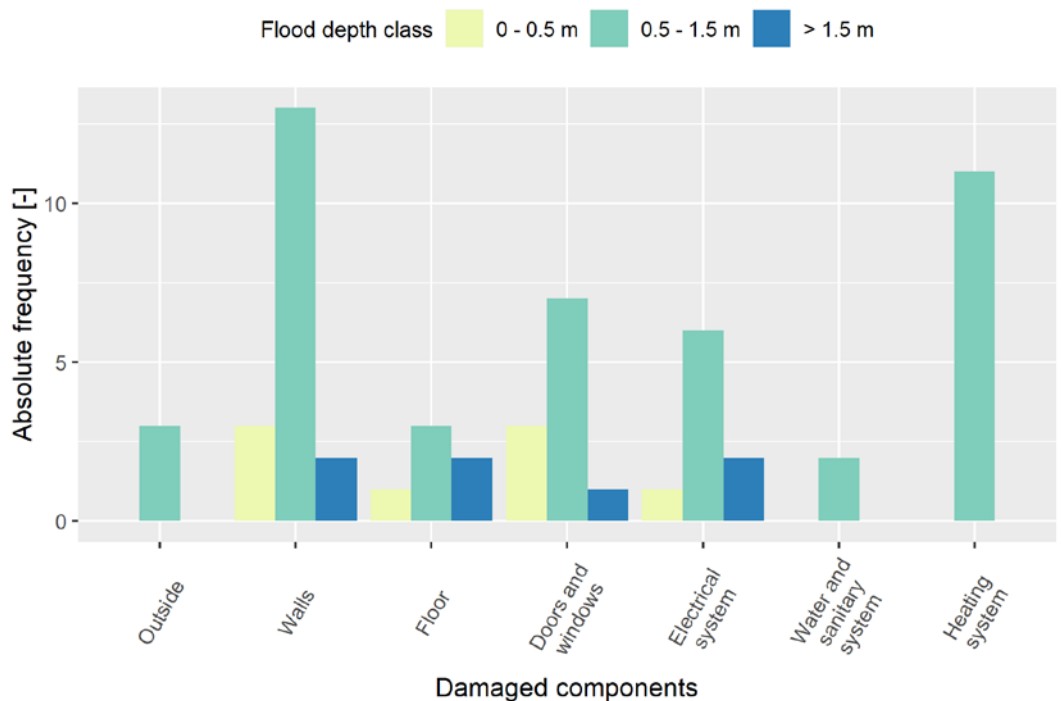

**Figure 9: Absolute frequency of declared damage to the different building components in the outlier dataset for different water depth (h) classes.**

From another perspective, in order to explain the scatter that is generally observed in real damage data with respect to water depth (note that the value of Pearson's correlation coefficient between observed damage and water depth is 0.11), we focused the attention on 13 paired buildings, whereby the term "paired" refers to buildings with the same vulnerability characteristics (i.e. building type, building structure, level of maintenance and finishing level) as well as similar values of hazard parameters

(i.e. water depth and flow velocity), but significant difference in the declared unit damage (€m$^{-2}$).

The analysis revealed that:

- considerable differences are attributable to declared or undeclared replacement costs of systems, rather than of doors and windows; this can be explained again by what is considered as monetary damage by citizens.

- in other cases, costs related to similar damage (e.g. cost of painting, cost of replacement of doors) differ a lot, even by a factor of 10. This discrepancy might be explained by wrong assumptions concerning the finishing level and/or

the building type. More specifically, the actual conditions of buildings with high damage values could have been better than what was assumed for the blind test, using cadastral data as reference (see Table 1).

- sometimes the above two factors add up, further increasing the differences among paired buildings in terms of declared damage.

Scatter in claims data can then be partially explained by the influence of local parameters (like the finishing level or the building

type) which are difficult to assess at the micro-scale without a detailed field survey; nonetheless, it seems that the influence of such parameters on damage estimation for the analysed models is very low (see Sect. 3.2) so that the latter are reliable only when applied at the meso-scale.

Overall, the analysis of claims highlighted that observed damage data need to be carefully analysed before being used for model validation, since their comparability with damage estimates is not always guaranteed.

**4 Discussion**

Results from the previous analyses were critically analysed in order to gain general knowledge on the transferability of damage models and reliability of damage estimates, and, in particular, to answer to the two specific research questions set in the Introduction.

Concerning the performance of local versus imported models, the blind test corroborated literature results (Cammerer et al.,

2013), suggesting that model transferability depends on the consistency between the context of implementation and the original calibration context, as far as both hazard and exposure/vulnerability features of exposed buildings are concerned. In fact, in the blind test, models developed for the Italian territory and for riverine floods performed generally better than models derived in other countries or for different flooding features, e.g. mountain areas. Such a result was not surprising as models providing good results have proven to perform well also in other Italian validation case studies, e.g., Arrighi et al. worked well also for

the 2010 flood in Veneto Region (Scorzini and Frank 2017); the same applies to INSYDE and the two models by Carisi et al., which were tested in other Italian flood events (Amadio et al. 2019). On the contrary, the imported model of Dutta et al. was already found to not properly work in Italian cases (Scorzini and Frank 2017). Still, the analysis of damage claims revealed that, as far as empirical models are considered, transferability could depend also on comparability of the compensation contexts, given that observed losses on which empirical models are calibrated may depend on citizens' expectations of

reimbursement.

Regarding instead the second question, literature suggests that the inclusion of several influencing variables should increase the accuracy of a model (Merz et al., 2013; Schröter et al., 2014; Van Ootegem et al., 2018). Still, the blind test highlighted that such an evidence can be invalidated by the lack of availability/consistency of input data between the calibration and the implementation context. Indeed, the models considered in the blind test were designed to be used with the type of data usually

available in the original context, which generally differ from the data available in the Lodi case study, i.e., models use different proxy variables for the same explicative parameters. For this reason, assumptions had to be undertaken to allow the application of a model in the case study area (see Sect. 2). For example, the building categories (BT) assumed by CEPRI ("apartment"/ "single storey building"/ "multi-storeys buildings") are different than the Italian ones ("apartment"/ "detached"/ "semi-detached") so that a correspondence has to be defined, also on the basis of the number of floors (NF); specifically apartment

is defined by BT ="apartment", "single-storey" is defined by BT = "detached" or "semi-detached" and NF = 1, "multiple-storeys" is defined by BT ="detached" or "semi-detached" and NF > 1. Correspondence among building categories was defined also for the implementation of FLEMO-ps, although in this case the task was quite straightforward, since the German building categories are almost coincident with the Italian ones (FLEMO-ps distinguishes between "Multi-family house" / "Semi-detached house" / "One-family home"). Assumptions on input variables may reduce the reliability of the original model

because of an improper/inaccurate "adaptation" of the available data, thus reducing the advantage of using many variables. This also explains why the simple models by Jonkman et al. and Carisi et al. - mono provided comparable or better results than those obtained from multi-variable models like FLEMO-ps or CEPRI. Also, the use of additional variables may have different impact depending if, in the application area and differently for the original model development strategy, this information is retrieved at the building scale or known as aggregated variable. Consultations of experts with local knowledge were needed to

help in the correct interpretation and use of the available input data for the Lodi case study. Importantly, the blind test highlighted that none of the tested models (being them local or imported, simple or multi-variable) seemed appropriate to estimate flood damage at the building scale in the given context; still, models' performance improved when aggregated damage data were taken into account. In fact, considering the 345 buildings for which a claim was known, all models' estimates differed significantly individually (Fig. 8), but some of them indicated a total damage figure close to the observations (Table 4). Besides

the already discussed potential biases of claim data, this duality suggests that model uncertainty may be balanced in aggregated results, i.e. the lump-sum might be more reliable than the individual results. This raises the question of which is the right spatial scale (that is the level of complexity) of analysis to get reliable results, and for which objective. For example, by implementing the simpler, lump-sum model DELENAH_M (Natho and Thieken, 2018), an adaptation of the UNISDR method for national damage estimates (UNISDR 2015) in developed countries taking Germany as a study case, the estimate of the aggregated

damage for the 345 buildings with claim data is 4.3 M€ This estimation is affected by an error which is comparable or lower than errors supplied by the micro-scale models (see Table 7), although being obtained with a simple calculation and in a blind mode, i.e. using the average damage ratio for severe floods and the average housing size derived from German survey data (Thieken et al., 2017) on flood losses in the housing sector (note that in this case underestimation of total damage is due to the adoption of a conservative housing size, so that the estimation must be intended as a minimum estimate or a lower bound). Is

this assessment useful for flood risk mitigation? Which is then the advantage of using micro-scale models? Is there a level of spatial aggregation which supply reliable, more informative estimation than a simple lump-sum at the municipality level? Answers to these questions will be objective of further investigations by the research groups involved in the test.

## 5 Conclusions: lessons learnt from a modeller's perspective

The blind test conducted in this study represented an opportunity not only to deeply investigate the transferability of tested
models and the reliability of their estimations, especially regarding the potentialities of local and multi-variable models, but also to increase authors' awareness on strengths and limits of flood damage modelling tools. As concluding remarks, we report in the following section take-home messages synthesising lessons learnt from the blind test, from a modeller's perspective.

First, a former source of variability among models' outcomes lies in the approach for exposure assessment, which then represents a critical, often overlooked, step in flood damage modelling. In particular, assessing exposure coherently with the
approach originally adopted in model development is key to preserve the original reliability; in this regard, the blind test showed that the different approaches applied within the models demand for a clear definition and differentiation of the terms "exposure value" and "building value". Nonetheless, the blind test indicated a common overestimation, confirmed also in other case studies (Zischg et al., 2018; Cammerer et al., 2013; Thieken et al., 2008; Fuchs et al., 2019b; Arrighi et al., 2018a, 2018b), in terms of number of buildings damaged by a flood event (i.e. the number of buildings with claims is significantly lower than
those exposed to the flood). This might be attributed to the fact that not all affected building owners asked for compensation, or that some buildings are not affected by the flood due to local micro-topographical conditions or due to the installation of protection measures. However, it might also highlight problems in the current strategy adopted to identify exposure (e.g. by not considering building elevation).

A second critical issue in flood damage modelling is the transfer of models in space and time, with difficulties on predicting
the expected performance of a given model when applied in a different context (e.g. Jongman et al. 2012, Cammerer et al., 2013; Wagenaar et al., 2018). Accordingly, flood damage modellers should always be cautious when applying a flood damage model to a new context. Their general trust towards the model performance in the new study area must be in the first instance limited; however, model validation (ideally, over multiple datasets) can significantly increase the trust level.

But validation of damage models invariably relies on observed damage data, either from insurance claims, governmental
reimbursement claims or direct surveys, all of which are generally intended as "reality". Indeed, it is often the case that empirical data are used in validation analyses without any possible preliminary evaluation on their quality and significance, simply because no ancillary information is available, as for instance for insurance data (André et al. 2013; Spekkers et al. 2013; Zhou et al. 2013; Wing et al. 2020). In this context, the blind test highlighted that "reality" depicted by observations is not univocal, so that observed data must be carefully investigated before their comparison with model outcomes, as they may be
addressing different types of damage, damage to different components, or being incomplete. Based on this consideration, flood damage modellers must be always cautious when drawing conclusions from validation analyses: if a model does not fit well

to some empirical data, this does not necessarily reflect the inability of the model in general terms but attention has to be drawn to input data quality and vice versa. This also points out the importance of collecting not only flood damage data, but also ancillary information on flood hazard and vulnerability of affected assets in the ex-post flood phase (Merz et al., 2004; Thieken et al., 2005; Ballio et al., 2015; Thieken et al., 2016; Molinari et al., 2017a; Molinari et al., 2019). Moreover, consultations of experts with local knowledge can help in the correct interpretation and use of observed damage data.

In absence of data (or appropriate data) for validation, the application of several models might be useful to quantify mean and variance and provide a range of uncertainty of the estimations (Figueiredo et al., 2018); a good agreement of model results, in particular with the models developed for context similar to the one under investigation, can significantly increase the trust level in model performance. In this regard, the blind test stressed that damage models have to be compared in their original form, meaning that, for instance, relative damage models relying on the total building value cannot be directly compared to the ones relying on only the first floor.

As a general recommendation, to select a damage model for an application in a different country, it is important to verify the comparability between the original and the investigated physical (in terms of hazard and building features) and compensation context, as well as the availability and coherence of the input data. Moreover, when transferring a model (in space or time), proxies of input variables are frequently needed, and the modeller must be prudent in this step. A good understanding of both the data used during the model development and the data gathered for the new application is crucial, as the attribution of uncertainty becomes elusive afterwards, if this step is neglected. The blind test highlighted that the real effort of transferring the models to the given implementation context was related to finding the "right" required data, while the costs of implementing assumptions about exposure and calculating the damage value were negligible. To support transferability, there is then a need to precisely describe how the models were developed, which variables were included and for which specific context. In this regard, a protocol or standardised information for all models would help in finding the most appropriate tool in a given context; in fact, at present, details about origin, calibration, assumptions, field of application, etc. of existing models in the literature are few and sparse. A new promising attempt in this direction is represented by the Flood Damage Model Repository, recently launched by Politecnico di Milano (www.fdm.polimi.it) as a research community effort.

Given these considerations, and in contrast with the general approach in which each research group develops its own model for a limited context, authors support a call for a community effort in setting up a common model, with different sub-modules useable for many purposes and regions, and with a flexibility in the required input data.

**Author contributions.** *Conceptualisation of the blind test:* Francesco Ballio; *Management of the blind test:* Daniela Molinari; *Data and results management:* Daniela Molinari, Alice Gallazzi, Marta Galliani; *Models implementation*: Chiara Arrighi, Francesca Carisi, Marta Galliani, Patric Kellermann, Markus Mosimann, Stephanie Natho, Claire Richert; *Elaboration of results:* Daniela Molinari, Anna Rita Scorzini, Alice Gallazzi; *Interpretation of Results – Original Draft:* Daniela Molinari, Anna Rita Scorzini, Francesco Ballio; *Interpretation of Results – Review:* all; *Writing-Original Draft:* Daniela Molinari,

Markus Mosimann, Francesca Carisi, Alessio Domeneghetti, Guilherme S. Mohor; *Writing-Review:* Daniela Molinari, all; *Figures and Tables:* Anna Rita Scorzini, Daniela Molinari.

**Acknowledgements.** Authors acknowledge with gratitude Andrea Nardini (from the Italian Centre for River Restoration – CIRF) and Marianne Skov (from Rambøll, Denmark) for their fruitful suggestions and hints during the developing of the test. Authors are also grateful to three anonymous Reviewers for their meaningful comments and suggestions.

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
