# Peer review of "Are flood damage models converging to "reality"? Lessons learnt from a blind test"

_Natural Hazards and Earth System Sciences, 2020_

## Referee Comment (RC1) · Anonymous Referee #1 · 22 Mar 2020

I find this a genuinely interesting paper and I would like to thank the authors for their work. I support the publication of the article with perhaps a few minor modifications.

There are a few areas where the grammar or writing could be polished although the sense is always clear.

Introduction: The introduction is clearly written, well referenced and summarizes the current state of the literature as I understand it.

Methodology / case study:

It would be interesting to see the maximum flood depths plotted on the map, so as to get a sense of how the flood varied across the flood zone.

[Figure]

Do the authors have any information on the flood map and how closely it matched the flood events that occurred. This is obviously a potentially large source of uncertainty in the results.

In Table 1, where there is a set of discrete responses, is it possible to see what these choices are (e.g. level of maintenance: Low, medium, high, etc).

Table 2 and associated text. Is it possible to give more detail on the adjustments needed to each model to get them to work. This could be due to my misunderstanding, but for example, taking the model of Jonkman et al, two variables are needed (h and FA). However, is the replacement value also needed?

Figure 3 - could the authors explain why there are buildings with no damage? Are these the buildings discussed in Section 3.4?

Line 442 - there is some text missing "given that both datasets show comparable values. in, as"

Line 486 - I find it highly interesting that there is an expectation that many people do not claim through their insurance - this is worthy of a new line of inquiry in its own right (although not within the paper as it is out of scope).

Discussion:

Could the authors say more about what is needed to apply models not developed for Italy to work - for example, does the research suggest simple steps that could increase the performance of models developed in the Netherlands or Germany?

Conclusions:

I would like to commend the authors for their initiative to collect and test damage models to better understand their performance and transferability. I find the discussion useful and I suggest this paper will be interesting for a wide audience working in this field.

Interactive
comment

---

## Referee Comment (RC2) · Anonymous Referee #2 · 7 Apr 2020

This paper compares many different flood damage models to a single validation case. This has been done before but since the last comparison a lot of new (supposedly better) models have become available that can be compared to each other. Furthermore, in previous studies and again in this study a lot of the results are surprising, and we clearly lack some understanding to explain all these results. I therefore believe this is a useful contribution to the literature because it provides yet more model comparison data and provides a lot of interesting speculation on the results. Especially the deep analysis of the limitations of the validation set are very interesting. However, the paper should be more cautious in the way it reasons in the results and discussion section and especially recognize the potential role of coincidences in the results.

• The paper is limited by there being only one validation set. This validation set

has problems as is correctly mentioned in section 3.4. In earlier similar studies (e.g. Jongman et al., 2012) that used multiple validation studies it was common to see that some models performed well on one validation set and bad on another. I think this limitation of the study should be mentioned a bit more clearly, also to point out for future studies of this kind that it may be a good idea to have multiple validation sets (such as in Jongman et al., 2012). Because of this and the next point, the main value of the paper is in a comparison between the models rather than an absolute value judgement of the models.

• The second weakness is also not really highlighted and that is that the number of compared models is too small to draw any strong conclusions. Obviously, it wasn't feasible to select more models but some of the good model performances shown in this paper are pure coincidence, I'm sure about that. A good or bad performance of a model should therefore be seen as a single data point (sample) that could be just be a coincidence. This should be taken into account when drawing conclusions. I think this goes well in the conclusions section but in the results and in the discussion section some of the speculations should be done more cautious. Maybe go through these sections and reevaluate some of the reasoning based on the idea that there is a high level of coincidence in the results. Also be clearer about this limitation and the general idea that some observations could be simply a coincidence, add a few sentences about that somewhere.

• The statement in the discussion about the value of multi-variable models against simple models (line 526) cannot be stated like this. Multi-variable models can only be transferred when there is some overlap in the context between the training and valida-tion data (see Wagenaar et al. 2008). When there is no overlap the transfer obviously wouldn't work no matter how many variables you add to the data. A multi-variable model should always be compared to a single-variable model based on the same data and not to a single-variable model from a different region to then conclude that a multi-variable model isn't useful. Maybe this could be done with a very large number of

models but not based on the tiny number tested in this paper (especially because this observation also contradicts common sense and earlier findings elsewhere).

• The paper title, abstract and introduction puts a lot of emphasis on it being a blind-test. All properly carried out model validations studies don't use any knowledge from the validation data for the model development (standard practice). The constant emphasize on that in this study is therefore a bit misplaced I think. Did earlier studies not follow this approach? I believe they did. Maybe they didn't advertise it this strongly, or maybe this study was more strict or systematic on that but does that really add anything special?

• This study is very similar to the paper Jongman et al., 2012. It might be interesting to add a paragraph in the discussion section to compare the results of the papers. Of course, this paper is comparing much newer and different models. Yet the general observation that the results are very different from each other and that its difficult to make sense of that is the same among the studies.

• The large group of authors suggests that an expert on each of the compared models was included in the paper. This is however clearly not the case for the Jonkman et al. model and hence the paper makes claims about this model that aren't true. Apart from fixing these mistakes (which I pointed out below), I think it should be clarified somewhere which experts worked on which models and whether the expert personally developed the model or interpreted information from literature (much more error prone).

Minor comments:

Line 46: This is grey literature and I can't find it on the internet. Perhaps add one of the very many peer-reviewed journal articles that could also be used to support this statement and are often much older than 2007.

Line 101: I have never seen the term "low-variable" model, maybe consider a different word? Do you mean "single-variable" here?

[Figure]

In section 2.3/table 2. Some basic but important information in this section seems missing for some of the models, such as the origin of the model (country/region) (for Carisi et al. – mono) and the intended flood type (for many of them) and the year the model was made (maybe can be retraced from reference list but not easy for reader). Maybe its good to add this information to table 2 or at least make it clear in the section. Also maybe try to make the different texts a bit more uniform (i.e. present same information in same order).

Line 245. I don't see the significance of this model using a mathematical function. I also know much older models doing that and think its an irrelevant characteristic. Some other damage functions may also follow a mathematical function but just communicate it different.

Table 2: I don't think its correct to classify the Jonkman et al., 2008 model as empirical. It was inspired by many different sources of empirical data but no systematic empirical method was applied and the model is basically expert judgement considering a few empirical data points.

Section 3: I don't like this title very much why not just "results" this is confusing

Line 293-294: Maybe clarify that you look at a subset of the models here and didn't adjust the whole building models to only use the ground floor value (unclear at first).

Line 298 and 299: Can you rephrase this sentence its currently confusing.

Line 305: Could you rephrase this sentence its difficult to follow: "Individual damage estimates differ on average by a factor of 28, with the more frequent factor around 10."

Table 3: Could you split the 3th column, this is confusing and an uncommon form a presenting it. The title of 3.1 is a bit unclear. Could you rephrase it? It's a bit long and I didn't get that with blind mode you mean that its not compared to observations yet (which I expected). Maybe call it "comparison of the models". Or keep the title and clarify in the first sentence that the reader shouldn't expect the comparison to real

observations yet.

Figure 4: This table is interesting but requires some additional discussion. When you have two very different exposure values but the variation among the buildings solely depends on the size of the buildings the correlation between the two very different exposure values is still one. So this figure mostly says something about the characteristics that differ per building (if I understand the figure correctly). This doesn't become clear from the text.

Section 3.2. I like the content of this section but I had to read it twice to fully understand it. Maybe the authors could try to clarify this section a bit more. I think especially the title and the first sentence don't make it very clear at first (its all correct but it's a bit of deciphering for a reader).

Table 4: Could you split the column again like in the previous table.

Table 4: Could you consider including the model origin of all models in this table. That would be a very interesting reminder for the reader. Especially for readers who don't read the entire text this would be very useful.

Line 487: In case of the Netherlands this isn't true. The empirical data in this model is at best used relative. Absolute values are 100% synthetic.

Line 554: Micro-models are essential when measures are undertaken at micro level (e.g. for insurance or studies about elevating specific houses as is common in some countries). When aggregated damages are assessed they may indeed add less information. A second reason why micro-models are important is for at least for the location of buildings. The difference between a house flooding or not is sometimes a matter of just meters. Its therefore important to work with precise models on location even if the other building characteristics are all the same.

Please also note the supplement to this comment:

https://www.nat-hazards-earth-syst-sci-discuss.net/nhess-2020-40/nhess-2020-40-RC2-supplement.pdf

---

## Referee Comment (RC3) · Anonymous Referee #3 · 19 Apr 2020

Review: Are flood damage models converging to reality? Lessons learnt from a blind test

D Molinari et al.

The paper addresses the characteristics, performance and transferability of various flood damage models. It compares the modelling results of nine models by means of a 'blind validation test'. This is a very interesting, innovative and challenging approach which has successfully be mastered by the authors revealing some interesting results. Overall the paper is well-structured and - with only few exceptions - well-written.

Within this very positive general picture of the paper there are some few points I would like to mention, which represent some kind of limitations, most of them of methodolog-

ical type. In my opinion, a main drawback of the whole study lies in the approach to 'validate' the results. The authors claim to assess the models' reliability through a comparison with observed damages of a real flood event (chapter 3.3) and show respective results (for example Table 4 and 5). The following chap 3.4 is then dedicated to explaining why the models results differ so strongly from the observed results. Crucial reasons for this discrepancy found are then assigned to inconsistencies in the damage claims, that is, the validation data. This seems for me like an odd approach. If the damage claim dataset was meant to be used as a validation set, more emphasis should have been put on clarifying inconsistencies and maybe on further filtering the dataset down to a set of reliable data on damage of a reduced number of buildings. The authors only declare having had some informal conversations with experts but the explanations about low damage values observed remain fuzzy. For the approach of this paper a more thorough survey of affected people would have been necessary. The statement in line 471ff "According to our interpretation, inconsistency between expected and declared damage can be attributed to the fact that what is declared by citizens does not correspond to the actual money required to replace or reconstruct the whole physical damage suffered by the building" cannot satisfy and actually triggers a the more philosophical question, whether the 'damage' targeted by the model represents the damage felt by the people affected. I suggest shifting those parts of the chapter 3.4 with the explanation about how the validation dataset was generated to an earlier part of the paper as background information about the approach (where also the various models are described).

Additionally, I am not fully convinced by the way some of the results are analysed and presented, see my more detailed comments below.

Below some additional more specific comments:

- Line 81: "Reality is hardly reproduced by observed data" – I would suggest not to use the term 'reality' since there is no univocal damage value as you prove later on yourself
- Line 86f: "comparative studies over a broad range of test cases are essential for

acquiring more confidence in the reliability of modelling tools" – after having read your paper I would not say that the test case increased the confidence in reliability, I would put the emphasis more towards understanding in detail how certain model results come about - Line 101: "the focus of this study lies in this specific set of models" – what does this mean? - Line 149: "was not uniform, as only some of the owners justified costs for fixing the damage by means of invoices" – if not earlier, here the reader should suspect that the quality of the 'validation' dataset is questionable. I would propose to already link here to further explanations about this 'observed' damage data. - Line 186: consider to call it variation rather than 'difference' - Line 195: unclear for me, until here I thought the observation data was derived from damage claims made after the flood event. How can you us 'official claims' to explain inconsistencies between estimations and observations? Or is the officially claimed data different from the claims mentioned earlier? Then you could have used the official ones for validation? - Line 207: does only this model use stage-damage curves? Or why is that here mentioned explicitly?

Comments on presentation of the model results: - The detailed statistical comparison of the model results with the overall average does in my opinion not really add value to the result analysis (this relates particularly to Table 3 and partly to Table 4, it is more adequately visualized in Figure 5!). Firstly, because the number of models is so small. Secondly, and more important, since it is a bit like comparing apple and pears as you say yourself in the interpretation. The large differences in the model results derives from the different types of models. Therefore, I would not list in detail the variation of the models to the average but only describe and explain the differences of those models with similar approaches. The analysis could be done in a more qualitative way since numbers such as average or variation does not make so much sense when the overall number of models is so small. - I consider some of the explanations about calculation findings as being far too long because too obvious when the conclusion is that the differences can be traced back to the different model approaches (for example Lin 332 ff)

Continuation of specific other comments: - Line 453 ff: it is not clear of the percentages refer to the amount of building or the outlier value (I suppose the former but that needs to be clarified) - Line 581: "Consultations of experts with local knowledge can ensure the correct interpretation and use of observed damage data" – I would not agree with that, it may help but does not ensure. … -

Text parts with language issues: - Line 95: "being them unknown" – unclear, pls consider reformulation - Line 442: something went wrong with "in, as."

1. Does the paper address relevant scientific and/or technical questions within the scope of NHESS? yes 2. Does the paper present new data and/or novel concepts, ideas, tools, methods or results? yes 3. Are these up to international standards? yes 4. Are the scientific methods and assumptions valid and outlined clearly? Yes, with some limitations (see above) 5. Are the results sufficient to support the interpretations and the conclusions? Yes, with some limitations (see above) 6. Does the author reach substantial conclusions? yes 7. Is the description of the data used, the methods used, the experiments and calculations made, and the results obtained sufficiently complete and accurate to allow their reproduction by fellow scientists (traceability of results)? yes 8. Does the title clearly and unambiguously reflect the contents of the paper? yes 9. Does the abstract provide a concise, complete and unambiguous summary of the work done and the results obtained? Yes 10. Are the title and the abstract pertinent, and easy to understand to a wide and diversified audience? yes 11. Are mathematical formulae, symbols, abbreviations and units correctly defined and used? If the formulae, symbols or abbreviations are numerous, are there tables or appendixes listing them? yes 12. Is the size, quality and readability of each figure adequate to the type and quantity of data presented? Yes, with some limitations 13. Does the author give proper credit to previous and/or related work, and does he/she indicate clearly his/her own contribution? yes 14. Are the number and quality of the references appropriate? yes 15. Are the references accessible by fellow scientists? yes 16. Is the overall presentation well structured, clear and easy to understand by a wide and general audience?

[Figure]

yes 17. Is the length of the paper adequate, too long or too short? Adequate, some parts are too long 18. Is there any part of the paper (title, abstract, main text, formulae, symbols, figures and their captions, tables, list of references, appendixes) that needs to be clarified, reduced, added, combined, or eliminated? See my detailed comments 19. Is the technical language precise and understandable by fellow scientists? yes 20. Is the English language of good quality, fluent, simple and easy to read and understand by a wide and diversified audience? Yes with some limitations (see my comments) 21. Is the amount and quality of supplementary material (if any) appropriate? N.a.
* * *

---

## Author Comment (AC1) · 20 May 2020

We would like to thank the Reviewers both for their interest in our work and for carefully reading our manuscript; we greatly appreciate the insightful comments as they may contribute to increase the manuscript robustness and, in general, to improve its quality and readability. In the following, we supply a point by point reply to the general and specific comments raised by the Reviewers.

**Reviewer 1:**

R1-C1: There are a few areas where the grammar or writing could be polished although the sense is always clear
Answer: In revising the manuscript we will take care of polishing the grammar and writing.

R1-C2: Figure 1: It would be interesting to see the maximum flood depths plotted on the map, so as to get a sense of how the flood varied across the flood zone.
Answer: We agree, so we will modify Figure 1 by showing the flood depth map for the 2002 event.

R1-C3: Do the authors have any information on the flood map and how closely it matched the flood events that occurred. This is obviously a potentially large source of uncertainty in the results.
Answer: This information is described in the paper quoted in the manuscript (Scorzini et al. 2018), however when revising the manuscript, we will include some additional details on the hydraulic modelling of the event. For the 2002 flood, information on observed water depths was available in more than 260 points within the inundated area, deriving from indications provided by municipal technicians and by citizens in the damage compensation forms, as well as from interpretation of photographs taken during or immediately after the event. These data were used in the validation of the 2D hydraulic model; the resulting average absolute differences between observed and calculated water depths within the inundated area range from 0.2 to 0.4 m, depending on the zone. This is surely a possible source of uncertainty, however, as also known from the literature, resulting differences could be considered to provide relatively small impacts on the overall damage estimation. Moreover, as the same hazard data were used for the implementation of all damage models, this would be a common source of uncertainty for the application of all our models that should not condition the results presented in the paper.

R1-C4: In Table 1, where there is a set of discrete responses, is it possible to see what these choices are (e.g. level of maintenance: Low, medium, high, etc).
Answer: In revising the manuscript, we will include in Table 1 the missing description of the parameter "level of maintenance", by including the three possible choices (low, medium, high).

R1-C5: Table 2 and associated text. Is it possible to give more detail on the adjustments needed to each model to get them to work. This could be due to my misunderstanding, but for example, taking the model of Jonkman et al, two variables are needed (h and FA). However, is the replacement value also needed?
Answer: Yes, in the revised version of the manuscript we will include in Step 2 of the "Methodology" some details on the adjustments needed for the models to work.
We will also make Table 2 more clear by adding the parameter "economic value" (the type of economic evaluation is already shown in the fifth column of Table 2) among the explicative variables for all the damage models (except for CEPRI, which is an absolute damage model).

R1-C6: Figure 3 - could the authors explain why there are buildings with no damage? Are these the buildings discussed in Section 3.4?
Answer: No, they are not. The zero damages are due to the specific assumptions behind the damage models (e.g. 0.25 m water depth threshold for damage occurrence in Arrighi et al.). We will add an explanation for this fact in the revised manuscript.

R1-C7: Line 442 - there is some text missing "given that both datasets show comparable values. in, as"
Answer: Yes, thank you. It was a typo and we will fix it in the revised manuscript.

R1-C8: Line 486 - I find it highly interesting that there is an expectation that many people do not claim through their insurance - this is worthy of a new line of inquiry in its own right (although not within the paper as it is out of scope).
Answer: Thank you for the comment.

R1-C9: Could the authors say more about what is needed to apply models not developed for Italy to work - for example, does the research suggest simple steps that could increase the performance of models developed in the Netherlands or Germany?
Answer: We think this aspect was already addressed in the original manuscript, although probably not stated very explicitly: to apply a damage model to a different country, it is important to verify the comparability between the original and the investigated physical (in terms of hazard and building features) and compensation context, as well as the availability and coherence of the input data. In the revised manuscript, we will include a summarizing statement on this point in the take-home messages reported in the conclusions of the paper.

**Reviewer 2**

R2-C1: The paper is limited by there being only one validation set. This validation set has problems as is correctly mentioned in section 3.4. In earlier similar studies (e.g. Jongman et al., 2012) that used multiple validation studies it was common to see that some models performed well on one validation set and bad on another. I think this limitation of the study should be mentioned a bit more clearly, also to point out for future studies of this kind that it may be a good idea to have multiple validation sets (such as in Jongman et al., 2012). Because of this and the next point, the main value of the paper is in a comparison between the models rather than an absolute value judgement of the models.
Answer: We certainly agree with the Reviewer that it is always desirable to have more validation datasets and that some models can work well in a case and worse in others, as shown in the study of Jongman et al. (2012). Unfortunately, most of the times it is even difficult to have only one dataset, given the well-known paucity of ex-post damage data. In any case, we will clarify the importance of having multiple validation datasets in the revised version of the manuscript (see also reply to comment R2-C5). Regarding the point on absolute value judgement of the models, we would like to stress that the main aim of our paper was not to identify the "best" damage model or to make any kind of "absolute" ranking, but rather to provide potential users of damage models with general considerations on the spatial transferability of the modelling tools and reliability of loss estimates. Besides, we would like to stress the additional innovative aspect of our study, e.g. in comparison with Jongman and others, which is the blind validation test providing more objective insights, than when modelers know the results they are aiming at. These points will be better stressed in the introduction and in the discussion/conclusion sections of the revised version of the manuscript (see also response to comment R2-C2).

R2-C2: The second weakness is also not really highlighted and that is that the number of compared models is too small to draw any strong conclusions. Obviously, it wasn't feasible to select more models but some of the good model performances shown in this paper are pure coincidence, I'm sure about that. A good or bad performance of a model should therefore be seen as a single data point (sample) that could be just be a coincidence. This should be taken into account when drawing conclusions. I think this goes well in the conclusions section but in the results and in the discussion section some of the speculations should be done more cautious. Maybe go through these sections and reevaluate some of the reasoning based on the idea that there is a high level of coincidence in the results. Also be clearer about this limitation and the general idea that some observations could be simply a coincidence, add a few sentences about that somewhere.
Answer: The Reviewer claims that one weakness of our paper is related to the use of a limited number (9) of damage models. It is true that this is not a huge number, however it is line with other studies testing damage models that can be found in the literature (e.g. Jongman et al. (2012) compared 7 models). Most importantly, as also pointed out in the manuscript, we selected only those models that were mastered by the authors, in order to avoid any possible bias in their application (see also response to comment R2-C6).

The second point raised by the Reviewer is that the good/bad performances of the models are due to a coincidence, based on his/her own belief. We based our discussion and conclusions on the results emerging from the empirical analysis carried out in our paper. Some of the outcomes were not surprising and are corroborated by previous studies: for instance, (i) the better performances provided by local models rather than imported ones; (ii) models providing good results have proven to perform well also in other validation case studies for other events in Italy (e.g. Arrighi et al. worked well also for the 2010 flood in Veneto Region (Scorzini and Frank 2017); the same applies to INSYDE and Carisi et al., which were tested in other Italian flood events (Amadio et al. 2019); similar considerations can be made for the model of Dutta et al. which was already found to not properly work in Italian cases (Scorzini and Frank 2017)); (iii) multi-variable models can provide worse performances than simpler ones if they are applied in contexts different from the original one, either in terms of physical features or availability of the input data. However, we are aware, that these results are associated with uncertainties and the general picture might be different when the models would be applied in different case study areas. Thus, we will thoroughly check the results and discussion section to avoid overconfident statements.

In addition, as mentioned in the reply to the previous comment, our paper was not aimed at identifying the "best" damage model, but rather to provide potential users of damage models with general considerations on the spatial transferability of damage models and reliability of loss estimates. This point will be better stressed in the introduction of the revised version of the manuscript.

R2-C3: The statement in the discussion about the value of multi-variable models against simple models (line 526) cannot be stated like this. Multi-variable models can only be transferred when there is some overlap in the context between the training and validation data (see Wagenaar et al. 2008). When there is no overlap the transfer obviously wouldn't work no matter how many variables you add to the data. A multi-variable model should always be compared to a single-variable model based on the same data and not to a single-variable model from a different region to then conclude that a multivariable model isn't useful. Maybe this could be done with a very large number of models but not based on the tiny number tested in this paper (especially because this observation also contradicts common sense and earlier findings elsewhere).

Answer: We fully agree with the Reviewer and our paper corroborates this point, given that the results indicated that multi-variable models applied in contexts different from the original one could perform worse than simple models and this should be considered as a "caveat" for models' users.

R2-C4: The paper title, abstract and introduction puts a lot of emphasis on it being a blind-test. All properly carried out model validations studies don't use any knowledge from the validation data for the model development (standard practice). The constant emphasize on that in this study is therefore a bit misplaced I think. Did earlier studies not follow this approach? I believe they did. Maybe they didn't advertise it this strongly, or maybe this study was more strict or systematic on that but does that really add anything special?

Answer: We agree that all model validation studies keep the validation data separate from the data used for model development, but this is not the point here. Commonly, model validation studies are carried out in a way, that the modellers know the validation data and thus the result they are aiming at. Thus, model applications can be tuned to get as close to the desired result as possible. The adoption of our blind approach prevents any possibility of "tuning" the input variables of the damage models, especially the parameters related to the more qualitative vulnerability features. We will stress this point in the revised version of the manuscript.

R2-C5: This study is very similar to the paper Jongman et al., 2012. It might be interesting to add a paragraph in the discussion section to compare the results of the papers. Of course, this paper is comparing much newer and different models. Yet the general observation that the results are very different from each other and that its difficult to make sense of that is the same among the studies.

Answer: In the revised version of the manuscript we will include some discussion on the mentioned study, by quoting it especially regarding the importance of having multiple validation sets (see also response to comment R2-C2), although we do not fully agree with the Reviewer on the similarity with

the paper of Jongman et al. 2012 (at least for the main objectives). In fact, our study aims at providing potential users with general considerations on the spatial transferability of the modelling tools and reliability of loss estimates, with a specific focus on micro-scale damage models for the residential sectors, while the study of Jongman et al. is focused on strengths and weaknesses in existing modelling approaches (working at different spatial scales and for different exposed sectors) towards the development of a harmonized European approach, which implies an adjustment of modelling tools that was not instead performed in our study, where models have been implemented in their original formulation.

R2-C6: The large group of authors suggests that an expert on each of the compared models was included in the paper. This is however clearly not the case for the Jonkman et al. model and hence the paper makes claims about this model that aren't true. Apart from fixing these mistakes (which I pointed out below), I think it should be clarified somewhere which experts worked on which models and whether the expert personally developed the model or interpreted information from literature (much more error prone).
Answer: The authors were either developers of the models (Arrighi et al., Carisi et al., CEPRI, FLEMO-ps, Insyde) or frequent users with a certain knowledge of the models (Dutta et al., Fuchs et al. and Jonkman et al. are commonly used in Switzerland by the group of authors from the University of Bern). In revising the manuscript, we will include an additional comment, in the introduction section, on the importance of having the contribution in the study of model's developers/experts, as this prevents any possible bias in the results that could arise from an incorrect application of the models (for example, a non-expert may experience a misunderstanding of any of the input variables which would affect the final results).

Minor comments:

R2-C7: Line 46: This is grey literature and I can't find it on the internet. Perhaps add one of the very many peer-reviewed journal articles that could also be used to support this statement and are often much older than 2007.
Answer: We will include the following additional reference in the revised manuscript: Merz, B., Kreibich, H., Thieken, A., & Schmidtke, R. (2004). Estimation uncertainty of direct monetary flood damage to buildings. Natural Hazards and Earth System Sciences, 4: 153-163.

R2-C8: Line 101: I have never seen the term "low-variable" model, maybe consider a different word? Do you mean "single-variable" here?
Answer: We decided to introduce a new term, as we think that "single-variable" is used incorrectly since damage models always consider at least two variables (footprint area and water depth). The meaning of the term will be explained in the revised text.

R2-C9: In section 2.3/table 2. Some basic but important information in this section seems missing for some of the models, such as the origin of the model (country/region) (for Carisi et al. – mono) and the intended flood type (for many of them) and the year the model was made (maybe can be retraced from reference list but not easy for reader). Maybe its good to add this information to table 2 or at least make it clear in the section. Also maybe try to make the different texts a bit more uniform (i.e. present same information in same order).
Answer: Thank you for the suggestion. We will include missing information in the revised version of Table 2.

R2-C10: Line 245. I don't see the significance of this model using a mathematical function. I also know much older models doing that and think its an irrelevant characteristic. Some other damage functions may also follow a mathematical function but just communicate it different.
Answer: The Reviewer is right. Then, we will revise the sentence regarding Dutta et al.'s model by deleting "with a mathematical function".

R2-C11: Table 2: I don't think its correct to classify the Jonkman et al., 2008 model as empirical. It was inspired by many different sources of empirical data but no systematic empirical method was applied and the model is basically expert judgement considering a few empirical data points.

Answer: Thank you very much for the specification. We checked the paper again. According to the paper, the function was developed based on empirical data combined with existing literature and expert judgment as we have written in the manuscript. To make it more clear, we will change the sentence in L279-281 to "The model by Jonkman et al. (2008) it is a simple relative damage model considering water depth, building area (of all floors) and building (replacement) value as explicative variables, developed on the basis of empirical flood damage data of the past in the Netherlands in combination with existing literature and expert judgment." Accordingly, we will also revise Table 2 for the Jonkman et al. model by changing "empirical" with "mixed".

R2-C12: Section 3: I don't like this title very much why not just "results" this is confusing

Answer: Ok, we will change the title of Section 3 with "Results".

R2-C13: Line 293-294: Maybe clarify that you look at a subset of the models here and didn't adjust the whole building models to only use the ground floor value (unclear at first).

Answer: Thank you. We will better clarify this point in the revised version of the manuscript.

R2-C14: Line 298 and 299: Can you rephrase this sentence its currently confusing.

Answer: The Reviewer is right. Also in line with comment R3-C13, we propose to simplify the sentence as follows: "Total damage estimations differ by a maximum factor of 12.6, suggesting that the shape of the damage functions exacerbate the variability of models' outcomes due to exposure estimation".

R2-C15: Line 305: Could you rephrase this sentence, it's difficult to follow: "Individual damage estimates differ on average by a factor of 28, with the more frequent factor around 10."

Answer: Also in line with comment R3-C13, we propose to simplify the sentence as follows: "Individual damage estimates differ on average by a factor of 28".

R2-C16: Table 3: Could you split the 3th column, this is confusing and an uncommon form a presenting it. The title of 3.1 is a bit unclear. Could you rephrase it? It's a bit long and I didn't get that with blind mode you mean that its not compared to observations yet (which I expected). Maybe call it "comparison of the models". Or keep the title and clarify in the first sentence that the reader shouldn't expect the comparison to real observations yet.

Answer: In the revised manuscript we will modify Table 3 as suggested by the Reviewer, while we would like to keep the title of section 3.1, given that the meaning of "blind" has been described earlier in the paper; however, to enhance clarity, in the revised manuscript we will include an introduction statement explaining that the results presented in that section do not yet involve any comparison with observed damage data given that, at the stage described in section 3.1, the models are still applied in a "blind mode".

R2-C17: Figure 4: This table is interesting but requires some additional discussion. When you have two very different exposure values but the variation among the buildings solely depends on the size of the buildings the correlation between the two very different exposure values is still one. So this figure mostly says something about the characteristics that differ per building (if I understand the figure correctly). This doesn't become clear from the text.

Answer: Based on Reviewer's comment, we realized that we missed to describe in detail Figure 4, as it provides information on a building-by-building comparison, so the size of the buildings does not have effects on the results shown in the Figure. We will then clarify this point in the revised manuscript.

R2-C18: Section 3.2. I like the content of this section but I had to read it twice to fully understand it. Maybe the authors could try to clarify this section a bit more. I think especially the title and the first sentence don't make it very clear at first (its all correct but it's a bit of deciphering for a reader)

Answer: Ok, we will try to better clarify this section in the revised manuscript.

R2-C19: Table 4: Could you split the column again like in the previous table.
Answer: This will be fixed in the revised version of the manuscript.

R2-C20: Table 4: Could you consider including the model origin of all models in this table. That would be a very interesting reminder for the reader. Especially for readers who don't read the entire text this would be very useful.
Answer: Thank you for the suggestion. In the revised manuscript we will show in the first column of the Table (in parentheses) the origin of the different models.

R2-C21: Line 487: In case of the Netherlands this isn't true. The empirical data in this model is at best used relative. Absolute values are 100% synthetic.
Answer: Thank you for the clarification. We will not mention the Netherlands in this sentence.

R2-C22: Line 554: Micro-models are essential when measures are undertaken at micro level (e.g. for insurance or studies about elevating specific houses as is common in some countries). When aggregated damages are assessed they may indeed add less information. A second reason why micro-models are important is for at least for the location of buildings. The difference between a house flooding or not is sometimes a matter of just meters. Its therefore important to work with precise models on location even if the other building characteristics are all the same.
Answer: We agree with the Reviewer on the importance of micro-scale models, but, in our opinion, the main point that deserves some discussion is not when and why micro-scale models are useful/important (as this is also well known from the literature), but rather on their actual usefulness if they are not able to provide reliable results.

**Reviewer 3**

R3-C1: In my opinion, a main drawback of the whole study lies in the approach to 'validate' the results. The authors claim to assess the models' reliability through a comparison with observed damages of a real flood event (chapter 3.3) and show respective results (for example Table 4 and 5). The following chap 3.4 is then dedicated to explaining why the models results differ so strongly from the observed results. Crucial reasons for this discrepancy found are then assigned to inconsistencies in the damage claims, that is, the validation data. This seems for me like an odd approach. If the damage claim dataset was meant to be used as a validation set, more emphasis should have been put on clarifying inconsistencies and maybe on further filtering the dataset down to a set of reliable data on damage of a reduced number of buildings
Answer: The reason why we didn't filter the dataset before performing the validation exercise, as the Reviewer is suggesting in his/her comment, is that we decided to apply a real "blind approach" also to the handling of the empirical data, avoiding any "adaptation of the observations to the model". The main reason behind this choice is the intention to underline a common problem we face in the scientific community, concerning the quality of damage data used for validation, and to warn about conclusions that can be derived from validation analyses: for instance, if a model does not fit well some empirical data, this does not mean that it is not a "good" model and vice versa.
Moreover, it is often the case that empirical data are used in validation analyses without any possible preliminary evaluation on their quality and significance, simply because no ancillary information is available. An example of this kind of data is represented by insurance data, which usually lack of useful information to obtain insights on the quality/significance of the damage databases (e.g. Denmark (Zhou et al. 2013), France (André et al. 2013), the Netherlands (Spekkers et al. 2013) and the US (Wing et al. 2020)). Then, the general question we would like to arise is the following: how one can be sure to derive solid conclusions on the results of a validation analysis if no information on the quality of used empirical data is available? We propose to include some comments on this point in the conclusions (after L580) of the revised version of the manuscript.

- *André, C., Monfort, D., Bouzit, M., and Vinchon, C. (2013). Contribution of insurance data to cost assessment of coastal flood damage to residential buildings: insights gained from Johanna (2008) and Xynthia (2010) storm events, Nat. Hazards Earth Syst. Sci., 13, 2003–2012.*

- *Spekkers, M.H., Kok, M., Clemens, F.H.L.R., and Ten Veldhuis, J.A.E. (2013). A statistical analysis of insurance damage claims related to rainfall extremes. Hydrology and Earth System Sciences, 17(3), 913-922.*
- *Wing, O.E., Pinter, N., Bates, P.D., and Kousky, C. (2020). New insights into US flood vulnerability revealed from flood insurance big data. Nature communications, 11(1), 1-10.*
- *Zhou, Q., Panduro, T.E., Thorsen, B.J., and Arnbjerg-Nielsen, K. (2013). Verification of flood damage modelling using insurance data. Water science and technology, 68(2), 425-432.*

R3-C2: The authors only declare having had some informal conversations with experts but the explanations about low damage values observed remain fuzzy. For the approach of this paper a more thorough survey of affected people would have been necessary.

Answer: Unfortunately, as many years have passed since the flood event, it was not possible to get in contact with all of the affected people (many of them have moved out). However, as indicated in the manuscript, we had the opportunity to have conversations with representatives of the Committee of Flooded Citizens in Lodi, who were able to give us descriptive information about occurred damages in large part of the town. Moreover, we want to stress that we did not only had conversations with people, but we also performed an analysis of the different damage components to have more insights on observed data (Section 3.4).

R3-C3: The statement in line 471ff "According to our interpretation, inconsistency between expected and declared damage can be attributed to the fact that what is declared by citizens does not correspond to the actual money required to replace or reconstruct the whole physical damage suffered by the building" cannot satisfy and actually triggers a the more philosophical question, whether the 'damage' targeted by the model represents the damage felt by the people affected.

Answer: We actually wanted to raise a modelling question, rather than a philosophical one. The analysis of the described post-event damage data suggested what is reported in the statement in L471; of course, as also highlighted in the text, it is our interpretation, but it fits well with the obtained results.

R3-C4: I suggest shifting those parts of the chapter 3.4 with the explanation about how the validation dataset was generated to an earlier part of the paper as background information about the approach (where also the various models are described).

Answer: We had long discussions among us on which could be the best choice for the structure of the paper and, finally, we felt that the proposed one was the most coherent with the adopted blind approach, so that the reader is not influenced from this information when reading previous results (see also reply to comments R3-C1 and R3-C9). This said, we would like to keep the mentioned parts of section 3.4. where they were in the original manuscript,

R3-C5: The detailed statistical comparison of the model results with the overall average does in my opinion not really add value to the result analysis (this relates particularly to Table 3 and partly to Table 4, it is more adequately visualized in Figure 5!). Firstly, because the number of models is so small. Secondly, and more important, since it is a bit like comparing apple and pears as you say yourself in the interpretation. The large differences in the model results derives from the different types of models. Therefore, I would not list in detail the variation of the models to the average but only describe and explain the differences of those models with similar approaches. The analysis could be done in a more qualitative way since numbers such as average or variation does not make so much sense when the overall number of models is so small.

Answer: The data reported in the Tables were intended to be used as a result of a sensitivity analysis and not of a detailed statistical analysis. Given that other Reviewers did not complain on this point, we would like to maintain the Tables as they are, better specifying the aim of our analysis so as not to create misunderstandings.

R3-C6: Line 81: "Reality is hardly reproduced by observed data" – I would suggest not to use the term 'reality' since there is no univocal damage value as you prove later on yourself

Answer: This was a provocative statement. Indeed, we also specified that there is not a "univocal" reality and therefore the term should not be used or used with caution. We would like to keep the

sentence as it is, however if the Editor thinks it is too strong, we may consider a softening or the possibility to write "Reality", in quotes.

R3-C7: Line 86f: "comparative studies over a broad range of test cases are essential for acquiring more confidence in the reliability of modelling tools" – after having read your paper I would not say that the test case increased the confidence in reliability, I would put the emphasis more towards understanding in detail how certain model results come about
Answer: This was a general statement in the Introduction section; however, according to Reviewer's suggestion we propose to modify it as follows: "comparative studies over a broad range of test cases are essential for acquiring a thorough understanding of the performances of the modelling tools that could help in enhancing the confidence in their reliability".

R3-C8: Line 101: "the focus of this study lies in this specific set of models" – what does this mean?
Answer: In the revised manuscript we will amend the sentence as follows: "This study focuses on micro-scale (i.e. individual item scale) direct damage assessment to residential buildings, in line with the larger availability of damage modelling approaches developed in Europe for this specific sector.".

R3-C9: Line 149: "was not uniform, as only some of the owners justified costs for fixing the damage by means of invoices" – if not earlier, here the reader should suspect that the quality of the 'validation' dataset is questionable. I would propose to already link here to further explanations about this 'observed' damage data.
Answer: See also reply to comment R3-C1. The case presented in the paper is a "fortunate" one, given that we were able to make some considerations on the quality of the data, based on the availability of ancillary information. In line with the adopted blind approach, we do not agree on anticipating in the presentation of the case study some of the results that come out only after the unblinding of the observed damage data.

R3-C10: Line 186: consider to call it variation rather than 'difference'
Answer: Thank you, we will consider the suggestion in the revised manuscript.

R3-C11: Line 195: unclear for me, until here I thought the observation data was derived from damage claims made after the flood event. How can you use 'official claims' to explain inconsistencies between estimations and observations? Or is the officially claimed data different from the claims mentioned earlier? Then you could have used the official ones for validation?
Answer: We thank the Reviewer for highlighting a possible source of misunderstanding. The same database of damage claims was used throughout the paper. In the revised manuscript, we will delete the word "official" in order to avoid confusion.

R3-C12: Line 207: does only this model use stage-damage curves? Or why is that here mentioned explicitly?
Answer: Clearly it is not the only model using stage-damage curves. To avoid confusion, in the revised manuscript the sentence in L205-208 will be rephrased as: "The model developed by Arrighi et al. (2018a, 2018b) is a relative synthetic model which expresses monetary damage as a function of water depth and recovery cost for buildings with and without basement. A zero-damage threshold is set for a water depth lower than 0.25 m for buildings without basement".

R3-C13: I consider some of the explanations about calculation findings as being far too long because too obvious when the conclusion is that the differences can be traced back to the different model approaches (for example Lin 332 ff)
Answer: We will try to shorten the explanations about calculation findings in the revised version of the manuscript.

R3-C14: Line 453 ff: it is not clear of the percentages refer to the amount of building or the outlier value (I suppose the former but that needs to be clarified)

Answer: This was already made clear in the original manuscript, as it can be noted in L453 where it was stated "examining in detail the outlier claims".

R3-C15: Line 581: "Consultations of experts with local knowledge can ensure the correct interpretation and use of observed damage data" – I would not agree with that, it may help but does not ensure. –
Answer: The Reviewer is right and then in the revised manuscript we will better specify that "Consultations of experts with local knowledge can help in the correct interpretation and use of observed damage data"

R3-C16: Line 95: "being them unknown" – unclear, pls consider reformulation
Answer: We will amend this sentence in the revised manuscript by changing the word "unknown" with "undisclosed".

R3-C17: Line 442: something went wrong with "in, as."
Answer: Yes, thank you. It was a typo and we will fix it in the revised manuscript.

---

## Author Response (AR1)

We would like to thank the Reviewers both for their interest in our work and for carefully reading our manuscript; we greatly appreciate the insightful comments as they may contribute to increase the manuscript robustness and, in general, to improve its quality and readability. In the following, we supply a point by point reply to the general and specific comments raised by the Reviewers.

**Reviewer 1:**

R1-C1: There are a few areas where the grammar or writing could be polished although the sense is always clear
Answer: In revising the manuscript we took care of polishing the grammar and writing.

R1-C2: Figure 1: It would be interesting to see the maximum flood depths plotted on the map, so as to get a sense of how the flood varied across the flood zone.
Answer: We agree, so we have modified Figure 1 by showing the flood depth map for the 2002 event.

R1-C3: Do the authors have any information on the flood map and how closely it matched the flood events that occurred. This is obviously a potentially large source of uncertainty in the results.
Answer: This information is described in the paper quoted in the manuscript (Scorzini et al. 2018); however, in the revised manuscript, we have included the following details on the hydraulic modelling of the event (L138): "*In detail, with respect to the hazard, information on observed water depths was available for more than 260 points within the inundated area, deriving from indications provided by municipal technicians and by citizens in damage compensation requests, as well as from interpretation of photographs taken during or immediately after the flood. These data were used for the validation of the 2D hydraulic simulation of the event: the resulting average absolute differences between observed and calculated water depths within the inundated area ranged from 0.2 to 0.4 m, depending on the validation zone in which observed water depth data were aggregated (Scorzini et al., 2018). This is surely a possible source of uncertainty; however, reported differences could be considered to provide relatively small impacts on the damage estimation. Moreover, given that all tested damage model shared the same hazard data, this would be a common source of uncertainty that should not affect the overall results of the blind test*".

R1-C4: In Table 1, where there is a set of discrete responses, is it possible to see what these choices are (e.g. level of maintenance: Low, medium, high, etc).
Answer: In the revised version of the manuscript, we have included in Table 1 the missing description of the parameter "level of maintenance", by including the three possible choices (low, medium, high).

R1-C5: Table 2 and associated text. Is it possible to give more detail on the adjustments needed to each model to get them to work. This could be due to my misunderstanding, but for example, taking the model of Jonkman et al, two variables are needed (h and FA). However, is the replacement value also needed?
Answer: The point on the adjustments needed for the models to work was already described in Step 2 of the "Methodology" (P7. L170-173), but in the revised version of the manuscript we have better exemplified this part by including the following text in the discussion (L 578): "*For example, the building categories (BT) assumed by CEPRI ("apartment"/ "single storey building"/ "multi-storeys buildings") are different than the Italian ones ("apartment"/ "detached"/ "semi-detached") so that a correspondence has to be defined, also on the bases of the number of floors (NF); specifically "apartment" is defined by BT ="apartment", "single-storey" is defined by BT = "detached" or "semi-detached" and NF = 1, "multiple-storeys" is defined by BT ="detached" or "semi-detached" and NF > 1. Correspondence among building categories was defined also for the implementation of FLEMO-ps, although in this case the task was quite straightforward, since the German building categories are almost coincident with the Italian ones (FLEMO-ps distinguishes between "Multi-family house" / "Semi-detached house" / "One-family home")*.
Moreover, we have made Table 2 more clear by adding the parameter "economic value" (the type of economic evaluation is already shown in the fifth column of Table 2) among the explicative variables for all the damage models (except for CEPRI and INSYDE, which are absolute damage models).

R1-C6: Figure 3 - could the authors explain why there are buildings with no damage? Are these the buildings discussed in Section 3.4?

Answer: No, they are not. The zero damages are due to the specific assumptions behind the damage models (e.g. 0.25 m water depth threshold for damage occurrence in Arrighi et al. and 0.01 m water depth threshold in Dutta et al. and Jonkman et al. to distinguish between flooding and surface water runoff). In the revised manuscript, we have included an explanation for these results in the caption of Figure 3.

R1-C7: Line 442 - there is some text missing "given that both datasets show comparable values. in, as"

Answer: Yes, thank you. It was a typo and we have fixed it in the revised manuscript.

R1-C8: Line 486 - I find it highly interesting that there is an expectation that many people do not claim through their insurance - this is worthy of a new line of inquiry in its own right (although not within the paper as it is out of scope).

Answer: Thank you for the comment.

R1-C9: Could the authors say more about what is needed to apply models not developed for Italy to work - for example, does the research suggest simple steps that could increase the performance of models developed in the Netherlands or Germany?

Answer: We think this aspect was already addressed in the original manuscript, although probably not stated very explicitly. In the revised manuscript we have included the following summarizing statement in the take-home messages reported in the conclusions of the paper (L650): "*As a general recommendation, to select a damage model for an application in a different country, it is important to verify the comparability between the original and the investigated physical (in terms of hazard and building features) and compensation context, as well as the availability and coherence of the input data*".

**Reviewer 2**

R2-C1: The paper is limited by there being only one validation set. This validation set has problems as is correctly mentioned in section 3.4. In earlier similar studies (e.g. Jongman et al., 2012) that used multiple validation studies it was common to see that some models performed well on one validation set and bad on another. I think this limitation of the study should be mentioned a bit more clearly, also to point out for future studies of this kind that it may be a good idea to have multiple validation sets (such as in Jongman et al., 2012). Because of this and the next point, the main value of the paper is in a comparison between the models rather than an absolute value judgement of the models.

Answer: We certainly agree with the Reviewer that it is always desirable to have more validation datasets and that some models can work well in a case and worse in others, as shown in the study of Jongman et al. (2012); we added a sentence in the conclusions highlighting this aspect. Unfortunately, most of the times it is even difficult to have only one dataset, given the well-known paucity of ex-post damage data. In any case, we clarified the importance of having multiple validation datasets in the introduction and in the discussion sections of the new version of the manuscript (see also reply to comment R2-C5). Regarding the point on absolute value judgement of the models, we would like to stress that the main aim of our paper was not to identify the "best" damage model or to make any kind of "absolute" ranking, but rather to provide potential users of damage models with general considerations on the spatial transferability of the modelling tools and reliability of loss estimates. Besides, we would like to stress the additional innovative aspect of our study, e.g. in comparison with Jongman and others, which is the blind validation test providing more objective insights, than when modellers know the results they are aiming at. Although we think that these points were already explicated in the first version of the manuscript, we further emphasized them in the introduction of the revised version (see also response to comment R2-C2).

R2-C2: The second weakness is also not really highlighted and that is that the number of compared models is too small to draw any strong conclusions. Obviously, it wasn't feasible to select more models but some of the good model performances shown in this paper are pure coincidence, I'm sure about that. A good or bad performance of a model should therefore be seen as a single data point (sample) that could be just be a coincidence. This should be taken into account when drawing conclusions. I think this goes well in the conclusions section but in the results and in the discussion section some of the speculations should be done more cautious. Maybe go through these sections and reevaluate some of the reasoning based on the idea that there is a high level of coincidence in the results. Also be clearer about this limitation and the general idea that some observations could be simply a coincidence, add a few sentences about that somewhere.

Answer: The Reviewer claims that one weakness of our paper is related to the use of a limited number (9) of damage models. It is true that this is not a huge number, however it is line with other studies testing damage models that can be found in the literature (e.g. Jongman et al. (2012) compared 7 models). Most importantly, as pointed out in the original version of the manuscript and better explained in the new version, we selected only those models that were mastered by the authors, in order to avoid any possible bias in their application (see also response to comment R2-C6) (L91): "*Tested models have been chosen among those mastered by the authors; indeed, the authors were either developers of the models or experienced users with significant knowledge of them, in order to prevent any possible bias in the results that could arise from an incorrect application of the models (for example, a non-expert user may misunderstand the meaning of some input variables, which would affect the final estimation)*".

The second point raised by the Reviewer is that the good/bad performances of the models are due to a coincidence, based on his/her own belief. We based our discussion and conclusions on the results emerging from the empirical analysis carried out in our paper. Some of the outcomes were not surprising and are corroborated by previous studies, as now better explained in the revised version of the paper (see discussion): for instance, (i) the better performances provided by local models rather than imported ones; (ii) models providing good results have proven to perform well also in other validation case studies for other events in Italy (e.g. Arrighi et al. worked well also for the 2010 flood in Veneto Region (Scorzini and Frank 2017); the same applies to INSYDE and Carisi et al., which were tested in other Italian flood events (Amadio et al. 2019); similar considerations can be made for the model of Dutta et al. which was already found to not properly work in Italian cases (Scorzini and Frank 2017)); (iii) multi-variable models can provide worse performances than simpler ones if they are applied in contexts different from the original one, either in terms of physical features or availability of the input data. However, we are aware, that these results are associated with uncertainties and the general picture might be different when the models would be applied in different case study areas. From another point of view, results were critically analysed, by considering both the features of the models and the context of investigation, and never taken from granted, despite their agreement with our expectation. For example, in section 3.3 we comment: "*The table also indicates the better performances of the Italian/local models with Arrighi et al. showing the lowest difference. However, by looking at its features, it is possible to state that even this last model tends to overestimate damage. First, because it does not consider clean-up costs (like INSYDE and CEPRI), which are instead included in the observations. Second, because the lower value of the total damage with respect to other models is partly due to the effect of the zero damage threshold for water depths lower than 0.25 m (see Sect. 3.1)*"; in section 3.4 we comment: "*inconsistency between expected and declared damage can be attributed to the fact that what is declared by citizens does not correspond to the actual money required to replace or reconstruct the whole physical damage suffered by the building (…) This would explain why synthetic models overestimate observed damage, as they are usually based on full replacement/reconstruction costs. Likewise, it would explain why the model by Arrighi et al. performs better than others: indeed, the recovery value adopted by this model is defined as the average difference between the market value of new buildings and that of equivalent, older buildings requiring renovation. It is then sensible that this value reflects a balance between the two opposite extreme behaviours of buyers (which, in turn, depend on their financial resources): i.e. to completely renovate the building or to bring the building back to a minimum level of functioning. In our view, such behaviours can be compared with those of flooded owners*" and, again "*Moreover, declared monetary damage is strongly correlated to the expectations that citizens have to be reimbursed. This expectation is low in Italy (…). This would also explain why*

*empirical models (derived from claims) developed in regions with high expectations and then high values of declared damage (like Germany or the Netherlands), overestimate the observed damage in this case study"*. Thus, after checking the paper again we think that no overconfident statements were included.

In addition, as mentioned in the reply to the previous comment, our paper was not aimed at identifying the "best" damage model, but rather to provide potential users of damage models with general considerations on the spatial transferability of damage models and reliability of loss estimates. This have been further stressed in the introduction of the revised version of the manuscript.

R2-C3: The statement in the discussion about the value of multi-variable models against simple models (line 526) cannot be stated like this. Multi-variable models can only be transferred when there is some overlap in the context between the training and validation data (see Wagenaar et al. 2008). When there is no overlap the transfer obviously wouldn't work no matter how many variables you add to the data. A multi-variable model should always be compared to a single-variable model based on the same data and not to a single-variable model from a different region to then conclude that a multivariable model isn't useful. Maybe this could be done with a very large number of models but not based on the tiny number tested in this paper (especially because this observation also contradicts common sense and earlier findings elsewhere).

Answer: We fully agree with the Reviewer and our paper corroborates this point, given that the results indicated that multi-variable models applied in contexts different from the original one could perform worse than simple models and this should be considered as a "caveat" for models' users.

R2-C4: The paper title, abstract and introduction puts a lot of emphasis on it being a blind-test. All properly carried out model validations studies don't use any knowledge from the validation data for the model development (standard practice). The constant emphasize on that in this study is therefore a bit misplaced I think. Did earlier studies not follow this approach? I believe they did. Maybe they didn't advertise it this strongly, or maybe this study was more strict or systematic on that but does that really add anything special?

Answer: We agree that all model validation studies keep the validation data separate from the data used for model development, but this is not the point here. Commonly, model validation studies are carried out in a way, that the modellers know the validation data and thus the result they are aiming at. Thus, model applications can be tuned to get as close to the desired result as possible. The adoption of our blind approach prevents any possibility of "tuning" the input variables of the damage models, especially the parameters related to the more qualitative vulnerability features. We stressed this point in the revised version of the manuscript (see introduction, L101): "*possible biases are avoided as participants cannot be influenced by validation data, being them undisclosed in the implementation phase of the models, e.g. by trying to adjust or tune their models, especially regarding the more qualitative input parameters, in light of observed damages"*.

R2-C5: This study is very similar to the paper Jongman et al., 2012. It might be interesting to add a paragraph in the discussion section to compare the results of the papers. Of course, this paper is comparing much newer and different models. Yet the general observation that the results are very different from each other and that its difficult to make sense of that is the same among the studies.

Answer: In the conclusion of the revised version of the manuscript we added a discussion on the mentioned study, by quoting it especially regarding the importance of having multiple validation sets (see also response to comment R2-C2), although we do not fully agree with the Reviewer on the similarity with the paper of Jongman et al. 2012 (at least for the main objectives). In fact, our study aims at providing potential users with general considerations on the spatial transferability of the modelling tools and reliability of loss estimates, with a specific focus on micro-scale damage models for the residential sectors, while the study of Jongman et al. is focused on strengths and weaknesses in existing modelling approaches (working at different spatial scales and for different exposed sectors) towards the development of a harmonized European approach, which implies an adjustment of modelling tools that was not instead performed in our study, where models have been implemented in their original formulation.

R2-C6: The large group of authors suggests that an expert on each of the compared models was included in the paper. This is however clearly not the case for the Jonkman et al. model and hence the paper makes claims about this model that aren't true. Apart from fixing these mistakes (which I pointed out below), I think it should be clarified somewhere which experts worked on which models and whether the expert personally developed the model or interpreted information from literature (much more error prone).

Answer: The authors were either developers of the models (Arrighi et al., Carisi et al., CEPRI, FLEMO-ps, Insyde) or experienced users with significant knowledge of the models (Dutta et al., Fuchs et al. and Jonkman et al. are commonly used in Switzerland by the group of authors from the University of Bern). In revising the manuscript, we included an additional comment, in the introduction section, on the importance of having the contribution in the study of model's developers/experts, as this prevents any possible bias in the results that could arise from an incorrect application of the models (for example, a non-expert may experience a misunderstanding of any of the input variables which would affect the final results).

Minor comments:

R2-C7: Line 46: This is grey literature and I can't find it on the internet. Perhaps add one of the very many peer-reviewed journal articles that could also be used to support this statement and are often much older than 2007.

Answer: We included the following additional reference in the revised manuscript: Merz, B., Kreibich, H., Thieken, A., & Schmidtke, R. (2004). Estimation uncertainty of direct monetary flood damage to buildings. Natural Hazards and Earth System Sciences, 4: 153-163.

R2-C8: Line 101: I have never seen the term "low-variable" model, maybe consider a different word? Do you mean "single-variable" here?

Answer: We decided to introduce a new term, as we think that "single-variable" is used incorrectly since damage models always consider at least two variables (footprint area and water depth). The meaning of the term is explained in the revised text (see introduction, L57).

R2-C9: In section 2.3/table 2. Some basic but important information in this section seems missing for some of the models, such as the origin of the model (country/region) (for Carisi et al. – mono) and the intended flood type (for many of them) and the year the model was made (maybe can be retraced from reference list but not easy for reader). Maybe its good to add this information to table 2 or at least make it clear in the section. Also maybe try to make the different texts a bit more uniform (i.e. present same information in same order).

Answer: Thank you for the suggestion. We included missing information in the revised version of Table 2.

R2-C10: Line 245. I don't see the significance of this model using a mathematical function. I also know much older models doing that and think its an irrelevant characteristic. Some other damage functions may also follow a mathematical function but just communicate it different.

Answer: The Reviewer is right. Then, we revised the sentence regarding Dutta et al.'s model by deleting "with a mathematical function".

R2-C11: Table 2: I don't think its correct to classify the Jonkman et al., 2008 model as empirical. It was inspired by many different sources of empirical data but no systematic empirical method was applied and the model is basically expert judgement considering a few empirical data points.

Answer: Thank you very much for the specification. We checked the paper again. According to the paper, the function was developed based on empirical data combined with existing literature and expert judgment as we have written in the manuscript. To make it more clear, we changed the sentence to "*The model by Jonkman et al. (2008) it is a simple relative damage model considering water depthand building (replacement) value of all floors as explicative variables, developed on the basis of empirical flood damage data of the past in the Netherlands in combination with existing*

*literature and expert judgment.*" Accordingly, we also revised Table 2 for the Jonkman et al. model by changing "empirical" with "mixed".

R2-C12: Section 3: I don't like this title very much why not just "results" this is confusing
Answer: Ok, we changed the title of Section 3 with "Results".

R2-C13: Line 293-294: Maybe clarify that you look at a subset of the models here and didn't adjust the whole building models to only use the ground floor value (unclear at first).
Answer: Thank you. We better clarified this point in the revised version of the manuscript.

R2-C14: Line 298 and 299: Can you rephrase this sentence its currently confusing.
Answer: In the revised version of the manuscript, the sentence was rephrased as following (L309): *"Total damage estimations differ among the modelling approaches by a maximum factor of 12.6, which is limited to 3.1 with respect to the mean value of total damage estimations, suggesting that the shape of the damage functions exacerbate the variability of models' outcomes due to exposure estimation".*

R2-C15: Line 305: Could you rephrase this sentence, it's difficult to follow: "Individual damage estimates differ on average by a factor of 28, with the more frequent factor around 10."
Answer: we simplified the sentence as follows (L315): "*Individual damage estimates differ on average by a factor of 28*".

R2-C16: Table 3: Could you split the 3th column, this is confusing and an uncommon form a presenting it. The title of 3.1 is a bit unclear. Could you rephrase it? It's a bit long and I didn't get that with blind mode you mean that its not compared to observations yet (which I expected). Maybe call it "comparison of the models". Or keep the title and clarify in the first sentence that the reader shouldn't expect the comparison to real observations yet.
Answer: In the revised manuscript we modified Table 3 as suggested by the Reviewer, while we preferred to keep the title of section 3.1, given that the meaning of "blind" has been described earlier in the paper; however, to enhance clarity, in the revised manuscript we included a statement explaining that the results presented in that section do not yet involve any comparison with observed damage data given that, at the stage described in section 3.1, the models are still applied in a "blind mode".

R2-C17: Figure 4: This table is interesting but requires some additional discussion. When you have two very different exposure values but the variation among the buildings solely depends on the size of the buildings the correlation between the two very different exposure values is still one. So this figure mostly says something about the characteristics that differ per building (if I understand the figure correctly). This doesn't become clear from the text.
Answer: Based on Reviewer's comment, we realized that we missed to describe in detail Figure 4, as it provides information on a building-by-building comparison, so the size of the buildings does not have effects on the results shown in the Figure. We clarified this point in the revised manuscript (L338): "*Figures 2 and 3 further highlight a common trend in exposure and damage values supplied by the different models, also confirmed in Fig. 4 and 5, showing the Pearson's correlation coefficients for individual (i.e. building by building) exposure and damage estimates. (…)*"

R2-C18: Section 3.2. I like the content of this section but I had to read it twice to fully understand it. Maybe the authors could try to clarify this section a bit more. I think especially the title and the first sentence don't make it very clear at first (its all correct but it's a bit of deciphering for a reader)
Answer: We tried to better clarify this section in the revised manuscript, by amending the title (now as "*Role of input variables in the determination of divergent models' outcomes*") and the first sentences of the section, as follow (L382): "*In order to explain the differences observed in the blind implementation, models were compared in terms of trends and variance of individual damage estimates, for classes of values of input variables, and by considering one variable at a time. The objectives of the analyses were to investigate whether the consideration of a specific input variable*

*influences the outcome of a model with respect to the other ones, whether the inclusion of more explicative variables may be considered as a possible source of variation, and to identify the most influencing parameters on the final output of the models. (…)"*

R2-C19: Table 4: Could you split the column again like in the previous table.
Answer: This has been fixed in the revised version of the manuscript.

R2-C20: Table 4: Could you consider including the model origin of all models in this table. That would be a very interesting reminder for the reader. Especially for readers who don't read the entire text this would be very useful.
Answer: Thank you for the suggestion. In the revised manuscript we show in the first column of the Table (in parentheses) the origin of the different models. We did the same also for Table 5.

R2-C21: Line 487: In case of the Netherlands this isn't true. The empirical data in this model is at best used relative. Absolute values are 100% synthetic.
Answer: Thank you for the clarification. In the revised version of the paper, we do not mention the Netherlands in this sentence

R2-C22: Line 554: Micro-models are essential when measures are undertaken at micro level (e.g. for insurance or studies about elevating specific houses as is common in some countries). When aggregated damages are assessed they may indeed add less information. A second reason why micro-models are important is for at least for the location of buildings. The difference between a house flooding or not is sometimes a matter of just meters. Its therefore important to work with precise models on location even if the other building characteristics are all the same.
Answer: We agree with the Reviewer on the importance of micro-scale models, but, in our opinion, the main point that deserves some discussion is not when and why micro-scale models are useful/important (as this is also well known from the literature), but rather on their actual usefulness if they are not able to provide reliable results.

**Reviewer 3**

R3-C1: In my opinion, a main drawback of the whole study lies in the approach to 'validate' the results. The authors claim to assess the models' reliability through a comparison with observed damages of a real flood event (chapter 3.3) and show respective results (for example Table 4 and 5). The following chap 3.4 is then dedicated to explaining why the models results differ so strongly from the observed results. Crucial reasons for this discrepancy found are then assigned to inconsistencies in the damage claims, that is, the validation data. This seems for me like an odd approach. If the damage claim dataset was meant to be used as a validation set, more emphasis should have been put on clarifying inconsistencies and maybe on further filtering the dataset down to a set of reliable data on damage of a reduced number of buildings
Answer: The reason why we didn't filter the dataset before performing the validation exercise, as the Reviewer is suggesting in his/her comment, is that we decided to apply a real "blind approach" also to the handling of the empirical data, avoiding any "adaptation of the observations to the model". The main reason behind this choice is the intention to underline a common problem we face in the scientific community, concerning the quality of damage data used for validation, and to warn about conclusions that can be derived from validation analyses: for instance, if a model does not fit well some empirical data, this does not mean that it is not a "good" model and vice versa.
Moreover, it is often the case that empirical data are used in validation analyses without any possible preliminary evaluation on their quality and significance, simply because no ancillary information is available. An example of this kind of data is represented by insurance data, which usually lack of useful information to obtain insights on the quality/significance of the damage databases (e.g. Denmark (Zhou et al. 2013), France (André et al. 2013), the Netherlands (Spekkers et al. 2013) and the US (Wing et al. 2020)). Then, the general question we would like to arise is the following: how one can be sure to derive solid conclusions on the results of a validation analysis if no information on the quality of used empirical data is available?

We have then included the following text in the conclusions of the revised version of the manuscript (L632): "*Indeed, it is often the case that empirical data are used in validation analyses without any possible preliminary evaluation on their quality and significance, simply because no ancillary information is available, as for instance for insurance data (André et al. 2013; Spekkers et al. 2013; Zhou et al. 2013; Wing et al. 2020). In this context, the blind test highlighted that "reality" depicted by observations is not univocal, so that data must be carefully investigated before their comparison with model outcomes, as they may be addressing different types of damage, damage to different components, or being incomplete. Based on this consideration, there is a need to be always cautious when drawing conclusions from validation analyses, given that if a model does not fit well some empirical data, this does not necessarily mean that it is not a "good" model and vice versa*".

R3-C2: The authors only declare having had some informal conversations with experts but the explanations about low damage values observed remain fuzzy. For the approach of this paper a more thorough survey of affected people would have been necessary.
Answer: Unfortunately, as many years have passed since the flood event, it was not possible to get in contact with all of the affected people (many of them have moved out). However, as indicated in the manuscript, we had the opportunity to have conversations with representatives of the Committee of Flooded Citizens in Lodi, who were able to give us descriptive information about occurred damages in large part of the town. Moreover, we want to stress that we did not only had conversations with people, but we also performed an analysis of the different damage components to have more insights on observed data (Section 3.4).

R3-C3: The statement in line 471ff "According to our interpretation, inconsistency between expected and declared damage can be attributed to the fact that what is declared by citizens does not correspond to the actual money required to replace or reconstruct the whole physical damage suffered by the building" cannot satisfy and actually triggers a the more philosophical question, whether the 'damage' targeted by the model represents the damage felt by the people affected.
Answer: We actually wanted to raise a modelling question, rather than a philosophical one. The analysis of the described post-event damage data suggested what is reported in the statement in L471; of course, as also highlighted in the text, it is our interpretation, but it fits well with the obtained results.

R3-C4: I suggest shifting those parts of the chapter 3.4 with the explanation about how the validation dataset was generated to an earlier part of the paper as background information about the approach (where also the various models are described).
Answer: We had long discussions among us on which could be the best choice for the structure of the paper and, finally, we felt that the proposed one was the most coherent with the adopted blind approach, so that the reader is not influenced from this information when reading previous results (see also reply to comments R3-C1 and R3-C9). This said, we have kept the mentioned parts of section 3.4. where they were in the original manuscript.

R3-C5: The detailed statistical comparison of the model results with the overall average does in my opinion not really add value to the result analysis (this relates particularly to Table 3 and partly to Table 4, it is more adequately visualized in Figure 5!). Firstly, because the number of models is so small. Secondly, and more important, since it is a bit like comparing apple and pears as you say yourself in the interpretation. The large differences in the model results derives from the different types of models. Therefore, I would not list in detail the variation of the models to the average but only describe and explain the differences of those models with similar approaches. The analysis could be done in a more qualitative way since numbers such as average or variation does not make so much sense when the overall number of models is so small.
Answer: The data reported in the Tables were intended to be used as a result of a sensitivity analysis and not of a detailed statistical analysis. Given that other Reviewers did not complain on this point, we would like to maintain the Tables as they are, better explaining the aim of our analysis so as not to create misunderstandings. Therefore, when presenting Table 3 in the revised manuscript we have clarified that (L296): "*With the aim of understanding the impact of exposure estimation on damage assessment and identifying possible common features in the results, Table 3 shows the total*

*exposure and loss figures obtained by applying the nine models to all buildings within the simulated inundation area (877 in total; see Figure 1)"*. The same was done when introducing Table 4 (L430): "*Table 4 summarises the results of the sensitivity analysis by comparing the total observed damage to the total damage estimates obtained with the implementation of the nine models to the subset of buildings"*.

R3-C6: Line 81: "Reality is hardly reproduced by observed data" – I would suggest not to use the term 'reality' since there is no univocal damage value as you prove later on yourself
Answer: This was a provocative statement. Indeed, we also specified that there is not a "univocal" reality and therefore the term should not be used or used with caution. We preferred to keep the sentence as it was, however we have written "Reality" in quotes in order to highlight its "non-uniqueness".

R3-C7: Line 86f: "comparative studies over a broad range of test cases are essential for acquiring more confidence in the reliability of modelling tools" – after having read your paper I would not say that the test case increased the confidence in reliability, I would put the emphasis more towards understanding in detail how certain model results come about
Answer: This was a general statement in the Introduction section; however, according to Reviewer's suggestion we have modified it as follows: "*comparative studies over a broad range of test cases are essential for acquiring a thorough understanding of the performances of the modelling tools that could help in enhancing the confidence in their reliability*".

R3-C8: Line 101: "the focus of this study lies in this specific set of models" – what does this mean?
Answer: In the revised manuscript we have amended the sentence as follows: "*This study focuses on micro-scale (i.e. individual item scale) direct damage assessment to residential buildings, in line with the larger availability of damage modelling approaches developed in Europe for this specific sector*".

R3-C9: Line 149: "was not uniform, as only some of the owners justified costs for fixing the damage by means of invoices" – if not earlier, here the reader should suspect that the quality of the 'validation' dataset is questionable. I would propose to already link here to further explanations about this 'observed' damage data.
Answer: See also reply to comment R3-C1. The case presented in the paper is a "fortunate" one, given that we were able to make some considerations on the quality of the data, based on the availability of ancillary information. In line with the adopted blind approach, we do not agree on anticipating in the presentation of the case study some of the results that come out only after the unblinding of the observed damage data.

R3-C10: Line 186: consider to call it variation rather than 'difference'
Answer: Thank you. In the revised manuscript we have amended the text accordingly.

R3-C11: Line 195: unclear for me, until here I thought the observation data was derived from damage claims made after the flood event. How can you use 'official claims' to explain inconsistencies between estimations and observations? Or is the officially claimed data different from the claims mentioned earlier? Then you could have used the official ones for validation?
Answer: We thank the Reviewer for highlighting a possible source of misunderstanding. The same database of damage claims was used throughout the paper. In the revised manuscript, we have deleted the word "official" in order to avoid confusion.

R3-C12: Line 207: does only this model use stage-damage curves? Or why is that here mentioned explicitly?
Answer: Clearly, it is not the only model using stage-damage curves. To avoid confusion, in the revised manuscript the original sentence in L205-208 has been rephrased as: "*The model developed by Arrighi et al. (2018a, 2018b) is a relative synthetic model which expresses monetary damage as*

*a function of water depth and recovery cost for buildings with and without basement. A zero-damage threshold is set for a water depth lower than 0.25 m for buildings without basement*'.

R3-C13: I consider some of the explanations about calculation findings as being far too long because too obvious when the conclusion is that the differences can be traced back to the different model approaches (for example Lin 332 ff)
Answer: We think that explanations on calculation findings, although obviously linked to modelling approaches, are fundamental to justify results, especially to non-expert readers. So we preferred to keep the text as it is.

R3-C14: Line 453 ff: it is not clear of the percentages refer to the amount of building or the outlier value (I suppose the former but that needs to be clarified)
Answer: This was already made clear in the original manuscript, as it can be noted in L453 where it was stated "examining in detail the outlier claims".

R3-C15: Line 581: "Consultations of experts with local knowledge can ensure the correct interpretation and use of observed damage data" – I would not agree with that, it may help but does not ensure. –
Answer: The Reviewer is right and then in the revised manuscript we have better specified that "*Consultations of experts with local knowledge can help in the correct interpretation and use of observed damage data*"

R3-C16: Line 95: "being them unknown" – unclear, pls consider reformulation
Answer: We have amended this sentence in the revised manuscript by changing the word "unknown" with "undisclosed".

R3-C17: Line 442: something went wrong with "in, as."
Answer: Yes, thank you. It was a typo and we have fixed it in the revised manuscript.

[revised manuscript text omitted]
, thea 2D hydraulic modelling of the event was availableobtained from a previous study (Scorzini et al., 2018). In detail, , for the 2002 floodwith respect to the hazard,, information on observed water depths was available for more than 260 points within the inundated area, deriving from indications provided by municipal technicians and by citizens in damage compensation requests, as well as from interpretation of photographs taken during or immediately after the flood. These data were used for the

155 validation of the 2D hydraulic simulation of the event: the resulting average absolute differences between observed and calculated water depths within the inundated area ranged from 0.2 to 0.4 m–, depending on the validation zone in which observed water depth data were aggregated (Scorzini et al., 2018). This is surely a possible source of uncertainty; however, reported differences could be considered to provide relatively small impacts on the damage estimation. Moreover, given that all tested damage model shared the same hazard data, this would be a common source of uncertainty that should not affect the

160 overall results of the blind test.
as well asAvailable micro-scale information data on exposure and vulnerability of residential buildings (are instead shown see in Table 1). NonethelessAltogether, observed damage was known for 345 of the 877 buildings in the flooded area (after hydraulic simulation; Fig. 1), as derived from claims compiled by citizens after the occurrence of the flood, to ask for public compensation.

165

[Figure]

[Figure]

**Legend**

- Flooded area (Adda River 2002)
- Buildings with damage claims
- Buildings without damage claims

Adda River

0   250   500        1000
                        m

[revised manuscript text omitted]

---

## Author Response (AR2)

We would like to thank the Reviewers and the Editor for their work and their positive evaluation of our research; we greatly appreciate the comments as they contributed to increase the manuscript's robustness and to improve its quality and readability. In the following, we supply a point by point reply to the last comments raised by one of the Reviewers.

**Reviewer 1:**

R1-C1: As from figure 8, table 5 and comments in the text, it is clear that all the observed and estimated damage are not correlated. This fact is very interesting and deserve to be investigated in more detail. How these differences are located in space? Is there any spatial pattern of model overestimation and, are these patterns dependent on topographic/hydraulic parameters?
Answer: We agree with the referee that the investigation of spatial patterns of (un)correlations and the identification of their possible causes (hazard or vulnerability related) would be interesting, but (i) it is beyond the objective of this paper and (ii) would require an ad-hoc paper to be exhaustive. However, we agree on the importance of underlining the role of hydraulic parameters in explaining differences. To this aim we added Figure S1 in the Supplementary material, showing the ratios between estimated and observed damage at the building scale for different flood depth classes. The figure reveals that the identified overestimations do not depend on the hydraulic features in the inundated area but from the damage models. Accordingly, a sentence has been added at pg. 20 line 345: "Figure S1 in the Supplementary Material, displaying the ratios between estimated and observed damage at the building scale for different flood depth classes, suggests that detected differences do not depend on the hydraulic features in the inundated area but mainly on the damage models, for which the individual differences are similar across all flood depth classes".

R1-C2: The "transferability" of a model is recalled in different points of the text. However, the study does not give a real support to the transferability because only a single case study is analyzed. Instead, "internal transferability" could be useful and should be discussed in more detail: I would have expected that the models were also calibrated (or parameterized on the basis of a blinded dataset) only on a subset of the 345 known building and then validated on the remaining complementary subset. Results from this application could then be used to "transfer" the application to the set of (877-345) "ungauged" buildings where no damage data is available. Do the authors have tried this option?
Answer: Despite only one case study is analysed in the paper, we think that results suggest some important and partly new findings concerning models transferability; in particular, as reported in discussion/conclusions the research suggests:

- that model transferability depends on the consistency between the context of implementation and the original calibration context, as far as both hazard and exposure/vulnerability features of exposed buildings are concerned, corroborating previous findings. In fact, in the blind test, models developed for the Italian territory and for riverine floods performed generally better than models derived in other countries or for different flooding features;
- that transferability could depend also on comparability of the compensation contexts;
- assumptions on input variables may reduce the reliability of the original model because of an improper/inaccurate "adaptation" of the available data, thus reducing the advantage of using many variables;
- assessing exposure coherently with the approach originally adopted in model development is key to preserve its original reliability;
- "reality" depicted by observations is not univocal, so that observed data must be carefully investigated before their comparison with model outcomes, as they may be addressing different types of damage, damage to different components, or being incomplete. Accordingly, flood damage modellers must be always cautious when drawing conclusions from validation analyses:

Concerning instead the calibration of models with observations, although this is in principle a good idea when models have to be transferred to different spatial contexts, this does interfere with a blind approach. Basically, we wanted to test how well models that have been tested and applied elsewhere perform in a different setting without calibration, like non-expert/non-scientist users would do in practical applications.

R1-C3: Page 7, line 166 and throughout the text. It is not clear if each research group have implemented all the selected models (and there are multiple realizations of the same model) or if each research group have implemented one model.

Answer: to clarify, we changed the sentence at pg. 7 line 175 as follows: "The models were implemented independently by the research groups (i.e. each group applied one up to three models, according to its specific expertise)"

R1-C4: Page 7, step 4. This step is commonly referred to as sensitivity analysis; the authors may evaluate if recalling this definition can improve readability.

Answer: we do not agree with the referee as this is not a sensitivity analysis but an investigation of results. We prefer to keep the current title.

R1-C5: Page 21, line 442. Delete "in, as."

Answer: This typo has been already amended in the previous version of the manuscript.

[revised manuscript text omitted]